# 🔥 InPhyRe Discovers: Large Multimodal Models Struggle in Inductive Physical Reasoning

## Abstract

Large multimodal models (LMMs) encode universal physical laws observed during training, such as momentum conservation, as *parametric knowledge*. It allows LMMs to answer physical reasoning queries, such as the outcome of a potential collision event from visual input. However, since parametric knowledge includes only the physical laws seen during training, it is insufficient for reasoning when the inference scenario follows physical laws unseen during training. In contrast, humans can adapt their physical reasoning to unseen physical environments with only a few visual examples. This *inductive physical reasoning* ability is indispensable for LMMs if they are to replace human agents in safety-critical applications. Despite its importance, existing visual benchmarks evaluate only the parametric knowledge in LMMs, and not inductive physical reasoning. To this end, we propose InPhyRe, the first visual question answering benchmark to measure inductive physical reasoning in LMMs. InPhyRe evaluates LMMs on their ability to predict the outcome of collision events in algorithmically generated synthetic videos. By inspecting over 13 open-source and proprietary LMMs, InPhyRe informs us that **(1)** LMMs struggle to apply their limited parametric knowledge about universal physical laws to reasoning, **(2)** inductive physical reasoning in LMMs is weak when inference scenarios obey physical laws unseen during training, and **(3)** inductive physical reasoning in LMMs suffers from language bias and largely ignores the visual inputs, questioning the trustworthiness of LMMs regarding visual inputs.

## 1    Introduction

> **Case Study**
>
> **Premise**: A large multimodal model (LMM) is used to determine whether a car crash will occur on a snowy road from a video. To ensure the LMM understands that the physical coefficients of the snowy road differ from those of a dry road, a few demonstration videos of snowy roads with and without collisions are provided as context. The LMM predicts that a crash is unlikely.
>
> **Question**: Did the LMM account for the unseen physical coefficients using the demonstration videos, or did it use only its parametric physical knowledge to make its prediction?

Large multimodal models (LMMs) are known to encode universal physical laws (*e.g.*, momentum conservation) observed during training as *parametric knowledge* to answer physical reasoning queries (*e.g.*, whether a collision occurs or not) from visual input (Chen et al., 2024d; Cherian et al., 2024; Mudur et al., 2025). However, since parametric knowledge includes only the physical laws seen during training, it is insufficient in scenarios that potentially follow unobserved physical laws and conditions, such as the case study of the snowy road. In contrast, humans would easily adapt their physical knowledge about collisions to snowy road conditions with the help of demonstration videos to predict any collision, if presented with the same case study. This crucial skill, that we refer to as ***inductive physical reasoning***[1], is a hallmark of intelligence that humans develop at a very young age (Hayes, 2007; Ricco, 2015). Inductive physical reasoning is an indispensable ability that LMMs must possess in addition to parametric knowledge if they are to be deployed in safety-critical applications such as autonomous driving (Zhou et al., 2024; Zhang, 2025) to replace human agents.

---

[1]Inductive physical reasoning is closer to inductive reasoning than general visual reasoning. See § D.

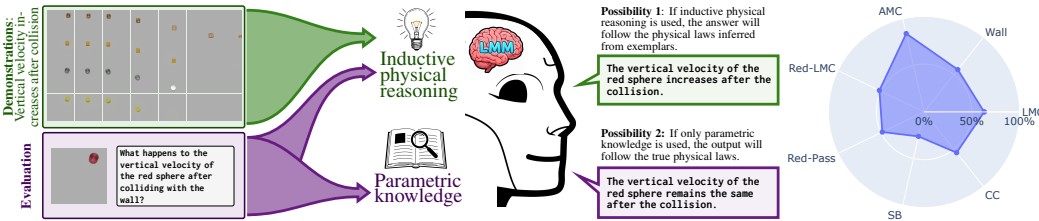

Figure 1: (*Left*) A large multimodal model (LMM) is asked to predict the change in vertical velocity of an object colliding with a vertical wall. The model will output "possibility 2" if it uses its **parametric knowledge** that encodes the universal physical laws (in this case, the momentum conservation principle). However, parametric knowledge would be insufficient if the collision event violated the physical laws encoded in the model. For the model to infer the underlying physical laws, we provide the model with exemplar videos of collisions that violate the momentum conservation principle. The model may now rely on its **inductive physical reasoning** capabilities to generate "possibility 1". (*Right*) INPHYRE shows that LMMs struggle with inductive physical reasoning.

Despite its vital nature, there are no visual benchmarks that quantitatively evaluate inductive physical reasoning in LMMs. Existing benchmarks (Baradel et al., 2020; Chen et al., 2022; Tung et al., 2023; Chow et al., 2025) evaluate only the parametric knowledge of LMMs, particularly about universal physical concepts observable in natural videos, such as friction and gravity, and not the inductive physical reasoning ability. However, creating inductive physical reasoning benchmarks is challenging.

To evaluate inductive physical reasoning separated from parametric knowledge in LMMs, the benchmark must exclude scenarios seen during LMMs' training. Since we cannot always know which scenarios have been seen by LMMs, we opt to build our benchmark using physically impossible scenarios that violate universal physical laws, as they are less likely to have been seen by LMMs during training. However, it is inherently impossible to find such scenarios in natural videos, and prohibitively expensive to manually edit and repurpose existing benchmarks that measure parametric knowledge. Thus, algorithmically generated synthetic videos become the only viable option. Algorithmic generation of physical event videos violating universal physical laws still requires careful design and manual interventions on environment design, modified physical laws, and object trajectories.

We propose **INPHYRE** (Inductive Physical Reasoning), the *first* visual question answering benchmark that evaluates how well LMMs can infer the underlying physics from demonstration samples and use it to make physical reasoning predictions. INPHYRE comprises algorithmically generated synthetic videos of collision events. During evaluation, LMMs are given the first frame of a video and asked questions about the outcome of a textually described collision involving the objects in this frame. However, some of these scenarios violate universal physical laws. In these scenarios, where parametric knowledge is ineffective, LMMs must infer the underlying physics from demonstration videos taken from the same scenarios for physical reasoning. INPHYRE quantifies inductive physical reasoning in LMMs as the performance disparity between scenarios that follow the true physical laws and those that do not. INPHYRE's goal is not to evaluate the utility of LMMs in unrealistic scenarios that violate universal physical laws, but rather to evaluate their adaptability in scenarios with unseen physical laws. This goal also differs from that of "intuitive physics understanding" (see § C).

**Scope of INPHYRE:** Since it is difficult to create a benchmark visualizing the violations of every universal physical law exhaustively, we limit our studies to the laws of mechanics, such as momentum and energy conservation principles. Nonetheless, the outcomes of the collision events in INPHYRE violate the most fundamental laws of mechanics. Therefore, we believe that the conclusions from INPHYRE are likely valid for inductive physical reasoning in LMMs about other branches of physics.

**What does INPHYRE find?** Our results show that INPHYRE is a formidable benchmark for current LMMs, despite its visual simplicity. Our results indicate that both **open-source and proprietary LMMs struggle to infer and utilize unseen physical laws from demonstration samples** (Fig. 1). From empirical evidence, we conjecture that LMMs do not understand physical laws as transferable mathematical models, but rather as a fixed set of rules that all objects obey. Moreover, we show that LMMs primarily derive their inductive physical reasoning capabilities from their language components and ignore the visual inputs in the demonstration samples. Chain-of-thought prompting and fine-tuning are also futile for conditional physical reasoning tasks in INPHYRE (§§ E.3 and E.4). This means that it is possible that the LMM in our case study did not account for the snowy conditions.

> **Contributions and Findings**
>
> ✦ We introduce INPHYRE, the ***first*** visual question answering benchmark to evaluate the inductive physical reasoning capabilities of large multimodal models (LMMs).
>
> ✦ Finding 1: LMMs have only limited parametric knowledge about universal physical laws and struggle to apply these laws even in scenarios that follow these physical laws (§ 4.2).
>
> ✦ Finding 2: Demonstration samples improve LMMs' predictions only when the samples agree with the models' parametric knowledge, resulting in poor inductive physical reasoning on scenarios that violate the true physical laws (§§ 4.3, 4.4 and E.2).
>
> ✦ Finding 3: Inductive physical reasoning in LMMs suffers from strong language bias. As a result, visual inputs in the exemplars play very little role in the final prediction (§ 4.5).

## 2 RELATED WORKS

| Benchmark | What are the physical reasoning tasks | Physical conditions change between training and testing? | Require test-time inference of physical conditions? |
|---|---|---|---|
| CLEVRER [1] | Factual and counterfactual physical reasoning | No ✗ | No ✗ |
| ComPhy [2] | Physical reasoning requiring latent property prediction | No ✗ | Yes ✓ |
| CoPhy [3] | Counterfactual physical reasoning | No ✗ | No ✗ |
| PhysBench [4] | Physical reasoning about object properties and dynamics | No ✗ | No ✗ |
| IntPhys [5] | Physical plausibility prediction | Yes ✓ | No ✗ |
| Physion [6] | Object contact prediction | No ✗ | No ✗ |
| Physion++ [7] | Object contact prediction involving property prediction | No ✗ | No ✗ |
| ContPhy [8] | Physical property and dynamics prediction | No ✗ | No ✗ |
| **INPHYRE** | Infer physical laws from demo samples and apply them. | **Yes ✓** | **Yes ✓** |

Table 1: **How does INPHYRE differ from prior physical reasoning benchmarks?** [1] (Yi et al., 2020), [2] (Chen et al., 2022), [3] (Baradel et al., 2020), [4] (Chow et al., 2025), [5] (Riochet et al., 2021; Bordes et al., 2025), [6] (Bear et al., 2021), [7] (Tung et al., 2023), [8] (Zheng et al., 2024)

The proliferation of large language models (LLMs) has increased focus on zero-shot commonsense physical reasoning, where the objective is to evaluate the ability of LLMs to reason and provide instructions for everyday tasks such as picking up objects or cutting fruits (Bisk et al., 2020; Aroca-Ouellette et al., 2021; Wang et al., 2023). Other physical reasoning benchmarks focus on the theoretical physics knowledge in LLMs (Pang et al., 2025; Mudur et al., 2025; Yu et al., 2025) and their ability to reason about the latent physical properties (Chen et al., 2024d; Chow et al., 2025), sometimes using interactive simulators (Cherian et al., 2024). Synthetic collision events are also commonly used in physical reasoning benchmarks (Yi et al., 2020; Baradel et al., 2020; Chen et al., 2022). However, **these works only evaluate the parametric knowledge** of LLMs about the tools in the evaluation environment, and do not consider inductive physical reasoning in LMMs at all. These datasets are also not suitable to evaluate inductive physical reasoning, as the presence of true physical laws in these datasets may confound the evaluation of inductive physical reasoning. **Example:** Benchmarks that measure counterfactual physical reasoning do not check if the model predicted using its memorized physical laws/conditions, or based on the provided factual scenario. In contrast, by relying on impossible physics, we ensure that LMMs can only answer the queries using the physics dynamics inferred from demonstration samples. Our work is also in contrast with "intuitive physics understanding" (Riochet et al., 2021; Garrido et al., 2025), where parametric knowledge of learned models is evaluated through their ability to detect violations of universal physical laws. The key differences between INPHYRE and some prior physical reasoning benchmarks are shown in Tab. 1. See § C for more discussion on related works.

## 3 INPHYRE: INDUCTIVE PHYSICAL REASONING BENCHMARK

In this section, we will describe our proposed benchmark, INPHYRE – Inductive Physical Reasoning and how we will utilize it to evaluate the inductive physical reasoning capabilities of large multimodal

models (LMMs). Our benchmark comprises collision event videos that violate real-world physical laws, such as momentum conservation and object continuity. During evaluation, the inputs to the LMM are the first frame from a collision video and a multiple-choice question about the outcome of a described collision event. The model is then evaluated in zero-shot and few-shot settings to quantify its relative strengths of parametric knowledge and inductive physical reasoning.

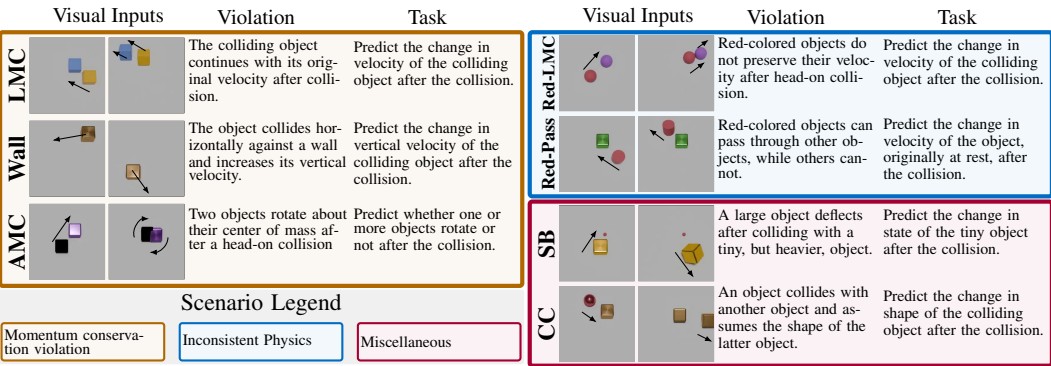

Figure 2: INPHYRE comprises videos ("visual inputs") of collision events that violate a real-world physical law ("violation"). LMMs must predict state changes in objects due to the collisions, while accounting for the violated physical law ("task"). The videos are grouped into "scenarios", which are further grouped into three categories based on the nature of physical law they violate. Arrows indicate object motion and are not part of the actual images in the dataset.

**Research Question**: We will first explain the key prerequisite concepts and then state the research question addressed by INPHYRE. As mentioned before, each evaluation sample consists of the first frame of a collision video and a multiple-choice question about the collision's outcome. The model may answer this question using its implicit knowledge of the physical laws that it obtained during training. We refer to this as physical reasoning using parametric knowledge. However, if the physical reasoning required to correctly answer the question does not follow the physical laws embedded in the model, parametric knowledge is insufficient. To let the model infer the physical laws required to answer the question, we provide demonstration samples[2] containing videos of similar events governed by the same physical laws required to answer the question. This ability to infer the physical laws from exemplars and answer the question is referred to as inductive physical reasoning.

INPHYRE is designed to answer the following research question about physical reasoning in LMMs:

> **(RQ)** Can LMMs flexibly switch between parametric knowledge and inductive physical reasoning by comparing the physical laws in the exemplars to those encoded in the models' parameters?

To answer **(RQ)**, INPHYRE evaluates LMMs in scenarios that follow the true physical laws and those that do not. An LMM with strong inductive physical reasoning will perform identically in both situations, as parametric knowledge and inductive physical reasoning are not competing qualities.

**Task Description**: INPHYRE is a visual question answering benchmark for physical reasoning. Each evaluation sample includes an image of a scene with one or more objects and a question about the outcome of a described collision event involving the objects in that image. Similar to (Johnson et al., 2017; Yi et al., 2020; Chen et al., 2022), the objects are primitive shapes (cubes, cylinders, and spheres) of various colors and textures lying on a plain surface (Fig. 2). The question is about the change in the state of an object after the collision. *E.g.*, `What happens to the velocity of the red cube after colliding with the blue sphere?` To answer the question, we provide four options as possible answers to the model. E.g., "`A. red cube's velocity increases, B. red cube's velocity decreases, C. cannot be determined, D. no change in velocity.`" The questions may also contain additional information about the objects or the collision event. E.g., "`red cube and yellow cylinder have equal mass`" or "`blue sphere collides elastically against the wall.`" The option chosen by the model is parsed from its generation output. See §§ B.2 and B.3 for more details about prompting and parsing, respectively.

---

[2]Henceforth referred to as "exemplars".

**Irregular Scenarios**: The collision event videos in INPHYRE are grouped into "scenarios." Each scenario is characterized by the true physical law that it violates. We call them "irregular scenarios" and denote them with their shorthand notations shown in Fig. 2. The irregular scenarios are further grouped into three categories based on the nature of the violated physical laws (see Fig. 2):

**(1)** Scenarios in the momentum conservation violation category evaluate the inductive physical reasoning of LMMs when the principle of momentum conservation is violated. It comprises three scenarios: linear momentum conservation (LMC), angular momentum conservation (AMC), and directional linear momentum conservation (Wall). In LMC, a moving object collides elastically with an object of equal mass at rest, and, instead of losing its momentum, continues with the same velocity. A similar collision event occurs in AMC, except the objects rotate about their center of mass, despite the collision being head-on, violating the principle of angular momentum conservation. In Wall, the vertical velocity of an object increases after colliding with a vertical wall, violating the linear momentum conservation principle, but only along the vertical direction.

**(2)** In the inconsistent physics category, objects with certain visual properties follow physical laws different from other objects. In the real world, these visual properties would not have affected the modified physical laws. The objective is to examine whether LMMs can logically combine parametric knowledge and inductive physical reasoning based on the object's visual properties. For this category, few-shot evaluations will include exemplars with both sets of physical laws. This category includes two scenarios: (i) Red-LMC, where red-colored objects violate linear momentum conservation, and (ii) Red-Pass, where only red-colored objects can physically pass through other objects.

**(3)** INPHYRE also includes some miscellaneous scenarios to evaluate whether LMMs have a visually biased perception of physical laws. For instance, in size-bias (SB), a dimensionally large object deflects after colliding with a dimensionally small, but much heavier, object. We include information about mass in the question. SB evaluates whether LMMs conflate the concepts of volume and mass. In color-constancy (CC), a moving object collides with an object at rest and assumes the visual appearance of the latter object. This collision obeys linear momentum conservation. CC evaluates the object permanence of LMMs as it may seem to the model that the colliding object has disappeared.

**Regular Scenarios**: For each irregular scenario, there is a corresponding "regular" scenario depicting similar collisions while following the true physical laws. Since multiple types of violations are possible for every universal physical law, regular scenarios are fewer than irregular scenarios. INPHYRE includes regular versions of LMC, SB, and AMC that act as real-world counterparts of various irregular scenarios. These regular scenarios are used to evaluate the parametric knowledge of LMMs. INPHYRE uses both regular and irregular scenarios jointly to answer **(RQ)**.

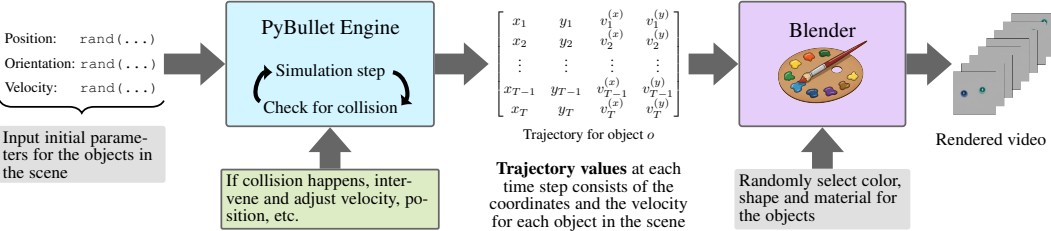

Figure 3: We initialize the object states in PyBullet. When a collision occurs during the simulation, we intervene and manually adjust the objects' states such that the resulting trajectory violates some real-world physical law. The object trajectories are then used by Blender to render the final video.

**Video generation details**: We modify the video generation pipeline from (Yi et al., 2020) to generate synthetic videos using PyBullet (Coumans & Bai, 2016–2021) and Blender (Community, 2025). First, we define objects and their properties such as mass and lateral friction. Then we randomly initialize their state variables, such as position and velocity, in a PyBullet environment. The trajectories are obtained by running the simulation. However, unlike in (Yi et al., 2020), our trajectories are governed by custom physical laws that differ from real-world physical laws. To simulate these custom physical laws, we intervene when collisions occur during simulation and manually adjust states of objects such as linear/angular velocity and direction. These trajectories are then used by Blender to render the final video. Visual object properties such as color and texture are randomly chosen. See Fig. 3.

**Question generation details**: Each scenario comprises around 2000 samples, of which 10 randomly chosen samples are set aside as exemplars. Question-answer pairs are generated for each sample from pre-defined templates. Since each scenario concerns a particular query (e.g., change in the vertical velocity in Wall), we use multiple templates for the questions and the answer options to avoid lexical repetition. For instance, in Wall, the templates for the question are {"What happens to the vertical velocity of <obj> when it collides with wall?", "What is the outcome of <obj> colliding with wall?", "What occurs to the vertical velocity of <obj> when <obj> and wall collide?"}. A similar multi-template approach is used to generate answers. The answer options are also shuffled for each sample so that the model may not simply repeat the answer options from the exemplars.

## 4 WHAT DOES INPHYRE DISCOVER ABOUT PHYSICAL REASONING IN LMMS?

Before we examine the physical reasoning abilities of LMMs, we will describe the evaluation setup and codify the procedure to answer specific queries about physical reasoning using INPHYRE.

### 4.1 EVALUATION SETUP

**Evaluated LMMs**: We use INPHYRE to evaluate the quality of physical reasoning (both parametric and inductive) in a diverse cohort of LMMs. To represent the variety of choices in model design and training datasets, we include **13 open-source LMMs**: LLaVA-NeXT-Video (Zhang et al., 2024), LLaVA-OneVision (Li et al., 2024a), LLaVA-NeXT-Interleave (Li et al., 2024c), Gemma 3 (Kamath et al., 2025) herd, Aria (Li et al., 2024b), VideoLLaMA3 (Zhang et al., 2025) herd, InternVL3 (Zhu et al., 2025) herd, Qwen2-VL (Wang et al., 2024a), and Qwen2.5-Omni (Xu et al., 2025). The chosen models use different vision encoders and language models, and cover an extensive parameter count range (from 1B to nearly 25B). The image encoders in the chosen LMMs are pre-trained and then fine-tuned. The majority of the chosen LMMs fine-tune separately trained LLMs, while Gemma and Aria train their language models from scratch. Aria uses a mixture-of-experts (MoE) architecture. More details are listed in Tab. 3. **Results on closed models** such as GPT-4 and Gemini are in § E.2.

**Evaluation tools**: INPHYRE contains regular scenarios that follow real-world physical laws and irregular scenarios that violate one or more real-world physical laws. For each LMM, we conduct zero-shot evaluation in regular scenarios and few-shot evaluations in both regular and irregular scenarios. The few-shot setting is further categorized into two sub-settings: (i) "visual-text", where the exemplars contain collision videos along with a question-answer pair, and (ii) "visual-only", where the exemplars include only videos. In all settings, the evaluation metric is the model's accuracy in choosing the correct option for the multiple-choice question. Below, we enunciate our specific questions about physical reasoning in LMMs and how to quantitatively answer them using INPHYRE:

§ 4.2 *How much parametric knowledge do LMMs have about universal physical laws?* We answer this using the zero-shot predictive accuracy of the model in regular scenarios that follow universal physical laws.

§ 4.3 *Can LMMs augment their parametric knowledge with exemplars?* To answer this question, we compare the few-shot performance of the model in regular scenarios with the zero-shot accuracy. Here, exemplars are taken from the evaluated regular scenario.

§ 4.4 *How strong is inductive physical reasoning in LMMs?* Inductive physical reasoning in LMMs is evaluated by comparing their few-shot performances in regular and irregular scenarios.

§ 4.5 *How much of this inductive physical reasoning is aided by language?* We answer this question by computing the difference between the few-shot performances of the model in irregular scenarios under "video-text" and "video-only" settings.

### 4.2 HOW MUCH PARAMETRIC KNOWLEDGE DO LMMS HAVE ABOUT UNIVERSAL PHYSICS?

INPHYRE includes regular versions of LMC, SB, and Wall that obey universal physical laws of mechanics, such as principles of momentum and energy conservation. We measure the parametric knowledge about these universal laws in LMMs as their zero-shot accuracy in these regular scenarios.

| LMM | LMC (Regular) | | SB (Regular) | | Wall (Regular) | | Average | |
|---|---|---|---|---|---|---|---|---|
| | Zero-shot | 3-shot | Zero-shot | 3-shot | Zero-shot | 3-shot | Zero-shot | 3-shot |
| InternVL3-1B (Zhu et al., 2025) | 71.46 | 33.27 (-38.19) | 87.42 | 47.54 (-39.88) | 5.68 | 9.30 (+ 3.62) | 54.85 | 30.03 (-24.82) |
| VideoLLaMA3-2B (Zhang et al., 2025) | 56.13 | 62.96 (+ 6.83) | 51.70 | 63.93 (+12.23) | 3.72 | 37.69 (+33.97) | 37.18 | 54.86 (+17.68) |
| InternVL3-2B (Zhu et al., 2025) | 77.89 | 90.00 (+12.11) | 66.87 | 76.92 (+10.05) | 29.90 | 91.61 (+61.71) | 58.22 | 86.17 (+27.95) |
| Gemma 3-4B (Kamath et al., 2025) | 20.35 | 52.46 (+32.11) | 58.35 | 69.51 (+11.16) | 77.69 | 74.42 (- 3.27) | 52.13 | 65.46 (+13.34) |
| LLaVA-NeXT-Vid (Zhang et al., 2024) | 50.10 | 56.18 (+ 6.08) | 16.44 | 35.06 (+18.62) | 2.46 | 7.59 (+ 5.13) | 23.00 | 32.94 (+ 9.94) |
| InternVL3-8B (Zhu et al., 2025) | 60.60 | 99.65 (+39.05) | 94.47 | 99.85 (+ 5.38) | 57.19 | 98.74 (+41.56) | 70.75 | 99.41 (+28.66) |
| LLaVA-OneVision (Li et al., 2024a) | 75.83 | 99.10 (+23.27) | 83.97 | 99.29 (+15.32) | 4.57 | 99.75 (+95.18) | 54.79 | 99.38 (+44.59) |
| VideoLLaMA3-7B (Zhang et al., 2025) | 78.69 | 98.04 (+19.35) | 69.10 | 93.46 (+24.35) | 7.44 | 51.76 (+44.32) | 51.74 | 81.08 (+29.34) |
| LLaVA-NeXT-IL (Li et al., 2024c) | 83.42 | 97.14 (+13.72) | 65.20 | 63.72 (- 1.47) | 0.55 | 78.84 (+78.29) | 49.72 | 79.90 (+30.18) |
| Qwen2-VL (Wang et al., 2024a) | 66.83 | 98.14 (+31.31) | 73.57 | 98.73 (+25.16) | 7.09 | 99.25 (+92.16) | 49.16 | 98.71 (+49.54) |
| Qwen2.5-Omni (Xu et al., 2025) | 57.99 | 45.83 (-12.16) | 94.52 | 86.66 (- 7.86) | 3.72 | 44.27 (+40.55) | 52.08 | 58.92 (+ 6.84) |
| Gemma 3-12B (Kamath et al., 2025) | 35.68 | 85.33 (+49.65) | 86.71 | 76.51 (-10.20) | 68.84 | 60.75 (- 8.09) | 63.74 | 74.20 (+10.45) |
| Aria (Li et al., 2024b) | 39.70 | 53.12 (+13.42) | 77.32 | 92.29 (+14.97) | 36.03 | 65.88 (+29.85) | 51.02 | 70.43 (+19.41) |

Table 2: Zero-shot and 3-shot evaluation results on regular scenarios.

Tab. 2 shows the zero-shot accuracy of LMMs in regular scenarios. In each scenario, the task was to predict the change in velocity of an object after colliding with another object at rest from the initial image frame of the collision video. Surprisingly, **many LMMs struggle to answer even these simple questions using the momentum conservation principle**, and the models achieve above 80% accuracy in only 6 out of 39 scenarios, mainly in SB (Reg.). The performances of LMMs also vary greatly between scenarios, despite these scenarios following the same physical laws, *e.g.*, most models performed poorly in Wall (Reg.) compared to other scenarios. Qualitative inspection of their outputs in § E.6 reveals that LMMs can state universal physical laws (*e.g.*, "kinetic energy is conserved in an elastic collision") but struggle to apply them for physical reasoning. They also hallucinate irrelevant assumptions (*e.g.*, about material) and incorrect physical laws that further hurt their reasoning. We conclude that LMMs memorize the laws of mechanics and can recollect them as factual information, but fail to apply this knowledge for physical reasoning. A similar conclusion was made in (Yu et al., 2025), but for abstract physical reasoning.

> **CONCLUSION**
>
> LMMs have limited parametric knowledge about the laws of mechanics. Although LMMs can state these universal laws, they often struggle to apply them for physical reasoning.

## 4.3 CAN LMMS AUGMENT THEIR PARAMETRIC KNOWLEDGE WITH EXEMPLARS?

Before evaluating the inductive physical reasoning of LMMs in irregular scenarios, we must verify that exemplars can improve physical reasoning in LMMs. Therefore, we evaluate LMMs in regular scenarios in the few-shot setting with exemplars that do not contradict any universal physical laws encoded in the models' parameters. Specifically, we consider the "visual-text" setting, where question-answer pairs accompany videos in exemplars. Then we compare the few-shot performance of LMMs in regular scenarios with their corresponding zero-shot performance.

Tab. 2 compares the 3-shot performance of LMMs in regular scenarios with their corresponding zero-shot performance. **All LMMs significantly improved their performance when provided with exemplars in at least one scenario.** On average, we observe that all models except InternVL3-1B improved their performance with exemplars. Among the LMMs evaluated, Qwen2-VL achieved the highest average increase in performance with exemplars. LLaVA-Onevision, Qwen2-VL, and InternVL3-8B also achieved nearly 100% average accuracy over all scenarios. These results clearly demonstrate that LMMs can use exemplars to improve their prediction accuracy.

> **CONCLUSION**
>
> Exemplars that obeyed universal physical laws support parametric knowledge in LMMs successfully. With only three exemplars, several LMMs achieve nearly 100% prediction accuracy.

## 4.4 HOW STRONG IS INDUCTIVE PHYSICAL REASONING IN LMMS?

We established that exemplars that obey universal physical laws improve the performance of LMMs in regular scenarios. We will now evaluate whether LMMs can leverage exemplars that do not follow the true physical laws to reason in irregular scenarios. Following the previous experiments, the task is to predict the outcome of a potential collision event from an image with the help of exemplar videos

from the same scenario. Exemplars also include question-answer pairs. However, these collision events (and the provided exemplars) violate the true physical laws. Therefore, to correctly answer the questions, LMMs must infer the underlying physical laws from exemplars through inductive physical reasoning.

Fig. 4 compares 3-shot accuracy of each combination of model and irregular scenario against that model's best performance among zero-shot and few-shot evaluations in the corresponding regular scenario that depicts similar events while following universal physical laws. For SB and Wall, the corresponding regular scenarios are SB (Reg.) and Wall (Reg.), respectively. For all other irregular scenarios, LMC (Reg.) is the corresponding regular scenario. A negative value means poor inductive physical reasoning in that scenario. In Fig. 4, we observe that **most models show a drop in accuracy compared to regular scenarios, indicating weak inductive physical reasoning**. However, the drop in accuracy varies with

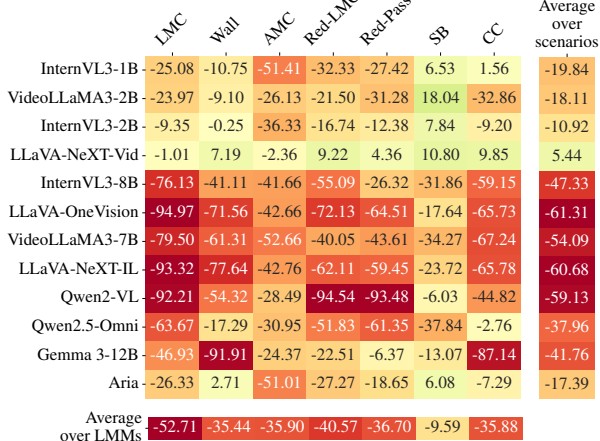

| | LMC | Wall | AMC | Red-LMC | Red-Pass | SB | CC | Average over scenarios |
|---|---|---|---|---|---|---|---|---|
| InternVL3-1B | -23.22 | 9.20 | 11.76 | -13.01 | -15.47 | -76.97 | -23.52 | -18.75 |
| VideoLLaMA3-2B | -25.83 | -15.33 | -3.62 | -27.28 | -13.49 | -55.84 | -16.93 | -22.62 |
| InternVL3-2B | -74.92 | -91.36 | 9.05 | -64.94 | -68.90 | -76.81 | -80.65 | -64.08 |
| Gemma 3-4B | 33.92 | -21.56 | 7.24 | 28.19 | 9.64 | -44.88 | -15.38 | -0.40 |
| LLaVA-NeXT-Vid | -39.50 | 27.69 | -10.60 | -48.56 | -44.40 | -16.62 | -24.57 | -22.37 |
| InternVL3-8B | -5.33 | -0.20 | 0.35 | -27.87 | -52.38 | -53.87 | -30.70 | -24.29 |
| LLaVA-OneVision | -0.90 | -2.56 | 0.30 | -24.16 | -31.38 | -67.88 | -33.32 | -22.84 |
| VideoLLaMA3-7B | -14.42 | 29.65 | -0.30 | -48.97 | -44.01 | -29.28 | -21.76 | -18.44 |
| LLaVA-NeXT-IL | -3.42 | 15.33 | -4.97 | -34.63 | -37.29 | -21.48 | 0.95 | -12.21 |
| Qwen2-VL | -2.11 | 0.75 | -2.81 | 1.01 | -1.45 | -84.96 | -7.79 | -13.91 |
| Qwen2.5-Omni | 5.83 | -4.97 | 32.11 | -4.36 | 5.62 | -38.29 | -13.72 | -2.54 |
| Gemma 3-12B | -28.04 | 23.52 | 6.18 | -57.66 | -63.92 | -68.62 | 1.81 | -26.68 |
| Aria | -25.88 | -65.18 | 27.24 | -24.54 | -33.17 | -85.35 | -45.63 | -36.07 |
| Average over LMMs | -15.68 | -7.31 | 5.53 | -26.67 | -30.04 | -55.45 | -23.94 | |

Figure 4: Difference in 3-shot accuracy of LMMs between irregular and regular scenarios when exemplars contain both videos and QA pairs.

the models. InternVL3-2B and Gemma3-4B show the highest and the lowest average drop in accuracy, respectively. Performance deterioration also varies with the scenario. Almost all models suffer a considerable drop in accuracy in SB, indicating that LMMs struggle to differentiate between volume and mass. Surprisingly, several LMMs performed better in AMC than LMC (Reg.). However, as we show in § E.7, this apparent inductive physical reasoning is due to LMMs possessing the wrong parametric knowledge about angular momentum conservation, thus inadvertently performing well in irregular scenarios. The absolute accuracy values are given in § B.4.

> **CONCLUSION**
>
> LMMs demonstrate only weak inductive physical reasoning when exemplars violate parametric knowledge. Almost all LMMs showed significant deterioration in performance.

### 4.5 HOW MUCH OF THIS INDUCTIVE PHYSICAL REASONING IS AIDED BY LANGUAGE?

In our previous experiments, exemplars included both videos and question-answer pairs. However, multimodal models are well known to exploit their language bias in visual question answering (VQA) tasks (Goyal et al., 2017). This raises the possibility that the observed inductive physical reasoning, limited as it may be, originated primarily from the language component of LMMs. The existence of language bias can undermine the trustworthiness of LMMs on visual inputs. To detect the presence of language bias, we repeat our few-shot experiments from § 4.4 but with exemplars that contain *only* the collision videos. This setup entirely restricts LMMs to their visual inductive physical reasoning capabilities. Follow-

| | LMC | Wall | AMC | Red-LMC | Red-Pass | SB | CC | Average over scenarios |
|---|---|---|---|---|---|---|---|---|
| InternVL3-1B | -25.08 | -10.75 | -51.41 | -32.33 | -27.42 | 6.53 | 1.56 | -19.84 |
| VideoLLaMA3-2B | -23.97 | -9.10 | -26.13 | -21.50 | -31.28 | 18.04 | -32.86 | -18.11 |
| InternVL3-2B | -9.35 | -0.25 | -36.33 | -16.74 | -12.38 | 7.84 | -9.20 | -10.92 |
| LLaVA-NeXT-Vid | -1.01 | 7.19 | -2.36 | 9.22 | 4.36 | 10.80 | 9.85 | 5.44 |
| InternVL3-8B | -76.13 | -41.11 | -41.66 | -55.09 | -26.32 | -31.86 | -59.15 | -47.33 |
| LLaVA-OneVision | -94.97 | -71.56 | -42.66 | -72.13 | -64.51 | -17.64 | -65.73 | -61.31 |
| VideoLLaMA3-7B | -79.50 | -61.31 | -52.66 | -40.05 | -43.61 | -34.27 | -67.24 | -54.09 |
| LLaVA-NeXT-IL | -93.32 | -77.64 | -42.76 | -62.11 | -59.45 | -23.72 | -65.78 | -60.68 |
| Qwen2-VL | -92.21 | -54.32 | -28.49 | -94.54 | -93.48 | -6.03 | -44.82 | -59.13 |
| Qwen2.5-Omni | -63.67 | -17.29 | -30.95 | -51.83 | -61.35 | -37.84 | -2.76 | -37.96 |
| Gemma 3-12B | -46.93 | -91.91 | -24.37 | -22.51 | -6.37 | -13.07 | -87.14 | -41.76 |
| Aria | -26.33 | 2.71 | -51.01 | -27.27 | -18.65 | 6.08 | -7.29 | -17.39 |
| Average over LMMs | -52.71 | -35.44 | -35.90 | -40.57 | -36.70 | -9.59 | -35.88 | |

Figure 5: Difference in 3-shot accuracy in irregular scenarios between video-only and video-text settings.

ing (Min et al., 2022), we include randomly chosen options in the exemplars to entice the model to choose an option instead of providing open-ended reasoning (see § B.2).

Fig. 5 shows the difference between performances under video-only and video-text settings. We find that the **inductive physical reasoning in LMMs, unfortunately, arises largely from the language**

**components** of LMMs. When the exemplars contained only videos, almost all LMMs showed a significant drop in accuracy compared to their performance when the exemplars contained both videos and question-answer pairs. In certain scenarios, LMMs could achieve only near-zero accuracy, *e.g.*, Qwen2.5-Omni in LMC and LLaVA-Next-Interleave in Red-LMC and Red-Pass. The accuracy drop was higher for larger models. Our findings echo the concurrent evidence of language bias in LMMs reported by (Baldassini et al., 2024; Chen et al., 2025), although their findings were not about physical reasoning in LMMs.

> **CONCLUSION**
>
> Inductive physical reasoning in the evaluated LMMs show strong language bias, relying primarily on the textual content of the exemplars to answer the question. The presence of language bias questions the trustworthiness of LMMs in scenarios with unseen physical laws.

## 5 FURTHER DISCUSSION

We investigate the effect of the exemplar retrieval method and the number of retrieved exemplars in this section. Refer to § E for experiments on quantized models, a visually more complex version of INPHYRE, evaluation with all evaluation sample frames included, effects of prompt perturbations, and qualitative results on INPHYRE.

**Effect of number of exemplars**: The number of exemplars in an evaluation sample is limited by the context length in which the models were trained/fine-tuned. Our few-shot experiments in § 4 used three exemplars per evaluation sample. In this section, we will examine if the number of exemplars significantly affects performance on INPHYRE. We vary the number of exemplars from 1 to 3 and average the prediction accuracy over the evaluated

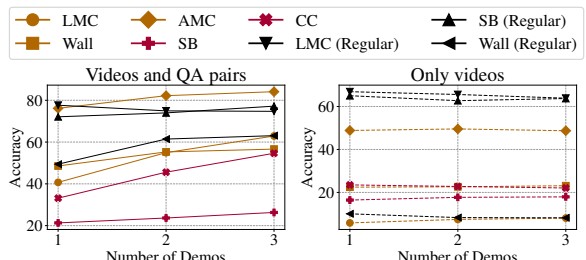

Figure 6: Effect of number of exemplars on the accuracy.

LMMs for each scenario. We evaluate on both "video-text" and "video-only" settings to investigate potential language bias. Fig. 6 (left) shows the change in prediction accuracy with the number of exemplars when the exemplars contain both videos and question-answer pairs. Across all scenarios, we observe a clear improvement in predictive performance. In particular, LMC and CC show significant improvements in accuracy. However, similar improvement is not present in Fig. 6 (right), where exemplars contained only videos. Performance even deteriorated in LMC (Reg.) as the number of exemplars increased, although the remaining scenarios were unaffected. These results reaffirm our observation about the language bias in LMMs.

**We evaluate human performance** on INPHYRE using ten human subjects. Each subject was provided with one demonstration sample from each scenario, without the accompanying question-answer pair. For Red-LMC and Red-Pass, they were provided four demonstration samples since they were required to infer conditional reasoning rules. Then, they were asked to answer one evaluation query from the same scenario. We asked them to answer only one query since all queries in a scenario shared the underlying physical logic. Despite being provided only demonstration samples without textual information, the subjects scored above 90% accuracy in many scenarios. They struggled relatively more in Red-LMC, Red-Pass, and CC. Detailed subject-wise results are provided in § E.9.

**Effect of CoT prompting and fine-tuning**: Chain-of-thought (CoT) prompting (Wei et al., 2022) and fine-tuning (FT) have been shown to improve reasoning in LMMs (Buschoff et al., 2025). However, the underlying physical law and the samples themselves must be available in advance for CoT prompting and fine-tuning, respectively. Therefore, these paradigms are not suitable for the premise of inductive physical reasoning, which posits that we have access to only visual samples from unseen scenarios immediately before inference. Nonetheless, we still evaluate the effects of CoT and FT on INPHYRE to obtain a "soft upper bound" on the LMMs' performance. The results with CoT prompting and FT are shown in Tabs. 8 and 9, respectively. We find that both CoT and FT can generally improve performance in most scenarios, except Red-Pass and Red-LMC, which, unlike other scenarios, require conditional physical reasoning, signaling the utility limits of CoT prompting and FT. As a reminder, in these scenarios, only red-colored objects violate the true physical laws.

**Causes of poor performance in irregular scenarios**: We provide a preliminary analysis of the causes of poor performance in irregular scenarios in § F. To this end, we use linear probes on the hidden states from pre-trained and fine-tuned InternVL3-1B, and visualize attention values over the tokens in the last layer of Gemma3-12B. We find that the hidden states of both pre-trained and fine-tuned models carry sufficient information to classify the underlying scenario (Fig. 25). Moreover, after fine-tuning, the hidden states adaptively include attribute information from demonstration and evaluation samples depending on the underlying scenario (Figs. 26 and 27). Visualization of attention values from the last layer of Gemma3-12B shows that the model spends an order of magnitude less attention over image tokens compared to text tokens (Fig. 24). We believe that these findings will assist in developing methods to explicitly improve inductive physical reasoning.

**Effect of exemplar retrieval method**: The choice of retrieved samples could affect the performance of LLMs (Liu et al., 2022; Peng et al., 2024). We now verify if this proposition holds for visual inductive physical reasoning. By adapting the textual exemplar retriever from (Liu et al., 2022) for vision, we design a "nearest-neighbor exemplar retriever" (NNER) that finds the top-$k$ video samples closest to the initial frame of the evaluation sample according to their cosine distance in the feature space of CLIP-L (Radford et al., 2021). To evaluate the effect of retrievers on visual inductive physical reasoning, we include only videos in exemplars. We do not evaluate on Red-LMC and Red-Pass since NNER does not guarantee

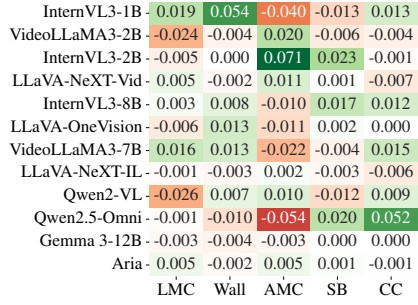

Figure 7: Change in accuracy when NNER is used to find exemplars.

that the retrieved samples include videos with and without red-colored objects. In Fig. 7, we observe either an insignificant or no change in performance between a random retriever and NNER, agreeing with our observation that LMMs rely primarily on language for inductive physical reasoning.

## 6 CONCLUSION

When inference scenarios violate the physical laws encoded in the model parameters, LMMs must ideally derive their physical reasoning from demonstration samples. Therefore, to ensure their trustworthiness, it is critical to evaluate how well LMMs can infer physical laws from these exemplars. To this end, we introduced INPHYRE, the ***first*** visual question answering benchmark to quantify parametric knowledge and inductive physical reasoning in LMMs. INPHYRE evaluates LMMs in collision scenarios that violate universal physical laws such as momentum conservation. Through zero-shot and few-shot experiments in these scenarios, we found that LMMs have limited parametric knowledge of universal physical laws and struggle to apply these laws during physical reasoning. LMMs demonstrate only weak inductive physical reasoning when exemplars violate universal physical laws. The observed inductive physical reasoning also suffered from language bias and relied little on visual input in exemplars, shedding doubt on the trustworthiness of LMMs on visual input.

**Limitations**: Although INPHYRE proved to be a formidable benchmark for LMMs in terms of physical laws, its simple synthetic scene could not have posed visual challenges to the models. Inductive physical reasoning of LMMs could be worse in a more crowded and realistic visual scene. However, since real-world videos cannot violate true physical laws, the closest alternative to evaluate inductive physical reasoning in LMMs is a hyperrealistic video benchmark generated using advanced video generation models (OpenAI, 2024; Kondratyuk et al., 2024; Chen et al., 2024b). However, such endeavors can be fruitful only with absolute control over physical realism, which is lacking in current video generation models (Cho et al., 2024; Motamed et al., 2025).

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

# Appendix

## Table of Contents

## A   DATA-GENERATION DETAILS

### A.1   RENDERING THE VIDEOS

We use Blender's Python wrapper[3] (v4.4.0) to render the video from the trajectories. We modified the image rendering code[4] from (Johnson et al., 2017). We first designed a base scene with lamp and camera positions suitable for capturing entire object trajectories in INPHYRE. The object textures were taken from (Johnson et al., 2017), but more hues were added. Each video consists of 240 frames from which 8 frames are uniformly sampled to form the video tokens by the corresponding video processors of the LMMs.

---

[3]https://pypi.org/project/bpy/
[4]https://github.com/facebookresearch/clevr-dataset-gen

A sample video from each scenario is shown below: regular scenarios in Figs. 8 to 10, momentum conservation violation scenarios in Figs. 11 to 13, inconsistency physics scenarios in Figs. 16 and 17, and miscellaneous irregular scenarios in Figs. 14 and 15.

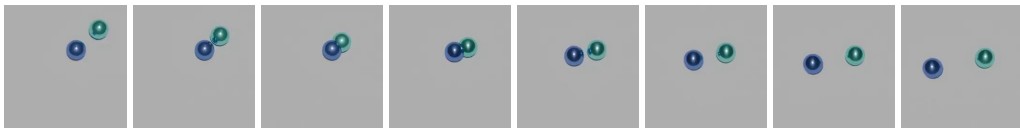

Figure 8: Regular scenario where linear momentum conservation is followed – LMC (Regular)

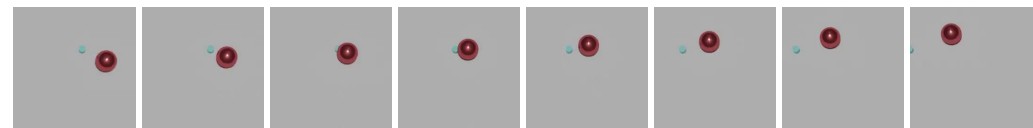

Figure 9: Regular scenario where the larger object has more mass – SB (Regular)

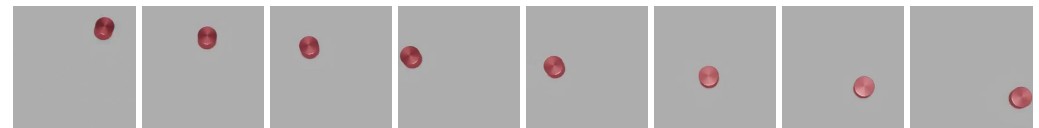

Figure 10: Regular scenario where linear momentum is conserved along the vertical direction – Wall (Regular)

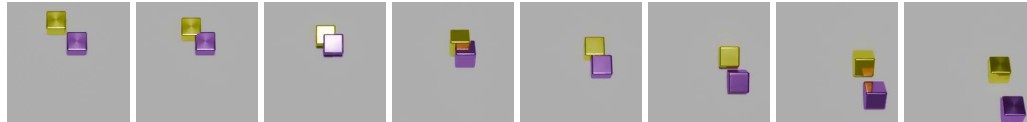

Figure 11: Irregular scenario where linear momentum conservation is violated – LMC

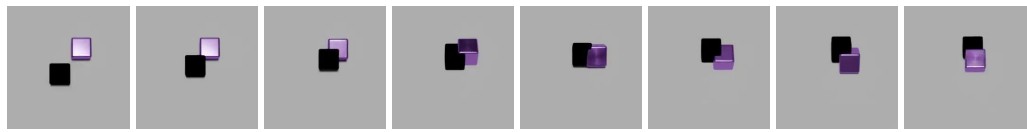

Figure 12: Irregular scenario where angular momentum conservation is violated – AMC

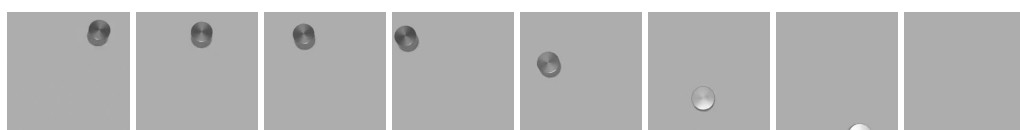

Figure 13: Irregular scenario where linear momentum conservation is violated along the vertical direction – Wall

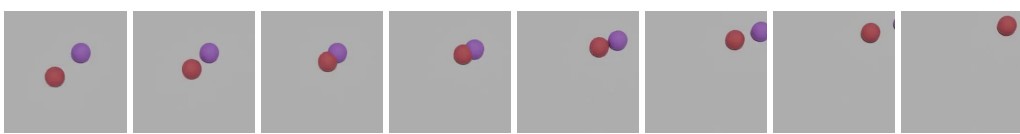

Figure 14: Irregular scenario where only red-colored objects violate linear momentum conservation – Red-LMC

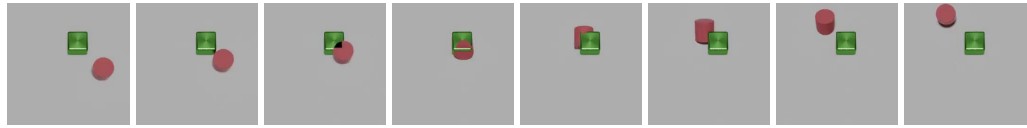

Figure 15: Irregular scenario where red-colored objects can pass through other objects – Red-Pass

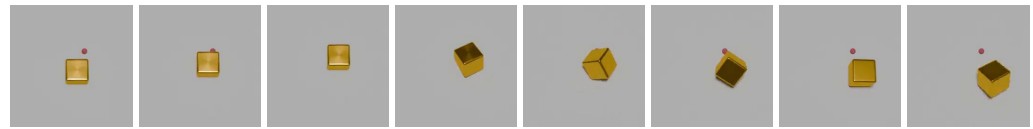

Figure 16: Irregular scenario where a large object deflects after colliding with a tiny, much heavier, object – SB

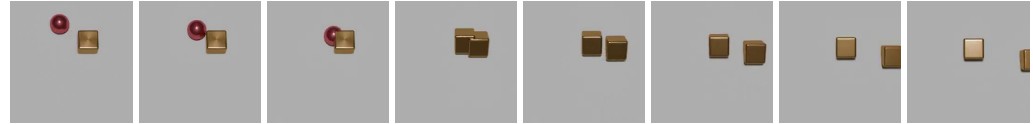

Figure 17: Irregular scenario where the colliding object assumes the hue and the shape of the other object after collision – CC

## B  DETAILS OF EXPERIMENTAL SETUP AND EVALUATION

### B.1  EVALUATED LMMS

We evaluate a diverse collection of LMMs trained on both public and proprietary datasets with different architectural design choices. The model weights were taken from Huggingface[5]. The models are listed in Tab. 3.

### B.2  PROMPTING METHODS

For prompting, we used the `apply_chat_template` method to convert conversations from a list of Python dictionaries to LMM-specific prompts. This helps in reusing the code. An example from LMC (regular) with one exemplar containing both video and question-answer pair in the conversation style of LLaVA-NeXT-Video is given below:

```
[
    {
    "role": "system",
    "content": [{"type": "text",
                "text": "Understand the underlying physics from the following videos and
                    choose only an option among A, B, C and D to answer the question. Do
                    not provide reasoning."}]
    },
    {
    "role": "user",
    "content": [{"type": "video", "path": "path/to/exemplar_video.mp4"},
```

---

[5]https://huggingface.co/models

| Model | HF ID | #Params (B) | Vision encoder | Language model | MoE |
|---|---|---|---|---|---|
| InternVL3 (1B) (Zhu et al., 2025) | OpenGVLab/InternVL3-1B-hf | 0.9 | InternViT (Chen et al., 2024c) | Qwen2.5 (Yang et al., 2024b) | ✗ |
| InternVL3 (2B) (Zhu et al., 2025) | OpenGVLab/InternVL3-2B-hf | 1.9 | InternViT (Chen et al., 2024c) | Qwen2.5 (Yang et al., 2024b) | ✗ |
| VideoLLaMA3 (2B) (Zhang et al., 2025) | DAMO-NLP-SG/VideoLLaMA3-2B | 2 | SigLIP (Zhai et al., 2023) | Qwen2.5 (Yang et al., 2024b) | ✗ |
| Gemma 3 (4B) (Kamath et al., 2025) | google/gemma-3-4b-it | 4 | SigLIP (Zhai et al., 2023) | * | ✗ |
| LLaVA-NeXT-Video (Zhang et al., 2024) | llava-hf/LLaVA-NeXT-Video-7B-hf | 7 | CLIP-L (Radford et al., 2021) | Vicuna 1.5 (Zheng et al., 2023) | ✗ |
| LLaVA-OneVision (Li et al., 2024a) | llava-hf/llava-onevision-qwen-7b-si-hf | 7 | SigLIP (Zhai et al., 2023) | Qwen2 (Yang et al., 2024a) | ✗ |
| LLaVA-NeXT-Interleave (Li et al., 2024c) | llava-hf/llava-interleave-qwen-7b-hf | 7 | SigLIP (Zhai et al., 2023) | Qwen1.5 (Bai et al., 2023a) | ✗ |
| VideoLLaMA3 (7B) (Zhang et al., 2025) | DAMO-NLP-SG/VideoLLaMA3-7B | 7 | SigLIP Zhai et al. (2023) | Qwen2.5 (Yang et al., 2024b) | ✗ |
| Qwen2-VL (Wang et al., 2024a) | Qwen/Qwen2-VL-7B-Instruct | 7 | Qwen-VL (Bai et al., 2023a) | Qwen2 (Yang et al., 2024a) | ✗ |
| Qwen2.5-Omni (Xu et al., 2025) | Qwen/Qwen2.5-Omni-7B | 7 | Qwen2.5-VL (Bai et al., 2025) | Qwen2.5 (Yang et al., 2024b) | ✗ |
| InternVL3 (8B) (Zhu et al., 2025) | OpenGVLab/InternVL3-8B-hf | 8.1 | InternViT (Chen et al., 2024c) | Qwen2.5 (Yang et al., 2024b) | ✗ |
| Gemma 3 (12B) (Kamath et al., 2025) | google/gemma-3-12b-it | 12 | SigLIP (Zhai et al., 2023) | * | ✗ |
| Aria (Li et al., 2024b) | rhymes-ai/Aria | 24.9 | SigLIP (Zhai et al., 2023) | * | ✓ |

Table 3: We evaluate a diverse assortment of LMMs with varying architectural and data choices. * indicates that the language component was trained from scratch. "HF ID" is the identifier for the model weights on Huggingface.

```
            {"type": "text", "text": "cyan cylinder and red cylinder have equal
                mass. How will the speed of cyan cylinder change after colliding
                with red cylinder? A: Speed does not change, B: cyan cylinder's
                speed decreases, C: cyan cylinder's speed will increase, D: Not
                enough data"}]
    },
    {
    "role": "assistant",
    "content": [{"type": "text", "text": "B"}]
    }
    {
    "role": "user",
    "content": [{"type": "video", "path": "path/to/evaluation_image.png"},
                {"type": "text", "text": "yellow cube and purple sphere have equal mass.
                    How will the speed of yellow cube change after colliding with purple
                    sphere? A: Not enough data, B: Speed does not change, C: yellow
                    cube's speed will decrease, D: yellow cube's speed will increase"}]
    }
]
```

The exact style of the dictionary differs with the LMM. All prompts include a system prompt. When exemplars are provided, the system prompt is "Understand the underlying physics from the following videos and choose an option among A, B, C and D to answer the question. Do not provide reasoning." When exemplars are absent, the system prompt is "Choose an option among A, B, C and D to answer the question based on your understanding of physical laws. Do not provide reasoning."

However, these instructions are insufficient to restrict the model's output to options. Since LLMs are first pre-trained for next-token prediction, they tend to complete the prompt instead of answering the question in it. Thus, the generated output tends to be descriptive, especially when there are no exemplars. For instance, suppose the model wants to choose option "D: Both objects move". Instead of simply outputting "D", the model may output "... Therefore, both objects may move." Even instruction-tuned models sometimes fail to format their outputs, despite including the instruction "Do not provide reasoning" in the system prompt. Parsing the chosen option from such descriptive outputs is difficult.

Therefore, to avoid descriptive outputs, we provide demonstration samples in the expected output format since LLMs can understand output formatting from exemplars (Min et al., 2022). Even when exemplars do not contain question-answer pairs, we include randomly chosen options among {"A", "B", "C", "D"} to condition the model to output only the option index. An example of such a conversation for zero-shot evaluation is provided below:

```
[
    {
    "role": "system",
    "content": [{"type": "text",
```

```
            "text": "Choose an option among A, B, C and D to answer the question
                based on your understanding of physical laws. Do not provide
                reasoning."}]
    },
    {
    "role": "assistant",
    "content": [{"type": "text", "text": "A"}]
    }
    {
    "role": "assistant",
    "content": [{"type": "text", "text": "D"}]
    }
    {
    "role": "assistant",
    "content": [{"type": "text", "text": "C"}]
    }
    {
    "role": "user",
    "content": [{"type": "video", "path": "path/to/evaluation_image.png"},
                {"type": "text", "text": "yellow cube and purple sphere have equal mass.
                    How will the speed of yellow cube change after colliding with purple
                    sphere? A: Not enough data, B: Speed does not change, C: yellow
                    cube's speed will decrease, D: yellow cube's speed will increase"}]
    }
]
```

Here, options in the assistant dictionaries do not have any relation to the video in the exemplar. Therefore, we still refer to this approach as "zero-shot evaluation" in the sense that no useful samples are provided as exemplars to the model.

### B.3 PARSING THE CHOSEN OPTION FROM GENERATED OUTPUT

We ask the model to generate a maximum of 10 new tokens. The tokens generated by the model are decoded using its corresponding tokenizer. To obtain the chosen option from this decoded output, we "clean" the decoded string first until the first character is the option index. We remove the following substrings in our "cleaning" procedure:

1. Placeholders for image and video tokens. E.g., `<|im_start|>`, `<fim_suffix>`.

2. Partial placeholders for image and video tokens. E.g., only `<|im_st` from `<|im_start|>`.

3. Strings such as "The correct answer is" prepending the chosen option. Such strings are first collected manually and then removed during the cleaning procedure.

4. Strings that follow the chosen option. E.g., "Human: Which movie...". Similar to the previous case, these strings can be collected for each model and then removed during the experimental evaluation.

If multiple options are chosen, then the model's output is marked to be incorrect. In some cases where not enough exemplars are available for the model, we allow the model to generate more tokens and attempt to find the chosen option based on textual overlap between the generated output and the given options.

### B.4 ABSOLUTE FEW-SHOT ACCURACY OF LMMS IN IRREGULAR SCENARIOS

Tabs. 4 and 5 show the absolute accuracy values for the experiments in §§ 4.4 and 4.5.

### B.5 OTHER DETAILS

**Compute**: Almost all experiments were run on individual A6000 GPUs on a server with 128 AMD EPYC 7502 (32-core) processors. Very few experiments were run on H200 and A100 GPUs.

| LMM | LMC | Wall | AMC | Red-LMC | Red-Pass | SB | CC |
|---|---|---|---|---|---|---|---|
| InternVL3-1B (Zhu et al., 2025) | 48.24 (-23.22) | 18.49 (+ 9.20) | 83.22 (+11.76) | 58.45 (-13.01) | 55.99 (-15.47) | 10.45 (-76.97) | 47.94 (-23.52) |
| VideoLLaMA3-2B (Zhang et al., 2025) | 37.14 (-25.83) | 22.36 (-15.33) | 59.35 (- 3.62) | 35.69 (-27.28) | 49.47 (-13.49) | 8.09 (-55.84) | 46.03 (-16.93) |
| InternVL3-2B (Zhu et al., 2025) | 15.08 (-74.92) | 0.25 (-91.36) | 99.05 (+ 9.05) | 25.06 (-64.94) | 21.10 (-68.90) | 0.10 (-76.81) | 9.35 (-80.65) |
| Gemma 3-4B (Kamath et al., 2025) | 86.38 (+33.92) | 56.13 (-21.56) | 59.70 (+ 7.24) | 80.65 (+28.19) | 62.11 (+ 9.64) | 24.62 (-44.88) | 37.09 (-15.38) |
| LLaVA-NeXT-Vid (Zhang et al., 2024) | 16.68 (-39.50) | 35.28 (+27.69) | 45.58 (-10.60) | 7.62 (-48.56) | 11.78 (-44.40) | 18.44 (-16.62) | 31.61 (-24.57) |
| InternVL3-8B (Zhu et al., 2025) | 94.32 (- 5.33) | 98.54 (- 0.20) | 100.00 (+ 0.35) | 71.78 (-27.87) | 47.27 (-52.38) | 45.98 (-53.87) | 68.94 (-30.70) |
| LLaVA-OneVision (Li et al., 2024a) | 98.19 (- 0.90) | 97.19 (- 2.56) | 99.40 (+ 0.30) | 74.94 (-24.16) | 67.72 (-31.38) | 31.41 (-67.88) | 65.78 (-33.32) |
| VideoLLaMA3-7B (Zhang et al., 2025) | 83.62 (-14.42) | 81.41 (+29.65) | 97.74 (- 0.30) | 49.07 (-48.97) | 54.04 (-44.01) | 64.17 (-29.28) | 76.28 (-21.76) |
| LLaVA-NeXT-IL (Li et al., 2024c) | 93.72 (- 3.42) | 94.17 (+15.33) | 92.16 (- 4.97) | 62.51 (-34.63) | 59.85 (-37.29) | 43.72 (-21.48) | 98.09 (+ 0.95) |
| Qwen2-VL (Wang et al., 2024a) | 96.03 (- 2.11) | 100.00 (+ 0.75) | 95.33 (- 2.81) | 99.15 (+ 1.01) | 96.69 (- 1.45) | 13.77 (-84.96) | 90.35 (- 7.79) |
| Qwen2.5-Omni (Xu et al., 2025) | 63.82 (+ 5.83) | 39.30 (- 4.97) | 90.10 (+32.11) | 53.63 (- 4.36) | 63.61 (+ 5.62) | 56.23 (-38.29) | 44.27 (-13.72) |
| Gemma 3-12B (Kamath et al., 2025) | 57.29 (-28.04) | 92.36 (+23.52) | 91.51 (+ 6.18) | 27.67 (-57.66) | 21.40 (-63.92) | 18.09 (-68.62) | 87.14 (+ 1.81) |
| Aria (Li et al., 2024b) | 27.24 (-25.88) | 0.70 (-65.18) | 80.35 (+27.24) | 28.57 (-24.54) | 19.95 (-33.17) | 6.93 (-85.35) | 7.49 (-45.63) |

Table 4: 3-shot evaluation results of LMMs on irregular scenarios. The exemplars contain **both** videos and QA pairs.

| LMM | LMC | Wall | AMC | Red-LMC | Red-Pass | SB | CC |
|---|---|---|---|---|---|---|---|
| InternVL3-1B (Zhu et al., 2025) | 23.17 (-25.08) | 7.74 (-10.75) | 31.81 (-51.41) | 26.12 (-32.33) | 28.57 (-27.42) | 16.98 (+ 6.53) | 49.50 (+ 1.56) |
| VideoLLaMA3-2B (Zhang et al., 2025) | 13.17 (-23.97) | 13.27 (- 9.10) | 33.22 (-26.13) | 14.19 (-21.50) | 18.20 (-31.28) | 26.13 (+18.04) | 13.17 (-32.86) |
| InternVL3-2B (Zhu et al., 2025) | 5.73 (- 9.35) | 0.00 (- 0.25) | 62.71 (-36.33) | 8.32 (-16.74) | 8.72 (-12.38) | 7.94 (+ 7.84) | 0.15 (- 9.20) |
| LLaVA-NeXT-Vid (Zhang et al., 2024) | 15.68 (- 1.01) | 42.46 (+ 7.19) | 43.22 (- 2.36) | 16.84 (+ 9.22) | 16.14 (+ 4.36) | 29.25 (+10.80) | 41.46 (+ 9.85) |
| InternVL3-8B (Zhu et al., 2025) | 18.19 (-76.13) | 57.44 (-41.11) | 58.34 (-41.66) | 16.69 (-55.09) | 20.95 (-26.32) | 14.12 (-31.86) | 9.80 (-59.15) |
| LLaVA-OneVision (Li et al., 2024a) | 3.22 (-94.97) | 25.63 (-71.56) | 56.73 (-42.66) | 2.81 (-72.13) | 3.21 (-64.51) | 13.77 (-17.64) | 0.05 (-65.73) |
| VideoLLaMA3-7B (Zhang et al., 2025) | 4.12 (-79.50) | 20.10 (-61.31) | 45.08 (-52.66) | 9.02 (-40.05) | 10.43 (-43.61) | 29.90 (-34.27) | 9.05 (-67.24) |
| LLaVA-NeXT-IL (Li et al., 2024c) | 0.40 (-93.32) | 16.53 (-77.64) | 49.40 (-42.76) | 0.40 (-62.11) | 0.40 (-59.45) | 20.00 (-23.72) | 32.31 (-65.78) |
| Qwen2-VL (Wang et al., 2024a) | 3.82 (-92.21) | 45.68 (-54.32) | 66.83 (-28.49) | 4.61 (-94.54) | 3.21 (-93.48) | 7.74 (- 6.03) | 45.53 (-44.82) |
| Qwen2.5-Omni (Xu et al., 2025) | 0.15 (-63.67) | 22.01 (-17.29) | 59.15 (-30.95) | 1.80 (-51.83) | 2.26 (-61.35) | 18.39 (-37.84) | 41.51 (- 2.76) |
| Gemma 3-12B (Kamath et al., 2025) | 10.35 (-46.93) | 0.45 (-91.91) | 67.14 (-24.37) | 5.16 (-22.51) | 15.04 (- 6.37) | 5.03 (-13.07) | 0.00 (-87.14) |
| Aria (Li et al., 2024b) | 0.90 (-26.33) | 3.42 (+ 2.71) | 29.35 (-51.01) | 1.30 (-27.27) | 1.30 (-18.65) | 13.02 (+ 6.08) | 0.20 (- 7.29) |

Table 5: 3-shot evaluation results of LMMs in irregular scenarios. The exemplars contain **only** videos.

**Modifications to Huggingface**: Some of the evaluated LMMs did not account for multiple videos in the prompt. Even the latest version of the Transformers library[6] had this bug. So we made minor changes to the codebase of LLaVA-NeXT-Video, LLaVA-OneVision, and InternVL3 models. The modified "transformers" library is included in the codebase.

# C    OTHER RELATED WORKS

In this section, we include some recent works that evaluated various reasoning aspects of LLMs and LMMs. We will also clarify that our objective has never been explored in any of these works.

**Intuitive Physics Understanding**: Physically impossible scenarios have been employed to evaluate physical reasoning in learned models, following the "violation of expectation" principle from cognitive theory (Margoni et al., 2024). Here, the key hypothesis is that a model with excellent physical reasoning can also understand when the underlying physical laws in the given scenario violate the known physical laws. However, violation of expectation in infants and children is often the initial step towards adaptation to a new physical environment (Kotovsky & Baillargeon, 1998; Denison & Xu, 2010; Faßbender et al., 2025). For instance, (Faßbender et al., 2025) showed that infants and children adapted their force while opening and closing drawers, whose friction was temporally altered. Violation of expectation in learned models in the context of physical laws can be used as a proxy for physical reasoning. An early example of such work is the IntPhys (Riochet et al., 2021) benchmark that quantified the physical reasoning abilities of models trained on visual datasets that obeyed universal physical laws using their next-frame prediction errors on physically impossible scenarios. More works on intuitive physics understanding have emerged since (Epstein et al., 2020; Weihs et al., 2022; Jassim et al., 2024; Garrido et al., 2025).

**Intuitive physics understanding is not inductive physical reasoning**: Intuitive physics understanding differs from INPHYRE in the final objective, as the underlying assumption in intuitive physics understanding does not impact our setting. INPHYRE evaluates how well a large multimodal model can infer the underlying physical laws from the demonstration samples and apply them for physical reasoning when given a scenario the model has not seen during its training. This property, which we refer to as inductive physical reasoning in the main paper, is the key question we pose. The absolute physical reasoning ability (that we refer to as parametric knowledge) of this model on regular physical

---

[6] https://github.com/huggingface/transformers

tasks that the model might have seen during training is not of interest to us. However, since we do not know which scenarios were observed during training and which were not, we rely on impossible scenarios to evaluate the inductive physical reasoning.

**Evaluation of physical reasoning in learned models**: Research interests in learning visual physical reasoning predate the era of large models. Early works generated synthetic vision datasets that depicted physical events such as collisions and falls, and trained models to predict future events (Lerer et al., 2016; Baradel et al., 2020; Bear et al., 2021), answer questions about cue events Mottaghi et al. (2016), or interact with the physical simulator to achieve an end goal (Bakhtin et al., 2019). Other tasks involved visually inferring latent physical properties, such as mass and friction, from the physical interactions (Chen et al., 2022; Tung et al., 2023) and causal physical reasoning (Yi et al., 2020; Ates et al., 2022).

**Reasoning from demonstration samples**: Prior efforts have attempted to reason *how* LLMs utilize demonstration samples. These works consider both parametric knowledge (Min et al., 2022; Li et al., 2024d; Nafar et al., 2025) and inductive reasoning hypotheses (Garg et al., 2022; Bai et al., 2023b; Wang et al., 2024b; Vacareanu et al., 2024; Nafar et al., 2025). However, their findings are usually limited to synthetic regression tasks on LLMs (Garg et al., 2022; Bai et al., 2023b; Wang et al., 2024b), and they do not consider physical reasoning tasks on LMMs.

**Glitch detection using LMMs**: Another task similar to intuitive physics understanding is "glitch detection." Here, the intuition is that a model that understands the underlying physics can also detect glitches in a given scenario. Some examples of glitch detection using LLMs are (Taesiri et al., 2022b;a; 2024; Cao et al., 2024).

**Use of synthetic data for physical reasoning**: Synthetically generated images and videos are commonly used for physical reasoning, as collecting visual data on real physical events is both taxing and time-consuming. Several of the works that we listed above and in § 2 also use synthetic data. In Tab. 6, we list some additional works that use synthetically generated collision events for physical reasoning. We also include CLEVR (Johnson et al., 2017) dataset in the table due to its visual resemblance with INPHYRE. Note: The tasks in ComPhy vary in terms of the underlying law required for reasoning (e.g., objects with the same charge repelling after a collision). All the events are still collision events.

| Benchmark | # of Tasks | Physics Engine | Renderer | Events other than collision events |
|---|---|---|---|---|
| CLEVR (Johnson et al., 2017) | - | No physics, only images | Blender | No events, only images |
| CLEVRER (Yi et al., 2020) | 4 question types, 5 description types | Bullet | Blender | None |
| ComPhy (Chen et al., 2022) | 4 (mass, charge, color, collision) | Bullet | Blender | None |
| CoPhy (Baradel et al., 2020) | 3 (BlocktowerCF, BallsCF, CollisionCF) | PyBullet | PyBullet | Stability of stacked objects |
| **INPHYRE** (ours) | 10 (spanning linear and angular momentum conservation, object permanence) | PyBullet | Blender | None |

Table 6: A tabular comparison of INPHYRE with other works that use synthetically generated collision events for physical reasoning.

**Different w.r.t. ContPhy and PhysBench**: ContPhy (Zheng et al., 2024) and PhysBench (Chow et al., 2025) are among the most comprehensive physical reasoning benchmarks that appeared recently. They include both real and synthetic videos that show physical events governed by a wide span of physical laws, such as Newton's laws of motion, friction, fluid mechanics, etc. Therefore, they successfully evaluate the knowledge of LMMs on diverse topics required for physical reasoning. The key difference between INPHYRE and these works is the objective. INPHYRE evaluates the ability of the models to adapt to an unseen scenario, while these works evaluate the parametric knowledge about physics in these models. As a consequence, ContPhy and PhysBench arrive at conclusions different from ours. Zheng et al. (2024) find that these models "struggle to perform well on our benchmark, highlighting their limited physical commonsense for the continuum, especially soft bodies, and fluids." Similarly, Chow et al. (2025) "identified significant gaps in physical world understanding, particularly in open-source models, due to inadequate training data" and postulated that these models "struggle with understanding the physical world – likely due to the absence of physical knowledge in their training data and the lack of embedded physical priors." In contrast, we find that LMMs struggle to

adapt to an unseen scenario. It is also not clear if more training data can improve inductive physical reasoning from demonstration samples.

## D  VARIOUS TYPES OF REASONING

In this section, we will distinguish between general visual reasoning, inductive reasoning, and inductive physical reasoning (our work).

**General visual reasoning** refers to the broad set of tasks that involve answering questions from visual signals (one or more images and/or videos). To address these tasks, the model must extract information from the visual signal and apply auxiliary information that the model already has about the content of the visual signal. This information is generally factual. For example, the input image may contain a knife and a fruit, and the auxiliary information corresponding to this content is that knives are sharp and can be used to cut fruits (Aroca-Ouellette et al., 2021; Bisk et al., 2020; Wang et al., 2023; Dong et al., 2024). The skills required for general visual reasoning include localization, understanding, and information retrieval.

Visual reasoning is different from physical reasoning, although they share the input modalities and skill set partially. In physical reasoning, the task is to apply the physical knowledge possessed by the model. Unlike factual information, physical knowledge is a framework that is actionable only when applied to a specific context. For example, in the previous example of a knife next to a fruit, a relevant physical knowledge is that an object remains at rest unless acted upon by an external force (Newton's first law of motion). To use this physical knowledge, the model must not only localize and understand the objects in the scene, but also realize that Newton's first law of motion applies to the objects. In contrast, if one of the objects in the scene were a fluid, the model must realize that the laws of fluid dynamics also apply to that object. In summary, physical reasoning involves an additional application of mathematical frameworks over visual reasoning and is, therefore, more challenging.

**Inductive reasoning** is the ability of an agent to infer the underlying rules from a few samples and then apply these rules in a new evaluation scenario. Since the samples may not fully inform the agent about every underlying rule, inductive reasoning involves a degree of uncertainty. Existing works that evaluate inductive reasoning in LLMs follow this premise. As an example of inductive reasoning, consider the following sequence: A000, B001, C010, D011, E100. Which is the next element in this sequence? A possible (not necessarily unique) underlying rule of the sequence that we can infer from the premise is that the alphabets follow their canonical alphabetical order, while the remaining numbers encode the position of the alphabets in binary format. According to this rule, the next two elements in the sequence are F101 and G110.

**Is evaluating inductive physical reasoning similar to evaluating inductive reasoning?** Unlike our work on inductive physical reasoning, evaluating inductive reasoning does not contradict any existing knowledge in the models. For instance, it is unlikely that an evaluated model has any knowledge regarding the above sequence example of inductive reasoning. It is also possible that the model has never seen any sequence like that during training. In contrast, we are evaluating the ability of the model to adapt any existing physical knowledge that it might have to the evaluation scenario. Thus, inductive physical reasoning is not only inferring the underlying physics from demonstration samples but also doing so when the inferred physics potentially contradicts the parametric knowledge of the model.

**Evaluating inductive reasoning in LLMs**: There exists a long line of works that evaluate the abstract reasoning abilities of LLMs. Most of the other prior efforts to evaluate inductive reasoning are restricted to reasoning from textual inputs about abstract tasks (Mirchandani et al., 2023; Gendron et al., 2024; Wang et al., 2024c; Cheng et al., 2024; He et al., 2024; Bowen et al., 2024; Yan et al., 2025; Li et al., 2025). One notable work is Abstract and Reasoning Corpus (ARC) (Chollet, 2019), containing abstract tests similar to traditional IQ tests to evaluate "general artificial intelligence" in large models. It provides a training set that allows the candidate (human or machine learning agent) to understand the reasoning required to solve the tasks. Unlike our work, these tasks are largely symbolic and used to evaluate LLMs.

# E  ADDITIONAL EXPERIMENTAL EVALUATION

## E.1  EFFECT OF WEIGHT QUANTIZATION ON INDUCTIVE PHYSICAL REASONING

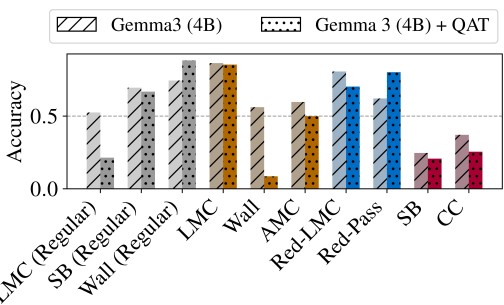

Figure 18: Effect of weight quantization on inductive physical reasoning

LLMs and LMMs are billion-parameter models with expensive inference. Therefore, weight quantization is used to reduce their resource needs. An important concern with weight quantization is the drop in performance due to lower precision. In this subsection, we examine whether weight quantization adversely affects inductive physical reasoning in LMMs. We compare the prediction accuracy of Gemma 3 (4B) with its quantized version, obtained by fine-tuning using Quantization Aware Training (QAT) (Jacob et al., 2018). The models are evaluated in both regular and irregular scenarios with exemplars including both videos and question-answer pairs.

Fig. 18 compares the accuracy of the quantized model against its non-quantized counterpart. Out of the ten evaluated scenarios, the quantized model performs worse than the full-precision model on all but two scenarios. In LMC, they perform comparably, while in Wall and LMC (regular), the quantized model shows a stark drop in accuracy. Notably, the quantized model outperforms the full precision model in Wall (regular) and Red-Pass. The results indicate that quantization indeed affects inductive physical reasoning, but its impact varies with the physical reasoning task.

## E.2  EVALUATION OF PROPRIETARY MODELS

| LMM | LMC | Wall | AMC | Red-LMC | Red-Pass | SB | CC |
|---|---|---|---|---|---|---|---|
| GPT 4.1 Mini | 5.08 (-69.05) | 31.61 (-61.26) | 68.89 (- 5.23) | 6.97 (-67.15) | 6.57 (-67.55) | 90.25 (+31.40) | 52.36 (-21.76) |
| GPT 4.1 Nano | 37.54 (+ 1.16) | 32.91 (-39.55) | 53.77 (+17.39) | 43.71 (+ 7.33) | 30.28 (- 6.11) | 29.05 (-21.94) | 18.34 (-18.04) |
| Gemini 2.5 Flash | 16.93 (-20.55) | 34.22 (-12.91) | 59.75 (+22.26) | 16.99 (-20.49) | 16.19 (-21.30) | 32.61 (-57.04) | 68.24 (+30.75) |
| Gemini 2.5 Flash Lite | 48.49 (- 3.57) | 54.82 (-28.39) | 82.66 (+30.60) | 30.13 (-21.93) | 24.76 (-27.30) | 50.75 (-34.69) | 22.31 (-29.75) |
| Gemini 2 Flash | 70.75 (-21.51) | 98.19 (- 0.50) | 94.72 (+ 2.46) | 15.59 (-76.67) | 10.68 (-81.58) | 87.14 (- 9.36) | 98.34 (+ 6.08) |

Table 7: Performance of some mainstream models in irregular scenarios. The numbers in parentheses show the difference between their performances in irregular scenarios and those in the corresponding regular scenarios.

In our main experiments in § 4, we evaluated open-source LMMs since they allowed more flexibility in prompting and output parsing. Proprietary LMMs generally only allow you to provide prompts and frames through their API and do not allow flexible output parsing. However, they are also more sophisticated and generally more accurate. In this section, we evaluate the inductive physical reasoning in these mainstream models. Specifically, we consider models from GPT (Achiam et al., 2023) and Gemini (Comanici et al., 2025) families. Due to the length limits of the prompts (since the API encodes images as string), we passed only 2 demonstration samples to the API. The results are shown in Tab. 7. The numbers show the accuracy in irregular scenarios, and the numbers in the parentheses show their difference with the accuracy in the corresponding regular scenario. We observe that mainstream models perform similarly to open-source models in inductive physical reasoning. The evaluated mainstream models fail in all scenarios except AMC.

### E.3 CAN CHAIN-OF-THOUGHT PROMPTING IMPROVE INDUCTIVE PHYSICAL REASONING?

The experiments in § 4 show that LMMs struggle to infer the underlying physical laws from demonstration samples and apply them for physical reasoning. These experiments also revealed the underlying language bias of these models. In this section, we evaluate if chain-of-thought (CoT) prompting (Wei et al., 2022) can help with inductive physical reasoning. Intuitively, CoT can alleviate the burden of inferring the underlying physical laws from video frames and multiple-choice question-answer pairs by enunciating the physical laws required for reasoning in the prompt.

| LMM | LMC (Reg.) | SB (Reg.) | Wall (Reg.) | LMC | Wall | AMC | Red-LMC | Red-Pass | SB | CC |
|---|---|---|---|---|---|---|---|---|---|---|
| InternVL3-1B | 73.32 (+40.05) | 97.82 (+50.28) | 91.21 (+81.91) | 90.20 (+41.96) | 70.65 (+52.16) | 97.69 (+14.47) | 33.23 (-25.21) | 69.42 (+13.43) | 9.90 (- 0.55) | 36.48 (-11.46) |
| VideoLLaMA3-2B | 98.89 (+35.93) | 98.88 (+34.96) | 97.84 (+60.15) | 99.65 (+62.51) | 69.55 (+47.19) | 98.49 (+39.15) | 3.16 (-32.53) | 34.24 (-15.24) | 27.34 (+19.25) | 89.15 (+43.12) |
| InternVL3-2B | 99.90 (+ 9.90) | 99.80 (+22.88) | 99.90 (+ 8.29) | 99.95 (+84.87) | 45.63 (+45.38) | 95.38 (- 3.67) | 1.55 (-23.51) | 12.08 (- 9.02) | 36.28 (+36.18) | 83.17 (+73.82) |
| Gemma 3-4B | 95.88 (+43.42) | 98.53 (+29.02) | 99.30 (+24.87) | 99.30 (+12.91) | 99.60 (+43.47) | 96.98 (+37.29) | 17.04 (-63.61) | 62.76 (+ 0.65) | 35.03 (+10.40) | 72.21 (+35.13) |
| LLaVA-NeXT-Vid | 68.29 (+12.11) | 61.95 (+26.89) | 82.86 (+75.28) | 75.43 (+58.74) | 89.60 (+54.32) | 69.45 (+23.87) | 10.98 (+ 3.36) | 33.43 (+21.65) | 29.30 (+10.85) | 62.36 (+30.75) |
| InternVL3-8B | 100.00 (+ 0.35) | 100.00 (+ 0.15) | 100.00 (+ 1.26) | 100.00 (+ 5.68) | 100.00 (+ 1.46) | 100.00 (0) | 2.66 (-69.12) | 29.67 (-17.59) | 49.05 (+ 3.07) | 100.00 (+31.06) |
| LLaVA-OneVision | 99.95 (+ 0.85) | 99.85 (+ 0.56) | 99.45 (- 0.30) | 98.89 (+ 0.70) | 99.75 (+ 2.56) | 98.29 (- 1.11) | 21.55 (-53.38) | 48.52 (-19.20) | 12.76 (-18.64) | 99.45 (+33.67) |
| VideoLLaMA3-7B | 99.80 (+ 1.76) | 99.70 (+ 6.24) | 94.72 (+42.96) | 99.35 (+15.73) | 99.45 (+18.04) | 98.34 (+ 0.60) | 2.46 (-46.62) | 21.50 (-32.53) | 52.41 (-11.76) | 100.00 (+23.72) |
| LLaVA-NeXT-IL | 100.00 (+ 2.86) | 94.47 (+30.75) | 95.68 (+16.83) | 99.65 (+ 5.93) | 96.48 (+ 2.31) | 97.64 (+ 5.48) | 8.77 (-53.73) | 61.20 (+ 1.35) | 52.86 (+ 9.15) | 99.65 (+ 1.56) |
| Qwen2-VL | 100.00 (+ 1.86) | 100.00 (+ 1.27) | 99.90 (+ 0.65) | 100.00 (+ 3.97) | 99.95 (- 0.05) | 96.18 (+ 0.83) | 4.56 (-94.59) | 30.88 (-65.81) | 28.29 (+14.52) | 99.95 (+ 9.60) |
| Qwen2.5-Omni | 96.38 (+50.55) | 99.49 (+12.84) | 93.07 (+48.79) | 99.05 (+35.23) | 99.55 (+60.25) | 99.50 (+ 9.40) | 4.36 (-49.27) | 31.38 (-32.23) | 58.24 (+ 2.01) | 93.67 (+49.40) |
| Gemma 3-12B | 82.46 (- 2.86) | 95.84 (+19.33) | 73.17 (+12.41) | 87.59 (+30.30) | 94.42 (+ 2.06) | 97.59 (+ 6.08) | 29.47 (+ 1.80) | 68.32 (+46.92) | 11.71 (- 6.38) | 98.99 (+11.86) |

Table 8: Performance of various LMMs in both regular and irregular scenarios when evaluated using chain-of-thought (CoT) prompting, where the underlying physical law is explicitly included along with the answers in demonstration samples.

Tab. 8 shows the reasoning accuracies for various LMMs in both regular and irregular scenarios. The red and green numbers in the parentheses show the decrease and increase in the accuracy against the corresponding accuracy without CoT. We note a significant improvement in the performance across almost all model-scenario combinations, except Red-Pass and Red-LMC. CoT prompting fails to help multiple models in Red-Pass and Red-LMC scenarios, even worsening the performance of some models. In Red-Pass and Red-LMC, all except red colored objects follow the true physical laws. That is, unlike the remaining scenarios, Red-Pass and Red-LMC require the LMM to apply conditional reasoning depending on the color of the object. Although the demonstration samples include collisions with and without red colored objects, it appears that LMMs are unable to infer the conditional nature of the underlying reasoning.

### E.4 EFFECT OF FINE-TUNING ON INDUCTIVE PHYSICAL REASONING

In this section, we explore the performance improvement on INPHYRE that we can obtain through fine-tuning. Fine-tuning through direct supervision and reinforcement learning has been shown to improve visual reasoning across diverse tasks (Zhai et al., 2024; Tan et al., 2025; Cai et al., 2025). Note that we cannot evaluate inductive physical reasoning by fine-tuning LMMs on INPHYRE samples, as it becomes unclear if the LMM's output is due to the fine-tuning or due to the inductive physical reasoning. We conduct this experiment as a proxy way to obtain an "upper bound" on what an LMM can achieve on INPHYRE. Since fine-tuning is an expensive process, we limit our experiment to the smallest LMM in the evaluated cohort, InternVL3-1B. We use supervised fine-tuning without any low-rank adaptation techniques such as LoRA (Hu et al., 2022). Since no single hyperparameter combination worked consistently well for all scenarios, presumably due to the diversity between the tasks, we report the results for all hyperparameter combinations from a grid search.

| Epochs | LR | LMC (Reg.) | SB (Reg.) | Wall (Reg.) | LMC | Wall | AMC | Red-LMC | Red-Pass | SB | CC |
|---|---|---|---|---|---|---|---|---|---|---|---|
| 20 | $2 \times 10^{-5}$ | 12.01 (-21.26) | 42.97 (- 4.57) | 18.29 (+ 8.99) | 56.38 (+ 8.14) | 23.72 (+ 5.23) | 62.56 (-20.65) | 68.57 (+10.13) | 48.27 (- 7.72) | 25.38 (+14.92) | 58.44 (+10.50) |
| 50 | $2 \times 10^{-5}$ | 32.36 (- 0.90) | 35.16 (-12.38) | 31.16 (+21.86) | 54.27 (+ 6.03) | 46.68 (+28.19) | 66.48 (-16.73) | 29.32 (-29.12) | 35.79 (-20.20) | 28.94 (+18.49) | 70.40 (+22.46) |
| 100 | $2 \times 10^{-5}$ | 38.09 (+ 4.82) | 54.19 (+ 6.65) | 42.21 (+32.91) | 75.03 (+26.78) | 93.47 (+74.97) | 89.25 (+ 6.03) | 42.76 (-15.69) | 33.28 (-22.71) | 56.48 (+46.03) | 83.97 (+36.03) |
| 120 | $2 \times 10^{-5}$ | 83.12 (+49.85) | 92.85 (+45.31) | 58.69 (+49.40) | 56.83 (+ 8.59) | 68.34 (+49.85) | 75.63 (- 7.59) | 54.94 (- 3.51) | 50.08 (- 5.91) | 38.94 (+28.49) | 90.10 (+42.16) |
| 20 | $2 \times 10^{-4}$ | 37.64 (+ 4.37) | 25.72 (-21.82) | 24.37 (+15.08) | 0.70 (-47.54) | 0.00 (-18.49) | 7.09 (-76.13) | 23.91 (-34.54) | 28.02 (-27.97) | 0.00 (-10.45) | 28.44 (-19.50) |
| 50 | $2 \times 10^{-4}$ | 73.57 (+40.30) | 95.08 (+47.54) | 27.99 (+18.69) | 43.22 (- 5.03) | 30.40 (+11.91) | 81.86 (- 1.36) | 27.42 (-31.03) | 28.92 (-27.07) | 23.92 (+13.47) | 70.60 (+22.66) |
| 100 | $2 \times 10^{-4}$ | 94.07 (+60.80) | 30.70 (-16.84) | 39.30 (+30.00) | 27.84 (-20.40) | 31.11 (+12.61) | 38.84 (-44.37) | 31.68 (-26.77) | 34.49 (-21.50) | 35.58 (+25.13) | 42.11 (- 5.83) |
| 120 | $2 \times 10^{-4}$ | 42.46 (+ 9.20) | 71.08 (+23.54) | 87.04 (+77.74) | 30.80 (-17.44) | 69.55 (+51.06) | 59.90 (-23.32) | 42.26 (-16.19) | 24.91 (-31.08) | 32.31 (+21.86) | 35.43 (-12.51) |
| 20 | $1 \times 10^{-3}$ | 0.00 (-33.27) | 0.00 (-47.54) | 14.42 (+ 5.13) | 0.00 (-48.24) | 0.00 (-18.49) | 0.00 (-83.22) | 0.00 (-58.45) | 0.00 (-55.99) | 0.00 (-10.45) | 0.00 (-47.94) |
| 50 | $1 \times 10^{-3}$ | 0.00 (-33.27) | 1.78 (-45.76) | 0.00 (- 9.30) | 0.00 (-48.24) | 0.00 (-18.49) | 0.00 (-83.22) | 0.00 (-58.45) | 25.56 (-30.43) | 0.00 (-10.45) | 0.00 (-47.94) |
| 100 | $1 \times 10^{-3}$ | 18.39 (-14.87) | 24.15 (-23.39) | 19.70 (+10.40) | 29.10 (-19.15) | 11.51 (- 6.98) | 26.38 (-56.83) | 6.37 (-52.08) | 27.42 (-28.57) | 15.78 (+ 5.33) | 23.12 (-24.82) |
| 120 | $1 \times 10^{-3}$ | 10.00 (-23.27) | 8.73 (-38.81) | 28.19 (+18.89) | 29.95 (-18.29) | 20.95 (+ 2.46) | 23.27 (-59.95) | 26.42 (-32.03) | 0.05 (-55.94) | 24.82 (+14.37) | 21.36 (-26.58) |
| **Best** |  | 94.07 (+60.80) | 95.08 (+47.54) | 87.04 (+77.74) | 75.03 (+26.78) | 93.47 (+74.97) | 89.25 (+ 6.03) | 68.57 (+10.13) | 50.08 (- 5.91) | 56.48 (+46.03) | 90.10 (+42.16) |

Table 9: Results of fine-tuned InternVL3-1B on regular and irregular scenarios for different hyperparameter combinations. The best results for each scenario are given in the last row.

Tab. 9 shows the accuracy of fine-tuned InternVL3-1B on both regular and irregular scenarios for different hyperparameter combinations. We observe that fine-tuning improves the performance in all scenarios with at least one hyperparameter combination, except for the Red-Pass scenario. These observations are similar to those from our experiments on CoT prompting. As we mentioned in the previous section, Red-Pass and Red-LMC require conditional reasoning, and it seems that fine-tuning cannot improve conditional physical reasoning in LMMs. The absolute accuracies are comparatively small for Red-LMC and SB.

## E.5 Do the Results from CoT Prompting and Fine-Tuning Experiments Invalidate Inductive Physical Reasoning?

In the previous sections, we evaluated the effects of CoT prompting and fine-tuning on the prediction accuracy for INPHYRE benchmark. Although both CoT prompting and fine-tuning improved the performance of multiple LMMs on most scenarios (except Red-Pass and Red-LMC), we emphasize that neither of these solutions is viable when the inductive physical reasoning ability is put to the test in practice. Specifically, both these techniques require prior access to the underlying physical laws or the inference samples themselves.

In CoT prompting, we included the underlying physical law in the impossible scenario as part of the prompt. This would not be possible in unseen scenarios where we do not possess any information about the physical laws involved. In contrast, throughout this paper, we had instead made a milder assumption that we only had access to prior visual samples and the right response in those samples. The more arduous and crucial task of inferring the underlying physical law was left to the model itself. Similarly, the models are trained on these samples in the fine-tuning experiments. Once these samples become part of the training data, it becomes nearly impossible to know if the model predictions are based on their inductive physical reasoning or because they remember their fine-tuning samples. Nonetheless, we included these experiments for the sake of completion, as they provide a surrogate for an upper bound on the performance achievable in INPHYRE.

## E.6 Qualitative Inspection of Generated Outputs

In § 4.2, we evaluated the parametric knowledge of LMMs in regular scenarios. We used the accuracy with which the model chose the correct option as our evaluation metric. Although this metric is useful for quantifying physical reasoning, it does not inform us how the model arrived at the option. This lack of clarity is outside the scope of a quantitative benchmark like INPHYRE. Nonetheless, we look at some of the open-ended generation output of these LMMs in regular scenarios to obtain insights into the underlying physical reasoning in these models. For this experiment, we provide the LMMs with only the input image and the question. Answer options are **not** included in the prompt so that the model will resort to open-ended generation to answer the question. We do not conduct this experiment on irregular scenarios as (1) video-only sub-setting did not show any competent performance, and (2) video-text sub-setting cannot be conducted without providing the options in exemplars, which would lead to the model predicting just the option alone.

The following subsubsections show the open-ended generation outputs for LMC (regular), SB (regular), and Wall (regular). Horizontal lines are used to demarcate the outputs of each LMM. In each output, incorrect physical reasoning is highlighted in red, correct physical reasoning relevant to the task is highlighted in green, and partially true physical reasoning in yellow. If any information from the prompt is ignored or if new assumptions are hallucinated, we highlight them in blue. After each generation output, we describe our observations. We do not list observations for the models that do not provide sufficient information about their reasoning. Note that the outputs of the same LMM could vary between individual samples. Therefore, the outputs of an LMM for this specific evaluation sample may not be similar to the model's outputs on other evaluation samples from the same scenario.

### E.6.1 Open-ended generation output for LMC (regular)

**Frames**: 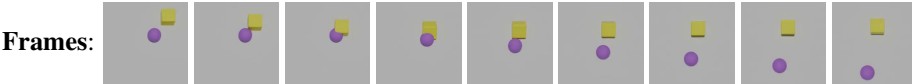

**System prompt**: Answer the question based on your understanding of physical laws.

**Question**: yellow cube and purple sphere have equal mass. How will the speed of yellow cube change after colliding with purple sphere?

**InternVL3-1B** (Zhu et al., 2025): If the yellow cube and the purple sphere have equal mass, the collision will result in both objects moving in opposite directions. Since the collision is elastic, the speed of the yellow cube will be halved. This is because the kinetic energy is conserved in an elastic collision.

**Observations**: InternVL3-1B hallucinated that the collision was elastic, instead of stating it as an assumption. Moreover, the speed will not be halved even if the collision is elastic.

**VideoLLaMA3-2B** (Zhang et al., 2025): After the collision, the yellow cube will move at a slower speed than before.

**InternVL3-2B** (Zhu et al., 2025): When the yellow cube collides with the purple sphere, the speed of the yellow cube will decrease. This is because the collision will transfer momentum from the yellow cube to the purple sphere, causing the yellow cube to slow down. The exact change in speed depends on the masses and the nature of the collision (elastic or inelastic). In an elastic collision, both momentum and kinetic energy are conserved, while in an inelastic collision, kinetic energy is not conserved, and some energy is converted into other forms, such as heat or deformation.

**Observations**: The reasoning is correct and derived from accurate physical laws.

**Gemma3-4B** (Kamath et al., 2025): Okay, let's analyze this collision based on physics principles.

**Here's the breakdown:**

1. **Conservation of Momentum:** In a closed system (like this one, assuming no external forces), the total momentum *before* a collision equals the total momentum *after* the collision. Momentum is calculated as mass (m) times velocity (v): p = mv.

2. **Newton's Third Law (Action-Reaction):** For every action, there is an equal and opposite reaction. This means that when the sphere hits the cube, the cube exerts an equal and opposite force back on the sphere.

3. **Collision Dynamics:** Because the cube and sphere have equal mass, and the collision is likely to be fairly brief, we can assume it's an *elastic* collision (or close to it – perfectly elastic collisions are rare in reality). In an elastic collision, kinetic energy is also conserved.

**Now, let's apply this:**

* **Before the collision:** Both the cube and sphere are at rest. Total momentum = 0. * **During the collision:** The sphere transfers its momentum to the cube. Since the masses are equal, the *change* in momentum for each object will be the same, but the *velocity* will change differently. * **After the collision:** The cube will move in one direction, and the sphere will move in the opposite direction. Because the masses are equal, the *speeds* of the cube and sphere will be equal, but in opposite directions.

**Therefore, the speed of the yellow cube will change, and it will move with a speed equal in magnitude, but opposite in direction, to the speed the purple sphere had before the collision.**

**Important Note:** This assumes a *perfect* elastic collision. In a real-world scenario, some kinetic energy would be lost as heat or sound due to friction between the objects, so the speeds after the collision wouldn't be *exactly* equal. However, the principle of conservation of momentum and Newton's Third Law still apply.

Would you like me to explain any part of this in more detail or consider a slightly different collision scenario?

**Observations**: Gemma3-4B incorrectly assumed that objects would be at rest before the collision. It also contradicts itself later and states that the purple sphere was in motion before the collision.

**LLaVA-NeXT-Video** (Zhang et al., 2024): The speed of the yellow cube will not change after colliding with the purple sphere because they have equal mass. The conservation of momentum is

a fundamental principle in physics that states that the total momentum of a closed system remains constant. When two objects collide, the total momentum of the system is the sum of the momenta of the individual objects. If the objects have equal mass, then the momentum of the system is the same before and after the collision.

**Observations**: The stated law of momentum conservation is accurate but is applied incorrectly. Additionally, momentum conservation is a universal law and does not depend on the masses of the individual objects.

**InternVL3-8B** (Zhu et al., 2025): To determine how the speed of the yellow cube will change after colliding with the purple sphere, we need to consider the principles of conservation of momentum and kinetic energy, assuming an elastic collision.

1. **Conservation of Momentum**: In an elastic collision, the total momentum before the collision is equal to the total momentum after the collision. If the masses of the yellow cube and the purple sphere are equal, and they are initially at rest, the momentum before the collision is zero. After the collision, the momentum of the cube and the sphere will be equal in magnitude but opposite in direction.

2. **Conservation of Kinetic Energy**: In an elastic collision, the total kinetic energy before the collision is equal to the total kinetic energy after the collision. Since the masses are equal and initially at rest, the kinetic energy before the collision is zero. After the collision, the kinetic energy will be distributed between the cube and the sphere.

Given these principles, let's analyze the situation:

- **Initial State**: Both the yellow cube and the purple sphere are at rest. Therefore, the initial momentum and kinetic energy are zero.

- **Collision**: When the yellow cube collides with the purple sphere, they will exchange velocities due to the conservation of momentum and kinetic energy. Since the masses are equal, the velocities will be equal in magnitude but opposite in direction.

- **Final State**: After the collision, the yellow cube will move in the opposite direction of the purple sphere with the same speed it had before the collision, and the purple sphere will move in the opposite direction of the yellow cube with the same speed it had before the collision.

Therefore, if the yellow cube and the purple sphere have equal mass and the collision is perfectly elastic, the speed of the yellow cube will remain the same after the collision, but its direction will be reversed.

**Observations**: InternVL3-8B states the physical laws correctly. However, it incorrectly believes that the objects are at rest before collision. This hallucinated assumption led it to provide the wrong reasoning.

**LLaVA-Onevision** (Li et al., 2024a): The speed of the yellow cube will decrease after colliding with the purple sphere. This is because the cube will transfer some of its kinetic energy to the sphere, causing it to slow down. The exact change in speed will depend on the masses and velocities of the cube and sphere before the collision, as well as the angle of the collision.

**Observations**: The stated law of kinetic energy transfer is accurate. The model seems to have ignored the information about the mass of the objects stated in the prompt.

**VideoLLaMA3-7B** (Zhang et al., 2025): decrease

**LLaVA-NeXT-Interleave** (Li et al., 2024c): The speed of the yellow cube will change after colliding with the purple sphere. The exact change depends on the nature of the collision, which is not specified in the image. If the collision is elastic and the yellow cube and purple sphere have equal mass, the speed of the yellow cube will change to a value that depends on the angle of impact and the initial speeds of the two objects.

**Observations**: The reasoning is accurate, although the reasoning principle is not explicitly stated. It additionally states that the exact change in velocity will depend on factors that are unavailable from the prompt.

**Qwen2-VL** (Wang et al., 2024a): The speed of the yellow cube will decrease after colliding with the purple sphere.

**Qwen2.5-Omni** (Xu et al., 2025): The speed of the yellow cube will decrease after colliding with the purple sphere. This is because, according to the law of conservation of momentum, the total momentum of a closed system remains constant. When the yellow cube collides with the purple sphere, they exert equal and opposite forces on each other. Since they have equal mass, the yellow cube will experience a force that will slow it down, while the purple sphere will experience a force that will accelerate it.

**Observations**: Qwen2-5-Omni reasoned that the yellow cube will slow down due to the objects having equal mass. In reality, the mass of the objects do not decide whether the colliding object slows down or not, but only how much it slows down.

**Gemma3-12B** (Kamath et al., 2025): Here's an analysis of what would happen, based on the laws of physics, assuming an elastic collision:

**Understanding the Principles**

* **Conservation of Momentum:** In a closed system (meaning no external forces like friction), the total momentum before a collision is equal to the total momentum after the collision. Momentum is mass x velocity. * **Elastic Collision:** This is a collision where kinetic energy is conserved. In reality, collisions are rarely perfectly elastic (some energy is typically lost as heat or sound), but we'll assume it for simplicity.

**Analysis of the Scenario**

1. **Initial State:** The cube is moving, and the sphere is stationary. Let's say the cube's velocity is *v*. The sphere's velocity is 0. 2. **Collision:** During the collision, the cube transfers some of its momentum to the sphere. 3. **Final State (Ideal Elastic Collision):** Since the masses are equal, the collision will result in the cube stopping and the sphere moving off with a velocity equal to the initial velocity of the cube (*v*).

**Therefore, the speed of the yellow cube will change and it will come to a complete stop.**

**Important Considerations:**

* **Perfect Elasticity:** The assumption of a perfectly elastic collision is rarely true. In a real-world scenario, the cube would likely not stop completely. Some of the cube's kinetic energy would be converted to other forms (sound, heat due to deformation). * **Direction:** This analysis only considers speed (magnitude of velocity). The direction of motion after the collision would also change.

**Observations**: Gemma3-12B states the physical principles accurately and provides the correct reasoning using these principles.

**Aria** (Li et al., 2024b): In a collision between two objects of equal mass, assuming an elastic collision, the speed of each object will change such that the yellow cube will come to a stop, and the purple sphere will move forward at the initial speed of the yellow cube. This is due to the principle of conservation of momentum and kinetic energy. The yellow cube transfers all its momentum and kinetic energy to the purple sphere.

**Observations**: The laws stated by Aria are correct, and the laws are applied accurately during physical reasoning.

E.6.2 OPEN-ENDED GENERATION OUTPUT FOR SB (REGULAR)

**Frames**: 

**System prompt**: Answer the question based on your understanding of physical laws.

**Question**: green cylinder has more mass than blue cube. What occurs when green cylinder and blue cube collide?

**InternVL3-1B** (Zhu et al., 2025): When the green cylinder and the blue cube collide, the following physical law applies:

The collision between the two objects will result in a change in their velocities. Specifically, the blue cube will move forward with a higher velocity than the green cylinder, and the green cylinder will move backward with a lower velocity. This is because the blue cube has more mass and thus a greater inertia, causing it to move faster than the green cylinder.

The direction of the velocities can be determined using the principle of conservation of momentum. The change in momentum of the blue cube will be equal to the change in momentum of the green cylinder. Mathematically, this can be expressed as:

$$\Delta p_{\text{blue}} = \Delta p_{\text{green}}$$

Where: - $\Delta p_{\text{blue}}$ is the change in momentum of the blue cube. - $\Delta p_{\text{green}}$ is the change in momentum of the green cylinder.

Since the blue cube has more mass, it will have a greater change in momentum, leading to a higher velocity for the blue cube. Conversely, the green cylinder will have a lower change in momentum and thus a lower velocity.

Thus, the blue cube will move forward with a higher velocity than the green cylinder.

**Observations**: InternVL3-1B confused the unequal mass relation between the objects from the prompt. The prompt said that the green cylinder has more mass than the blue cube, but the model inferred that the blue cube had more mass than the green cylinder. It additionally reasoned that an object with more mass will undergo a greater change in momentum. However, these mistakes canceled each other and accidentally led the model to the right answer.

**VideoLLaMA3-2B** (Zhang et al., 2025): green cylinder and blue cube collide

**InternVL3-2B** (Zhu et al., 2025): When the green cylinder and the blue cube collide, the green cylinder will likely deform or compress due to the force exerted by the blue cube. The blue cube will exert a force on the green cylinder, causing it to move or change shape. The exact outcome will depend on the masses, velocities, and other physical properties of the objects involved.

**Observations**: InternVL3-2B assumed about the material properties of the objects and reasoned that the green cylinder could potentially deform. It also incorrectly reasoned that the outcome would depend on the mass and the velocities. Since information was provided about the mass, the correct answer is that both objects would move in the same direction.

**Gemma3-4B** (Kamath et al., 2025): Here's what will likely happen when the green cylinder and blue cube collide, based on our understanding of physics:

**1. Impulse and Momentum Conservation:**

* **Impulse:** When the objects collide, they exert forces on each other for a brief period. This exchange of forces is called impulse. * **Momentum:** The total momentum of the system (cylinder + cube) before the collision must equal the total momentum after the collision. Momentum is mass times velocity (p = mv).

**2. The Outcome – The Cylinder Will Move Less**

* Since the green cylinder has more mass than the blue cube, it will have more momentum before the collision. * During the collision, the cylinder will decelerate (slow down) more quickly than the cube. * Because of its greater initial momentum, it will transfer more momentum to the cube. * As a result, the blue cube will experience a larger change in velocity, resulting in a greater deflection from its initial direction, while the green cylinder will experience a smaller change in velocity and will move less significantly.

**In simpler terms:** The heavier object (cylinder) will have a harder time changing its motion during the collision compared to the lighter object (cube).

**Important Note:** This assumes an *elastic* collision (no energy loss due to heat, sound, or deformation). If the collision was inelastic (e.g., the objects deform significantly and some energy is lost), the outcome would be slightly different.

If you would like, you can provide another scenario and I can analyze it using the same principles.

**Observations**: Gemma3-4B stated several physical laws accurately but applied them incorrectly. It also incorrectly stated that an object with a larger initial momentum will transfer more momentum.

**LLaVA-NeXT-Video** (Zhang et al., 2024): When a green cylinder and a blue cube collide, the collision will result in a change in their motion and possibly a change in their position relative to each other. The exact outcome depends on the relative speed and direction of the cylinder and the cube at the time of collision. If the cylinder is moving faster or in the same direction as the cube, the cylinder will push the cube to one side, and if the cylinder is moving in the opposite direction, the cube will push the cylinder to the other side. If the cylinder is moving at a right angle to the cube, the cylinder will change direction and the cube will change direction. The collision will also result in a transfer of momentum from the cylinder to the cube or vice versa, depending on the direction of the collision.

If the cylinder is moving faster than the cube, the cylinder will push the cube to one side, and the cube will move in the opposite direction. If the cylinder is moving slower, the cube will push the cylinder to the other side. If the cylinder is moving at a right angle to the cube, the cylinder will change direction and the cube will change direction.

**Observations**: Although the stated reasoning is true, the model does not use the information about the mass of the objects provided in the prompt. The reasoning is incomplete due to the model ignoring useful information from the prompt.

**InternVL3-8B** (Zhu et al., 2025): When the green cylinder and the blue cube collide, several factors determine the outcome, primarily based on the principles of physics, specifically Newton's laws of motion and the concept of momentum.

1. **Mass and Momentum**: The green cylinder has more mass than the blue cube. Momentum is the product of mass and velocity ($p = mv$). If both objects are moving with the same velocity before the collision, the green cylinder will have more momentum due to its greater mass.

2. **Elastic vs. Inelastic Collision**: The nature of the collision (whether it is elastic or inelastic) also plays a role. In an elastic collision, both momentum and kinetic energy are conserved. In an inelastic collision, momentum is conserved, but kinetic energy is not. The objects may stick together or deform upon impact.

3. **Resultant Motion**: - **Elastic Collision**: If the collision is perfectly elastic, the green cylinder will exert a greater force on the blue cube, potentially causing the blue cube to rebound with some velocity. The green cylinder might slow down slightly due to the transfer of momentum. - **Inelastic Collision**: If the collision is inelastic, the objects may stick together after the collision. The combined mass of the green cylinder and the blue cube will move with a velocity determined by the conservation of momentum.

4. **Energy Considerations**: In an elastic collision, the kinetic energy before and after the collision remains the same. In an inelastic collision, some kinetic energy is converted into other forms of energy, such as heat or deformation energy.

In summary, when the green cylinder and the blue cube collide, the green cylinder, having more mass, will exert a greater force on the blue cube. Depending on whether the collision is elastic or inelastic, the blue cube may rebound or stick to the green cylinder, and the green cylinder may slow down. The exact outcome depends on the initial velocities and the nature of the collision.

**Observations**: InternVL3-8B states the principles in physics correctly and applies them accurately.

**LLaVA-Onevision** (Li et al., 2024a): The green cylinder will likely push the blue cube away from it due to its greater mass and momentum. The blue cube will also experience a force in the opposite direction of the green cylinder's motion. The exact outcome of the collision will depend on the speed and angle of the green cylinder's impact with the blue cube.

**Observations**: LLaVA-Onevision states and applies the physical laws correctly.

**VideoLLaMA3-7B** (Zhang et al., 2025): green cylinder will not move

**LLaVA-NeXT-Interleave** (Li et al., 2024c): When the green cylinder and blue cube collide, the blue cube will likely move and change its position as a result of the interaction. The green cylinder may also change its position or velocity depending on the nature of the collision.

**Observations**: LLaVA-NeXT-Interleave reasons that the green cylinder *may* change its position or velocity, while, in reality, it would always change its position due to its larger mass, irrespective of the nature of the collision.

**Qwen2-VL** (Wang et al., 2024a): The green cylinder will push the blue cube.

**Qwen2.5-Omni** (Xu et al., 2025): When the green cylinder collides with the blue cube, the cylinder will exert a force on the cube, causing the cube to accelerate and move in the direction of the force. The cylinder will also experience a reaction force, which will cause it to slow down slightly. The exact outcome of the collision depends on the masses and velocities of the objects, as well as the nature of the collision (e.g., elastic or inelastic).

**Observation**: Qwen2.5-Omni correctly states and applies the relevant physical laws.

**Gemma3-12B** (Kamath et al., 2025): Here's what would likely happen based on physical laws, considering the image and the premise that the green cylinder has more mass than the blue cube:

**The Collision**

When the green cylinder and the blue cube collide, the following will generally occur:

1. **Momentum Transfer:** Momentum is a measure of mass in motion (mass x velocity). Since the green cylinder has more mass, it will have more momentum. During the collision, some of the green cylinder's momentum will be transferred to the blue cube.

2. **Green Cylinder's Slowdown:** The green cylinder, having lost some momentum, will slow down after the collision.

3. **Blue Cube's Speedup:** The blue cube, gaining momentum, will speed up after the collision. It will move in roughly the direction the green cylinder was traveling.

4. **Elasticity Matters:** The specifics will depend on how *elastic* the collision is: * **Perfectly Elastic Collision:** (Rare in the real world). Kinetic energy is conserved. The cylinder would slow down significantly, the cube would speed up significantly, and the total kinetic energy would remain the same. * **Inelastic Collision:** (More common). Some kinetic energy is lost as heat or sound during the impact. The cylinder will slow down more, and the cube will speed up less than in a perfectly elastic collision.

**In summary:** The green cylinder will slow down, and the blue cube will speed up. The greater the mass difference, the more pronounced these effects will be.

**Important Note:** This explanation assumes a simplified scenario. Factors like friction, the exact shapes of the objects, and how they bounce off each other would influence the outcome in a real-world situation.

**Observations**: The reasoning is accurate, although the laws were not explicitly stated.

**Aria** (Li et al., 2024b): When the green cylinder and blue cube collide, the green cylinder, having more mass, will likely exert a greater force on the blue cube due to its greater momentum. This could cause the blue cube to move or change direction, depending on the angle and speed of the collision. The outcome will also depend on factors such as the material properties and the speed of the objects. If the collision is elastic, both objects might bounce off each other; if inelastic, some energy may be lost as heat or sound.

**Observations**: Aria restates the information about the mass of the objects from the prompt, but reasons that the objects may bounce off each other, which cannot happen when one object has a higher mass than the other.

E.6.3  OPEN-ENDED GENERATION OUTPUT FOR WALL (REGULAR)

**Frames**: 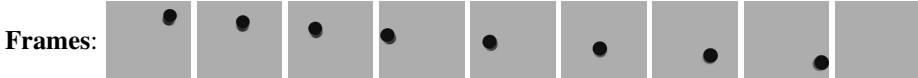

**System prompt**: Answer the question based on your understanding of physical laws.

**Question**: black cylinder collides elastically with the wall. What is the outcome of black cylinder colliding with wall?

**InternVL3-1B** (Zhu et al., 2025): When a black cylinder collides elastically with the wall, the outcome depends on the specific nature of the collision. Here are a few possible scenarios:

1. **Inelastic Collision:** - If the collision is inelastic, meaning the two objects stick together after the collision, the black cylinder will move with some velocity after the collision. The wall will exert a force on the moving cylinder, causing it to deform or change direction.

2. **Elastic Collision:** - If the collision is elastic, meaning the two objects bounce off each other, the black cylinder will move with some velocity after the collision. The wall will exert a force on the moving cylinder, causing it to change direction and possibly deform.

3. **Perfectly Inelastic Collision:** - If the collision is perfectly inelastic, meaning the two objects stick together, the black cylinder will move with some velocity after the collision. The wall will exert a force on the moving cylinder, causing it to change direction and possibly deform.

Without additional information about the specific nature of the collision, it is not possible to determine the exact outcome. However, in general, the outcome will depend on whether the collision is inelastic or elastic.

**Observations**: InternVL3-1B ignores the information in the prompt that the collision was elastic. It also generates a contradictory statement that the objects will move with some velocity after sticking to each other if the collision is inelastic.

**VideoLLaMA3-2B** (Zhang et al., 2025): black cylinder bounces

**InternVL3-2B** (Zhu et al., 2025): When a black cylinder collides elastically with a wall, the outcome is that the cylinder will bounce back with the same speed and direction it initially had. This is because the collision is perfectly elastic, meaning there is no loss of kinetic energy. The wall exerts an equal and opposite force on the cylinder, causing it to reverse its motion.

**Observation**: InternVL3-2B states and applies physical laws accurately.

**Gemma3-4B** (Kamath et al., 2025): Here's an analysis of what happens when a black cylinder collides elastically with a wall, based on the principles of physics:

**Understanding Elastic Collisions**

* **Elastic Collision:** In an elastic collision, both momentum and kinetic energy are conserved. This means the total momentum of the system before the collision equals the total momentum after the collision, and the total kinetic energy before equals the total kinetic energy after. * **Conservation

of Momentum:** The law of conservation of momentum states that the total momentum of a closed system remains constant. * **Conservation of Kinetic Energy:** The law of conservation of kinetic energy states that the total kinetic energy of a system remains constant.

**What Happens in the Collision**

1. **Momentum Transfer:** As the cylinder hits the wall, momentum is transferred from the cylinder to the wall. The cylinder initially has a certain velocity (let's call it *v*), and the wall is initially at rest. After the collision, the cylinder will likely have a velocity in the opposite direction (let's call it *-v*), and the wall will have a velocity (*-v*). 2. **Angle of Reflection:** The cylinder won't bounce straight back. It will bounce at an angle. The angle of reflection depends on the relative velocities of the cylinder and the wall *before* the impact. A more detailed calculation would require knowing the initial velocity and angle, and the coefficient of restitution (which describes the "bounciness" of the collision). 3. **Kinetic Energy is Preserved:** Since it is an elastic collision, the kinetic energy of the system will remain the same, meaning no energy will be lost as heat or sound.

**Outcome**

The black cylinder will bounce off the wall at an angle, and the wall will move in the opposite direction with the same speed as the cylinder.

**Important Note:** Without knowing the initial velocity and angle of the cylinder, it's impossible to predict the exact angle of reflection.

Let me know if you'd like a more detailed explanation or want to explore scenarios with specific initial conditions!

**Observations**: Although Gemma3-4B states and uses the appropriate physical laws, it treats the wall as an object that can move, leading to a wrong prediction.

**LLaVA-NeXT-Video** (Zhang et al., 2024): The image you've provided is a simple illustration of a black cylinder colliding with a wall, but it doesn't provide enough information to determine the outcome of the collision. The outcome would depend on the properties of the cylinder and the wall, such as their mass, size, material, and the speed at which they collide.

If the cylinder is a solid object and the collision is elastic, it would bounce off the wall without any permanent deformation or damage to either object. The cylinder would lose some of its kinetic energy during the collision, and the wall would also experience a deformation, but it would return to its original state after the collision.

If the cylinder is a liquid or a gas, the collision would be more complex, and the outcome would depend on the properties of the liquid or gas, such as its viscosity and the pressure. The cylinder would likely spread out upon impact, and the wall would experience a deformation, but the cylinder would not bounce back.

If the cylinder is a solid and the collision is inelastic, the cylinder would break or deform upon impact, and the wall would also experience a deformation.

Without more information, it's impossible to provide a specific outcome.

**Observations**: As one of the possibilities, LLaVA-NeXT-Video assumes the collision is elastic and correctly predicts that the colliding object would bounce off the wall. However, it then adds wrong physical reasoning that the object would lose its kinetic energy and the wall would be deformed, neither of which is possible during an elastic collision. It also hallucinates the extremely unlikely situation that the object could be a liquid or a gas as another possibility.

**InternVL3-8B** (Zhu et al., 2025): When a black cylinder collides elastically with a wall, the outcome can be described using the principles of elastic collisions and conservation of momentum and kinetic energy.

1. **Elastic Collision**: In an elastic collision, both momentum and kinetic energy are conserved. This means that the total momentum before the collision is equal to the total momentum after the collision, and the total kinetic energy before the collision is equal to the total kinetic energy after the collision.

2. **Collision with a Wall**: When the black cylinder collides with the wall, the wall is considered to be stationary and much more massive than the cylinder. This simplifies the analysis because the wall does not move significantly.

3. **Reversal of Velocity**: Since the wall is stationary and much more massive, the black cylinder will reverse its direction of motion upon collision. The velocity of the cylinder will change direction but not magnitude, assuming no other forces act on it (like friction or air resistance).

4. **Conservation of Momentum**: The momentum of the cylinder before the collision (p_initial = m * v_initial) will be equal to the momentum after the collision (p_final = m * v_final). Since the wall is stationary, the momentum of the wall remains zero. Therefore, the momentum of the cylinder before the collision is equal to the momentum of the cylinder after the collision but in the opposite direction.

5. **Conservation of Kinetic Energy**: The kinetic energy of the cylinder before the collision (KE_initial = 0.5 * m * v_initial$\hat{2}$) will be equal to the kinetic energy after the collision (KE_final = 0.5 * m * v_final$\hat{2}$). Since the magnitude of the velocity remains the same, the kinetic energy is conserved.

In summary, the outcome of the black cylinder colliding elastically with the wall is that the cylinder will rebound with the same speed but in the opposite direction. The wall remains stationary, and the collision conserves both momentum and kinetic energy.

**Observation**: InternVL3-8B states and applies the true physical laws to obtain the accurate output.

**LLaVA-Onevision** (Li et al., 2024a): The black cylinder will bounce back and move away from the wall.

**VideoLLaMA3-7B** (Zhang et al., 2025): The black cylinder will bounce back with the same speed it had before colliding with the wall.

**LLaVA-NeXT-Interleave** (Li et al., 2024c): The black cylinder collides elastically with the wall, which means that the kinetic energy is converted into potential energy, and the cylinder rebounds in the opposite direction. The final velocity of the cylinder after the collision is zero, as it is moving at the same speed as the wall before the collision.

**Observations**: LLaVA-NeXT-Interleave incorrectly states that kinetic energy is converted to potential energy during an elastic collision. Then it generates contradictory statements that the cylinder would rebound while having zero final velocity. The model also hallucinates that the wall was at the same speed as the cylinder before the collision.

**Qwen2-VL** (Wang et al., 2024a): The black cylinder will bounce off the wall.

**Qwen2.5-Omni** (Xu et al., 2025): When a black cylinder collides elastically with a wall, the outcome depends on the initial velocity and direction of the cylinder. In an elastic collision, both momentum and kinetic energy are conserved. If the cylinder is moving towards the wall, it will bounce back with the same speed but in the opposite direction. If the cylinder is moving away from the wall, it will continue moving in the same direction with the same speed. The exact outcome can be calculated using the conservation of momentum and kinetic energy equations.

**Observations**: Qwen2.5-Omni claims that the outcome would depend on the initial velocity of the cylinder and its direction. While the exact outcome would depend on the initial velocity, the general outcome that the cylinder would bounce back is independent of the velocity when the collision is elastic. After making the correct prediction with the appropriate reasoning, it also considers the scenario where the object was initially moving away from the wall, without realizing that the collision would not have occurred in that scenario.

**Gemma3-12B** (Kamath et al., 2025): Here's an analysis of what happens when the black cylinder collides elastically with the wall, based on physical laws:

**Understanding Elastic Collisions**

* **Elastic Collision:** In an elastic collision, both momentum and kinetic energy are conserved. This means: * The cylinder will bounce off the wall. * No energy is lost during the collision (no heat, sound, or deformation). * The wall is assumed to be rigid, meaning it doesn't move or deform.

**Outcome of the Collision**

* **Reversal of Velocity:** The key outcome is that the cylinder will reverse its direction of motion. If it was traveling towards the wall, it will now travel away from the wall. * **Speed is the same:** Because kinetic energy is conserved, the cylinder's speed immediately *after* the collision will be the same as its speed *before* the collision. * **Angle:** The angle at which it bounces off will depend on the angle of incidence (the angle at which it hit the wall). The angle of incidence equals the angle of reflection.

**Therefore, the black cylinder will bounce back away from the wall with the same speed it had before the collision, and with a trajectory determined by the laws of reflection.**

**Important Note:** True, perfectly elastic collisions are rare in the real world. However, for a solid, rigid object like a cylinder colliding with a solid wall, we often *approximate* the collision as elastic for simplicity in analysis.

**Observations**: Gemma3-12B generates correct physical laws and uses them for accurate reasoning.

**Aria** (Li et al., 2024b): In an elastic collision, the black cylinder would bounce off the wall with the same speed but in the opposite direction. This means if the cylinder was moving towards the wall, it would move away from the wall at the same speed after the collision. The direction of motion would be reversed, but the magnitude of the velocity would remain unchanged.

**Observation**: Aria provides the correct reasoning, although it does not state the underlying physical laws.

**Conclusion**: Most models can state the underlying physical laws, but often hallucinate assumptions such as the wall moving or the objects deforming (despite the collision described as "elastic" in the prompt). In some cases, such as Gemma3-4B and InternVL3-1B, the generated output has contradicting statements about the outcome of the collision.

**Overall Conclusion about Parametric Knowledge**: Most evaluated LMMs can state the principles of momentum and energy conservation accurately. However, they struggle to apply these laws to reasoning. In some cases, their reasoning contradicts the earlier parts of their output. Sometimes, they also hallucinate new assumptions (*e.g.*, about the object material) that stray their output from the correct reasoning. We conjecture that the LMMs know these physical laws as they know factual information, and not as mathematical models that must be applied flexibly.

### E.7 EXPLAINING THE APPARENT INDUCTIVE PHYSICAL REASONING IN AMC

In § 4.4, we observed that LMMs performed better in AMC compared to LMC (Reg.) when exemplars containing videos and question-answer pairs were provided. This is surprising since LMC (Reg.) followed universal physical laws and AMC did not. An LMM that can correctly infer custom physical laws from exemplars must also be able to do so when the scenarios follow the true physical laws. To explain this peculiar behavior of LMMs, we will manually inspect the open-ended generation outputs of LMMs on AMC. We follow the procedure from § E.6: the evaluation sample consists of the video and the question without any answer options. We also add "Will any object rotate?" at the prompt's end to elicit an output about the angular momentum conservation from the LMMs. We follow the highlighting style in the previous section. The outputs of various LMMs are given below.

#### E.7.1 OPEN-ENDED GENERATION OUTPUT FOR AMC (REGULAR)

**Frames**: 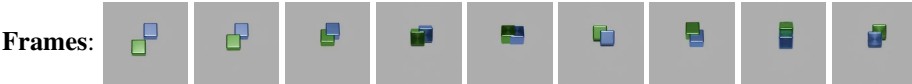

**System prompt**: Answer the question based on your understanding of physical laws.

**Question**: blue cube and green cube have equal mass. What is the outcome of green cube colliding with blue cube? Will any object rotate?

**InternVL3-1B** (Zhu et al., 2025): When two cubes collide, the outcome depends on the nature of the collision. In this case, the blue cube and the green cube have equal mass.

1. **Collision Outcome**: - If the collision is elastic, the two cubes will stick together after the collision. This means they will form a single larger cube. - If the collision is inelastic, the two cubes will separate and move apart.

2. **Rotation of Objects**: - In an elastic collision, the two cubes will stick together, so they will not rotate. - In an inelastic collision, the two cubes will separate, and they will not rotate.

Given that the collision is elastic, the outcome is that the two cubes will stick together. Therefore, the outcome of the collision is that the two cubes will stick together.

**Observations**: InternVL3-1B states illogical physical laws that objects stick together after an elastic collision and that objects stay separate after an inelastic collision. In reality, elastic collision ensures that the objects stay apart, while inelastic collision may result in objects sticking together. Interestingly, the output of the model is independent of this assumption as the model believes that neither elastic nor inelastic collision will result in the objects rotating.

**VideoLLaMA3-2B** (Zhang et al., 2025): When the green cube collides with the blue cube, both objects will rotate around their respective axes of symmetry. The collision is perfectly elastic and frictionless, meaning no energy or momentum is lost during the interaction.

**Observation**: VideoLLaMA3-2B incorrectly reasons that objects will rotate about the axes of symmetry due to the collision, but it does not provide any reasoning.

**InternVL3-2B** (Zhu et al., 2025): When the green cube collides with the blue cube, the outcome depends on the nature of the collision. If the collision is perfectly elastic, both cubes will continue to move after the collision, and no rotation will occur. If the collision is perfectly inelastic, the two cubes will stick together, and one of them will rotate while the other moves.

Since the problem states that the green cube and the blue cube have equal mass, we can infer that the collision is likely to be perfectly elastic. In this case, both cubes will continue to move after the collision, and no rotation will occur.

**Observations**: InternVL3-2B reasons correctly that no rotation will happen if the collision is elastic. However, it reasons that one of the objects may rotate while the other moves in the case of an inelastic collision. This is not true. Angular momentum is not preserved in the model's reasoning.

**Gemma3-4B** (Kamath et al., 2025): Okay, let's analyze this collision based on physical principles.

**Here's a breakdown of what will likely happen:**

1. **Conservation of Momentum:** The most important principle here is the conservation of momentum. In a closed system (assuming no external forces like friction or air resistance), the total momentum *before* the collision equals the total momentum *after* the collision. Momentum is calculated as mass x velocity.

2. **Equal Masses:** We are told that both cubes have the same mass. This simplifies the calculation considerably.

3. **Collision Dynamics:** Because both cubes have equal mass, when the green cube hits the blue cube, they will exert equal and opposite forces on each other. This means the green cube will recoil backward and the blue cube will move forward.

4. **Rotation:** Here's the key: Because the cubes are colliding with equal and opposite forces and they have the same mass, the system will conserve angular momentum. This means the overall rotation of the two cubes will be *zero*. While *each individual cube might rotate slightly* during the impact (due to the unevenness of the collision surface), the *combined* rotation of the pair will cancel out.

**Therefore, the outcome is:**

* **The green cube will recoil backward, and the blue cube will move forward.** * **Neither cube will rotate noticeably.** The rotation will be minimal and quickly damped out.

**Important Note:** This analysis assumes an *elastic* collision, where kinetic energy is conserved. In reality, some energy will be lost as heat and sound due to the inelastic deformation of the cubes during the impact. However, the fundamental principle of angular momentum conservation still holds.

Would you like me to delve deeper into a specific aspect of this, such as calculating the resulting velocities (if we knew the initial velocities)?

**Observations**: Gemma3-4B states the principle of angular momentum conservation and uses it correctly to predict that no rotation will happen.

**LLaVA-NeXT-Video** (Zhang et al., 2024): The image you've provided shows two cubes, one blue and one green, with equal mass. If they were to collide, the outcome would depend on the direction of the collision and the relative orientation of the cubes. If they were to collide head-on, the collision would be perfectly elastic, and no rotation would occur. If they were to collide at an angle, the cubes could rotate due to the conservation of momentum and the transfer of momentum.

In an elastic collision, the total momentum of the system is conserved, so if the blue cube is moving faster than the green cube, it would transfer its momentum to the green cube, causing it to move in the opposite direction. If the green cube is moving faster, it would transfer its momentum to the blue cube, causing it to move in the same direction. If they are moving at the same speed, they would pass each other without any change in momentum.

If the cubes were to collide at an angle, the collision would be inelastic, and the cubes would deform and possibly change shape. The cube that is struck would absorb the energy of the collision, and the other would rebound. The cube that absorbed the energy would rotate due to the conservation of angular momentum. The cube that rebound would have a change in its orientation, and the cube that struck it would have a change in its position.

Without more information about the specifics of the collision, it's impossible to predict the exact outcome.

**Observations**: LLaVA-NeXT-Video correctly states that no rotation will happen during an elastic collision. However, it additionally outputs that rotation is possible due to the conservation of momentum. Later, the model states that an object that absorbs energy could rotate due to the conservation of angular momentum. Here, the model states the right principle that objects could rotate to preserve angular momentum. But its output is wrong as the principle does not affect the rotation in this particular example.

**InternVL3-8B** (Zhu et al., 2025): When two objects with equal mass collide, the outcome depends on the nature of the collision (elastic or inelastic) and the initial velocities of the objects.

In an elastic collision, both momentum and kinetic energy are conserved. If the green cube and the blue cube have equal mass and they collide head-on with equal and opposite velocities, they will exchange velocities after the collision. This means the green cube will stop and the blue cube will continue moving with the velocity of the green cube before the collision.

In an inelastic collision, some kinetic energy is converted into other forms of energy (like heat or deformation), and the objects may stick together after the collision. If the green cube and the blue cube stick together after the collision, they will move as a single combined mass with a velocity determined by the conservation of momentum.

In either case, if the collision is perfectly head-on and the objects are rigid, they will likely rotate if they are not perfectly smooth or if there is friction between them. The rotation will depend on the distribution of mass and the point of contact during the collision.

So, yes, both objects could rotate depending on the specifics of the collision and the properties of the cubes.

**Observation**: The model outputs the true outcome of the collision first. To answer our specific question about the rotation, it outputs that the objects could rotate if there is friction between them. However, this is not possible if the collision is head-on as the model states.

**LLaVA-Onevision** (Li et al., 2024a): The outcome of the green cube colliding with the blue cube will depend on the angle and speed of the collision. If the collision is head-on and at a high speed, the cubes may not rotate at all. However, if the collision is at an angle, the cubes may rotate around their point of contact. The exact outcome would depend on the specific details of the collision, such as the mass and velocity of the cubes, as well as the surface they are colliding on.

**Observation**: This is correct reasoning since the model is only stating, under proper assumptions, the possibility of objects rotating around their point of contact.

**VideoLLaMA3-7B** (Zhang et al., 2025): yes

**LLaVA-NeXT-Interleave** (Li et al., 2024c): The outcome of the green cube colliding with the blue cube would depend on the initial velocity and angle of impact. If the green cube is moving towards the blue cube, it could potentially cause the blue cube to rotate. However, without knowing the initial conditions, it's not possible to predict the exact outcome.

**Observation**: LLaVA-NeXT-Interleave is wrong in stating that rotation can happen if one object moves towards (and collides) with another object.

**Qwen2-VL** (Wang et al., 2024a): Yes, the green cube will collide with the blue cube. When the green cube collides with the blue cube, the cubes will rotate around their center of mass. The exact amount of rotation will depend on the initial velocities and masses of the cubes.

**Observations**: Qwen2-VL provides the wrong output that the objects would rotate around their center of mass. It also supports this wrong reasoning with factually incorrect physical laws.

**Qwen2.5-Omni** (Xu et al., 2025): When the green cube collides with the blue cube, both cubes will experience a force that causes them to move in opposite directions. Since they have equal mass, the force exerted on each cube will be equal in magnitude but opposite in direction. This means that both cubes will rotate after the collision. The rotation will be due to the force of impact, which causes the cubes to change their orientation in space.

**Observations**: Qwen2.5-Omni generates the correct outcome first (linear motion) and then incorrectly outputs that both objects will rotate after the collision. It also provides invalid reasoning for this output.

**Gemma3-12B** (Kamath et al., 2025): Here's an analysis of what would likely happen when the green cube collides with the blue cube, based on physical laws:

**Outcome of the Collision:**

Given that the cubes have equal mass and assuming an elastic collision (meaning no energy is lost due to friction or deformation), here's what would happen:

1. **Momentum Exchange:** Momentum is conserved in the collision. The green cube is initially moving, and the blue cube is stationary. During the collision, the green cube will transfer some of its momentum to the blue cube. 2. **Equal and Opposite Velocities:** Because the masses are equal, ideally the green cube will come to a stop and the blue cube will move away with approximately the same velocity that the green cube originally had. 3. **Equal Velocity:** As a result of the above, both cubes would move in a similar direction with equal velocity.

**Will the Objects Rotate?**

It's highly unlikely that either cube will rotate significantly during the collision. Here's why:

* **Impulse Applied:** The force during the collision is applied along a line. For rotation to occur, there would need to be an off-center force (a torque). Unless the collision isn't head-on, it's unlikely to generate such a torque. * **Shape:** Cubes are not ideal shapes for rotating due to collisions. Spheres would rotate the most.

**\*\*Important Considerations/Simplifications:\*\***

\* **\*\*Elasticity:\*\*** We've assumed an elastic collision. In reality, some energy would likely be lost as heat or sound, making the outcome slightly different. \* **\*\*Perfect Alignment:\*\*** A perfectly head-on collision is assumed. If the cubes collide at an angle, the situation becomes more complex, and there could be some rotation, but it would still likely be minimal. \* **\*\*Friction:\*\*** Any friction with the surface would also influence the outcome, slowing the cubes down.

Let me know if you'd like a deeper dive into any of these aspects!

**Observations**: Gemma3-12B provides both correct and incorrect reasoning about the outcome regarding the motion of the objects after the collision. It also provides accurate reasoning about the possibility of rotation, although it adds a partially true statement that cubes are less likely to rotate.

**Aria** (Li et al., 2024b): When the green cube collides with the blue cube, assuming they have equal mass and the collision is head-on, the outcome will depend on the nature of the collision. If it is an elastic collision, both cubes will bounce off each other with the same speed they approached with. If it is an inelastic collision, they might stick together and move as one object.

In either case, since the cubes are symmetrical and have no initial angular momentum, they will not rotate as a result of the collision. The collision will primarily affect their linear momentum.

**Observations**: Aria states and applies the principles of linear and angular momentum conservation.

**Conclusion**: Some LMMs accurately predict the change in linear velocity, and then incorrectly add that the objects may rotate after the collision. Moreover, the laws stated to support their rotation argument are irrelevant to rotation. Comparing these outputs with those in § E.6.1, we conjecture that these models generated incorrect reasoning since they were explicitly prompted to make predictions about the rotation of the objects. However, we also note that some models accurately predicted that the colliding objects could rotate if certain conditions, such as colliding at an angle or frictional surfaces, were met. We believe that the apparent inductive physical reasoning in § 4.4 was due to these inaccurate beliefs that objects could rotate after a collision.

### E.8    DO THE FINDINGS HOLD FOR MORE COMPLEX SCENES?

In this section, we increase the scene complexity of INPHYRE by including background objects and randomly positioning illumination sources and cameras during the rendering stage. Our intuition is that, since LMMs failed to demonstrate inductive physical reasoning in a simple environment such as INPHYRE, they should fail worse as scene complexities increase. However, since the nature of object attributes in the complex version of INPHYRE (which we refer to as INPHYRE$_{com}$) is similar to INPHYRE, we may also expect the LMMs to at least partially overcome the scene complexities.

Similar to our main experiments in § 4, we will evaluate the LMMs on (1) regular scenarios without any demonstration samples, (2) regular scenarios with demonstration samples, and (3) irregular scenarios with demonstration samples. As before, inductive physical reasoning will be measured as the difference in performance between irregular and regular scenarios when demonstration samples were available. Our findings are discussed below.

**Findings**: Fig. 19a shows the zero-shot performance of LMMs in the regular scenarios of INPHYRE$_{com}$. Similar to our results on INPHYRE in Tab. 2, the LMMs perform fairly well in LMC (Reg.) and SB (Reg.), and poorly in Wall (Reg.). When demonstration samples comprising video frames and question-answer pairs are provided, the results improve significantly, and multiple LMMs achieve >90% accuracy in many scenarios (Fig. 19b). We measure inductive physical reasoning in Fig. 19c by comparing the 3-shot performance of LMMs in irregular scenarios against the LMMs' best corresponding performance in regular scenarios under zero-shot and few-shot settings. The results appear similar to those in Fig. 4 with many LMMs struggling in Red-LMC, Red-Pass, SB, and CC. Like in Fig. 4, the largest average drop in performance was observed for InternVL3-2B. One notable difference between the results on INPHYRE and INPHYRE$_{com}$ is that Gemma3-4B showed a larger drop in performance in Fig. 19c compared to Fig. 4. Also, although Qwen2.5-Omni has a smaller drop in Fig. 19c compared to Fig. 4, we also point out that Qwen2.5-Omni's performance in the regular scenarios was proportionately worse on INPHYRE$_{com}$. Fig. 19d shows the further drop in accuracy when the demonstration samples include only the video frames, and not the question-answer

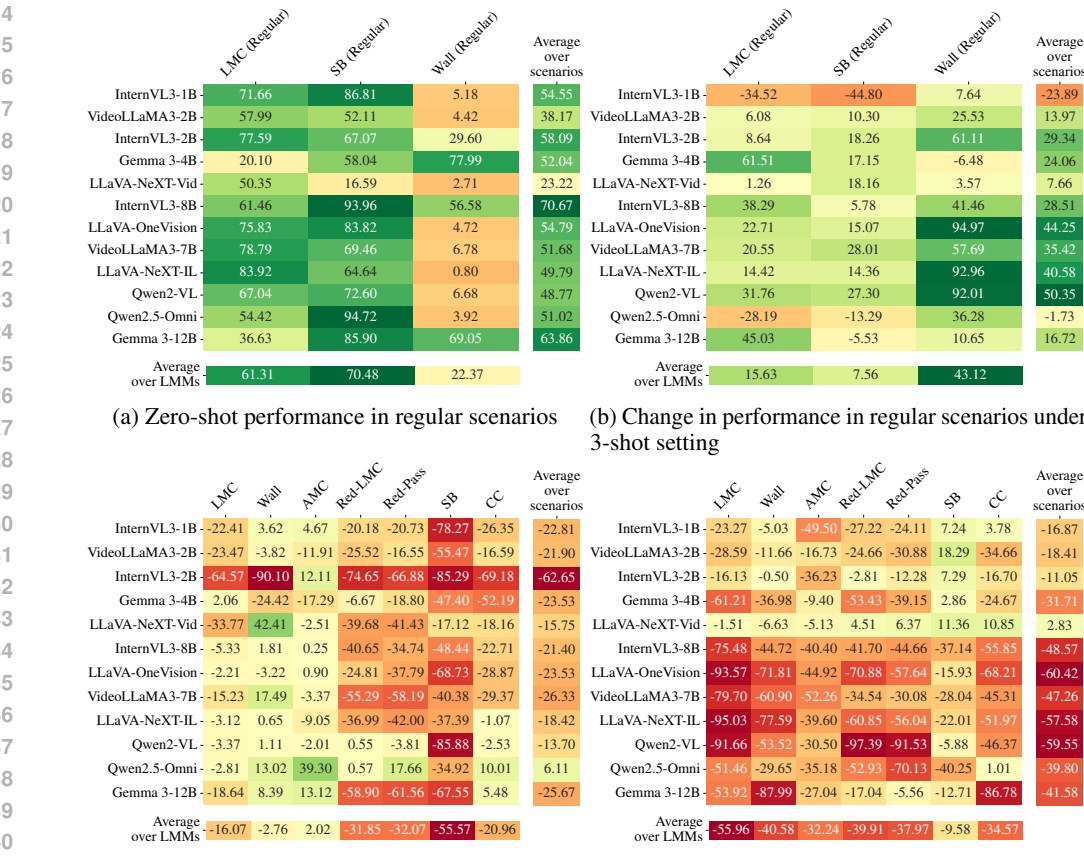

(a) Zero-shot performance in regular scenarios

(b) Change in performance in regular scenarios under 3-shot setting

(c) Change in performance in irregular scenarios under 3-shot setting

(d) Change in performance in irregular scenarios under 3-shot and video-only setting

Figure 19: Results on INPHYRE_com – a version of INPHYRE with more scene complexities in the form of changes in camera pose, lighting conditions, and background objects.

pairs. Similar to our previous observations in Fig. 5, we find that most LMMs show considerable language bias. LLaVa-NeXT-Video is the only model that did not show language bias in both INPHYRE and INPHYRE_com.

Although INPHYRE_com had more scene complexities than INPHYRE, the evaluation results were similar. This indicates that added complexities did not challenge most LMMs. In future works, we aspire to generate more complex scenes with realistic degradations that pose significant perception challenges to the LMMs. Nonetheless, the absence of inductive physical reasoning in a visually simple dataset, such as INPHYRE that does not confound our core evaluation, indicates the absence of inductive physical reasoning in more complex scenarios.

## E.9 HUMAN EVALUATION FOR INPHYRE

In this section, we evaluate how well humans fare on INPHYRE. We believe that future works on inductive physical reasoning can aspire to match or outperform this human baseline, and draw cognitive insights from our subject's explanations. To obtain the human baseline, we recruited 10 subjects and informed them about our research problem. We evaluated these subjects under a more difficult setting compared to our evaluation of LMMs – each subject was provided only one demonstration sample without the associated question-answer pair. Since the underlying logic to be inferred is the same for all samples in a given scenario, we evaluated the subjects on only one sample from each scenario. The demonstration samples and the evaluation samples from all scenarios were put together in a slideshow and shared with the subjects, along with the instructions on how to

complete their task. They were provided with the same information about the tasks as the LMMs. The only additional assistance provided during the test was regarding the meaning of certain questions.

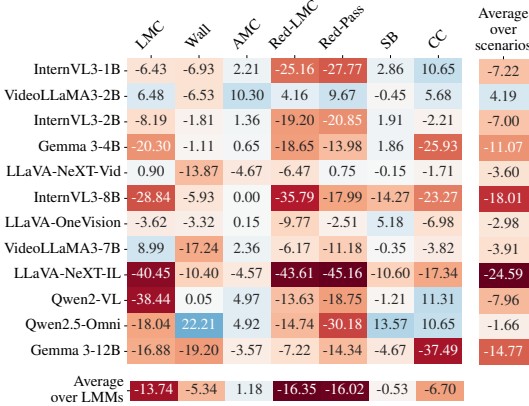

| | LMC (Reg.) | SB (Reg.) | Wall (Reg.) | LMC | Wall | AMC | Red-LMC | Red-Pass | SB | CC |
|---|---|---|---|---|---|---|---|---|---|---|
| Subject 1 | 0 | 1 | 1 | 0 | 1 | 1 | 1 | 1 | 1 | 1 |
| Subject 2 | 1 | 1 | 1 | 1 | 1 | 1 | 0 | 0 | 1 | 0 |
| Subject 3 | 1 | 1 | 1 | 1 | 1 | 1 | 0 | 0 | 1 | 0 |
| Subject 4 | 1 | 1 | 1 | 1 | 1 | 1 | 0 | 0 | 1 | 0 |
| Subject 5 | 1 | 1 | 1 | 0 | 1 | 1 | 1 | 0 | 1 | 1 |
| Subject 6 | 1 | 1 | 1 | 1 | 1 | 1 | 1 | 1 | 1 | 1 |
| Subject 7 | 1 | 1 | 1 | 1 | 1 | 1 | 1 | 0 | 1 | 1 |
| Subject 8 | 1 | 1 | 0 | 1 | 1 | 1 | 0 | 0 | 0 | 0 |
| Subject 9 | 1 | 1 | 1 | 0 | 1 | 1 | 1 | 0 | 1 | 1 |
| Subject 10 | 1 | 1 | 1 | 1 | 1 | 1 | 1 | 0 | 1 | 1 |
| Mean | 0.9 | 1.0 | 0.9 | 0.7 | 1.0 | 1.0 | 0.6 | 0.2 | 0.9 | 0.6 |

Figure 20: Accuracy of human subjects in various scenarios. "1" indicates correct prediction, and "0" indicates otherwise.

The results are shown in Fig. 20. We observe that most subjects had little trouble in both regular and irregular scenarios, despite being provided only one visual demonstration without any textual description. Here, the human subjects inferred laws from demonstration samples (*e.g.*, "the speed remained constant after collision") and applied them to the evaluation sample, irrespective of any contradictions between the inferred law and the premise given with the evaluation sample (*e.g.*, "the objects have equal mass and undergo elastic collision"). However, the subjects did struggle with Red-LMC and Red-Pass, where the required reasoning depended on the color of the objects. For these scenarios, we provided four demonstration samples: two scenarios with red-colored objects that violate the true physics, and two scenarios without any red-colored objects and that do not violate any true physics. Since we did not inform them specifically to use color information for reasoning in these scenarios, many subjects answered that there was not enough information to predict the answer.

### E.10 DOES USING ALL EVALUATION FRAMES IMPROVE PERFORMANCE?

| | LMC | Wall | AMC | Red-LMC | Red-Pass | SB | CC | Average over scenarios |
|---|---|---|---|---|---|---|---|---|
| InternVL3-1B | -6.43 | -6.93 | 2.21 | -25.16 | -27.77 | 2.86 | 10.65 | -7.22 |
| VideoLLaMA3-2B | 6.48 | -6.53 | 10.30 | 4.16 | 9.67 | -0.45 | 5.68 | 4.19 |
| InternVL3-2B | -8.19 | -1.81 | 1.36 | -19.20 | -20.85 | 1.91 | -2.21 | -7.00 |
| Gemma 3-4B | -20.30 | -1.11 | 0.65 | -18.65 | -13.98 | 1.86 | -25.93 | -11.07 |
| LLaVA-NeXT-Vid | 0.90 | -13.87 | -4.67 | -6.47 | 0.75 | -0.15 | -1.71 | -3.60 |
| InternVL3-8B | -28.84 | -5.93 | 0.00 | -35.79 | -17.99 | -14.27 | -23.27 | -18.01 |
| LLaVA-OneVision | -3.62 | -3.32 | 0.15 | -9.77 | -2.51 | 5.18 | -6.98 | -2.98 |
| VideoLLaMA3-7B | 8.99 | -17.24 | 2.36 | -6.17 | -11.18 | -0.35 | -3.82 | -3.91 |
| LLaVA-NeXT-IL | -40.45 | -10.40 | -4.57 | -43.61 | -45.16 | -10.60 | -17.34 | -24.59 |
| Qwen2-VL | -38.44 | 0.05 | 4.97 | -13.63 | -18.75 | -1.21 | 11.31 | -7.96 |
| Qwen2.5-Omni | -18.04 | 22.21 | 4.92 | -14.74 | -30.18 | 13.57 | 10.65 | -1.66 |
| Gemma 3-12B | -16.88 | -19.20 | -3.57 | -7.22 | -14.34 | -4.67 | -37.49 | -14.77 |
| Average over LMMs | -13.74 | -5.34 | 1.18 | -16.35 | -16.02 | -0.53 | -6.70 | |

Figure 21: Difference in 2-shot performance when all evaluation frames are provided instead of just the first frame. The results indicate that despite having access to all the frames, the LMMs generally struggle to infer the dynamics from visual inputs, further asserting language bias.

In our main experiments, we provided only the initial frame from each evaluation sample to the LMMs, following the standard procedure for predictive physical reasoning. Thus, we posed physical

reasoning as a normative task that requires LMMs to apply some rule to make their predictions. Intuitively, if all the frames from the evaluation sample were provided to the LMM, then any inferred physical rule becomes redundant. Interestingly, through the following experiments, we find that even when provided with all the frames in the evaluation sample, LMMs fail to perceive object dynamics from the visual inputs.

In our current experiment using all evaluation frames, we provide only two demonstration samples since using more frames (hence, more tokens) results in nearly approaching the LMMs' token limit. We compare the performance in the current experiment with that from a similar setup, but with only one evaluation frame. The differences in accuracy are shown in Fig. 21. Compared to the main experiments where only the initial frame was provided, we notice that across scenarios and LMMs, performance either remained roughly the same or decreased significantly. Our results convey two important findings:

1. LMMs struggle to perceive object dynamics from visual samples when the underlying physics contradict the existing parametric knowledge.

2. This behavior is further evidence of language bias, as LMMs were largely able to apply the underlying physical laws in Tab. 8 when these laws were provided as explicit reasoning in the demonstration samples. In § F, we will show that Gemma3-12B spent an order of magnitude less attention on image tokens compared to the text tokens, echoing similar findings in prior works. We believe that the phenomenon we observed in § F, along with the general *attention dilution* due to more tokens, is the primary reason for the accuracy drop observed in Fig. 21.

### E.11 Effect of Structural Perturbations to Prompts

This section evaluates the effect of minor prompt changes on the accuracy of LMMs. These minor changes, which we refer to as "structural perturbations," do not affect the semantic meaning of the demonstration samples. We ablate the effects of the following types of structural perturbations: (1) changing object attributes in the demonstration samples, (2) changing order of demonstration samples, (3) minor text rewrites in the question and the options (e.g., "the speed will increase" or "the speed increases"), and (4) option ordering in the evaluation and demonstration queries.

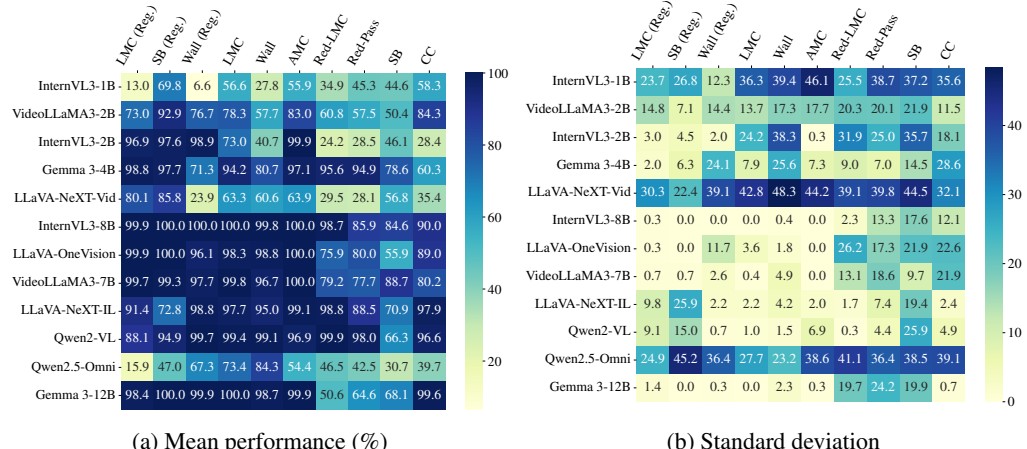

(a) Mean performance (%)         (b) Standard deviation

Figure 22: Mean performances of various LMM-scenario combinations and their standard deviations when there are structural perturbations in the prompt.

For this experiment, we create multiple smaller versions of INPHYRE with only the first 100 samples from each scenario. In each version, the structural perturbations are randomly chosen. The results over 10 such versions are shown in Fig. 22. For each LMM and each scenario, we compute the mean performance (Fig. 22a), the standard deviation in performance (Fig. 22b), and the coefficient of variation (CoV) in the performance (Fig. 23). The CoV is measured as the ratio between the mean

performance and its standard deviation. Our primary metric will be the CoV, as it captures the relative variation due to the chosen structural perturbations.

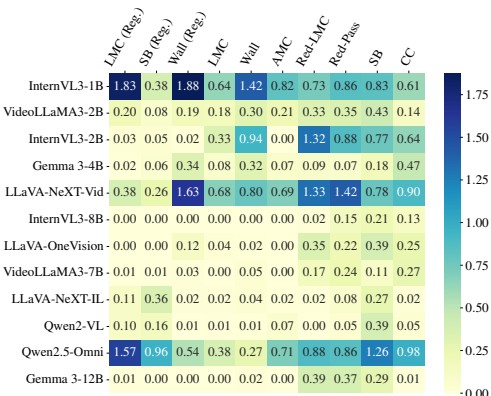

Figure 23: The coefficients of variation (CoV) for LMM-scenario combinations. CoV is computed as the ratio between the corresponding mean (Fig. 22a) and standard deviation (Fig. 22b) values.

The CoV for each LMM-scenario combination is shown in Fig. 23. Most LMM-scenario combinations have very low CoV values, indicating the robustness of the evaluation to structural perturbations. Some LMM-scenario combinations have unacceptable levels of CoV[7]. However, these values could either be due to the LMMs or the scenario setups. The higher CoV values concentrate around smaller LMMs ($< 7$B parameters), and there is no noticeable pattern where a specific scenario has consistently high CoV. This suggests that higher CoV values are due to the sensitivity of the smaller LMMs, and not due to the benchmark design.

# F  WHY DO LMMS PERFORM POORLY IN IRREGULAR SCENARIOS

We observed that LMMs generally struggled in irregular scenarios, although they performed well in regular scenarios. To develop methods to improve inductive physical reasoning in LMMs, we must first understand its causes. However, there are several practical challenges in investigating the causes of poor inductive physical reasoning in LMMs. The foremost challenge is the difficulty in interpreting the outputs of an LMM. Common approaches inspect the hidden states and the attention maps between tokens. In this section, we will use attention maps and linear probes on hidden states to gain insights into why LMMs fail in irregular scenarios. Specifically, we will (1) evaluate the attention values over image tokens and text tokens in Gemma3-12B, and (2) compare the hidden states of a pre-trained InternVL3-1B with those of a fine-tuned InternVL3-1B that demonstrated considerable inductive physical reasoning in § E.4 on regular and irregular scenarios. Although our findings do not fundamentally explain the lack of inductive physical reasoning in LMMs, we believe this discussion will foster future efforts.

## F.1  ANALYSIS OF ATTENTION MAPS IN GEMMA3-12B

We plot the normalized attention values over the tokens in the final layer of Gemma3-12B in Fig. 24. To obtain these plots, we first choose three corresponding regular-irregular scenario pairs – LMC and LMC (Reg.), SB and SB (Reg.), and Wall and Wall (Reg.). In each pair, the scenarios only vary in the underlying physical law. Note that the exact visual attributes may vary in the selected samples. The plots are obtained by summing the attention values from 20 different samples from these regular-irregular scenario pairs. Since each sample may have a different number of text tokens, we use the image tokens as the "hinge" around which we arrange the pre-image and the post-image text tokens. The x-axis shows the position of these tokens w.r.t. the image tokens. Orange- and blue-colored regions denote text and image tokens, respectively. The black dashed lines show the average attention values for each colored segment.

---

[7]Following standard practice, an acceptable CoV is less than 0.3.

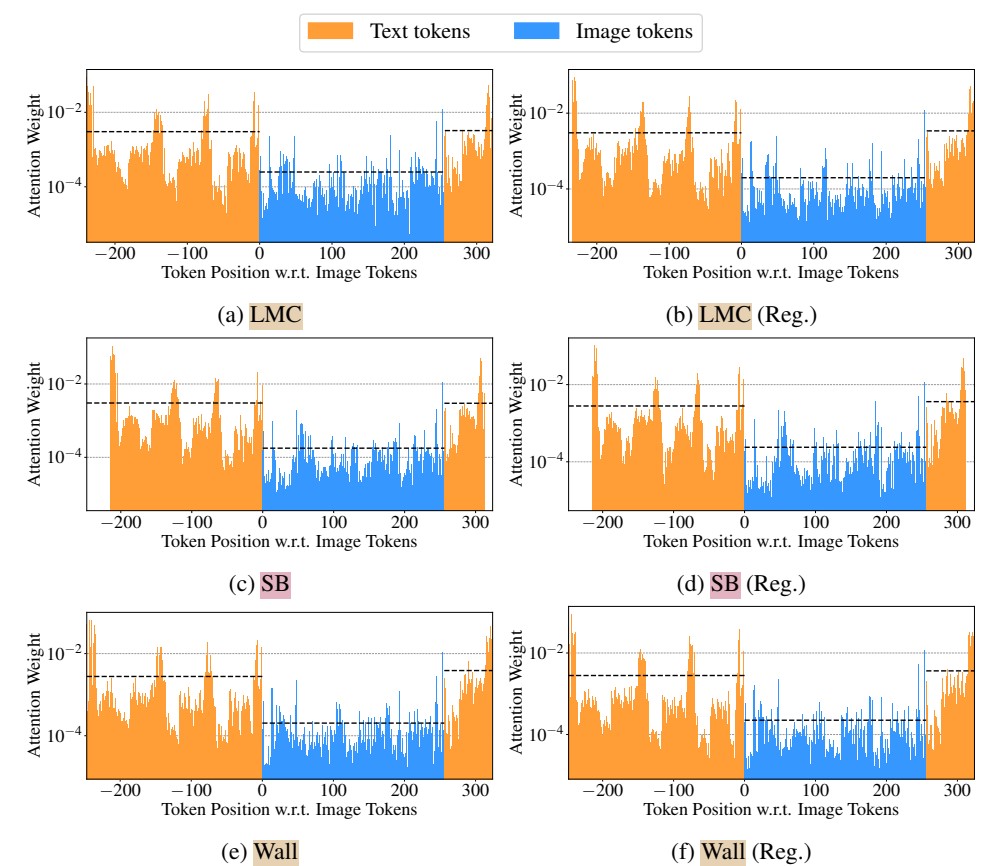

Figure 24: Attention values over image tokens are around an order of magnitude less compared to those over text tokens.

**Gemma3-12B spent an order of magnitude less attention on image tokens**, compared to text tokens. The lower attention values over image tokens are a possible reason for the observed language bias. It may also explain why LMMs failed to perceive the underlying physical laws even when all frames (including those that show the outcome of the collision event) from the evaluation sample were provided. Similar findings were reported in (Chen et al., 2024a), albeit in the context of natural images.

## F.2 ANALYSIS OF HIDDEN STATES IN INTERNVL3-1B

To understand why an LMM showed poor performance in irregular scenarios, we compare its hidden states to a similar LMM that performed well in irregular scenarios. To that end, we compare a pre-trained (PT) InternVL3-1B with a fine-tuned (FT) InternVL3-1B. The results for the FT model were shown in § E.4. We compare the hidden states of the two models by training linear probes on them. We consider two target tasks for the linear probes: (T1) predict whether the given scenario is regular or irregular, and (T2) predict the attributes (color and shape) of the objects in demonstration and evaluation samples. Through (T1), we understand whether the LMMs learn different hidden states for regular and irregular scenarios. Through (T2), we check for any discrepancy in the amount of information carried by the hidden states from demonstration and evaluation samples.

**(T1) Can we predict the underlying scenario from the hidden states?** The task here is to predict whether the hidden states correspond to samples from a regular scenario or an irregular scenario. We evaluate PT and FT LMMs on this task using the three corresponding pairs of regular-irregular scenarios that we used in our previous experiment – LMC, SB, and Wall. We also vary the number of demonstration samples from one to three. To train the linear probe, we collect hidden states from all samples in each of the scenarios. The results are shown in Fig. 25.

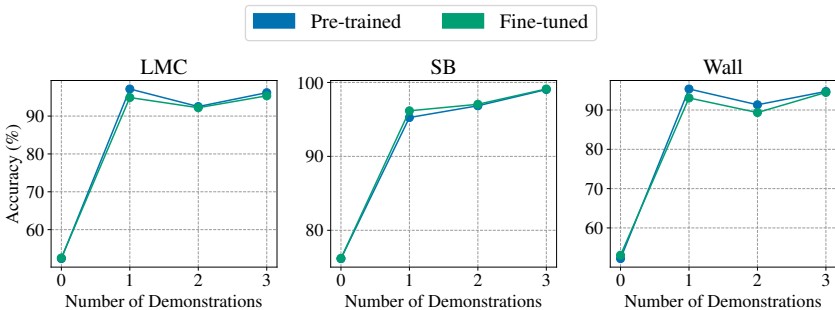

Figure 25: Accuracy of linear probes in predicting the underlying scenario from hidden states of pre-trained and fine-tuned models.

We find that the hidden states from both PT and FT LMMs have sufficient information to classify the underlying scenario with over 90% accuracy. The classification accuracy is not affected by the number of demonstration samples. We also compare these classification results against the case where no demonstration samples are provided. Note that without demonstration samples, the evaluation frame and the accompanying question-answer pair do not provide any discriminative information to suggest whether the underlying scenario is regular or not. Therefore, the classification accuracy in the case with no demonstration samples only acts as a baseline that validates our observations with $\geq 1$ demonstration samples. In all scenario pairs, the linear probes achieve significantly less classification accuracy without demonstration samples. In LMC and Wall, they only match random chance performance. In the next task, we will understand how PT and FT LMMs understand the object attributes from the prompt differently.

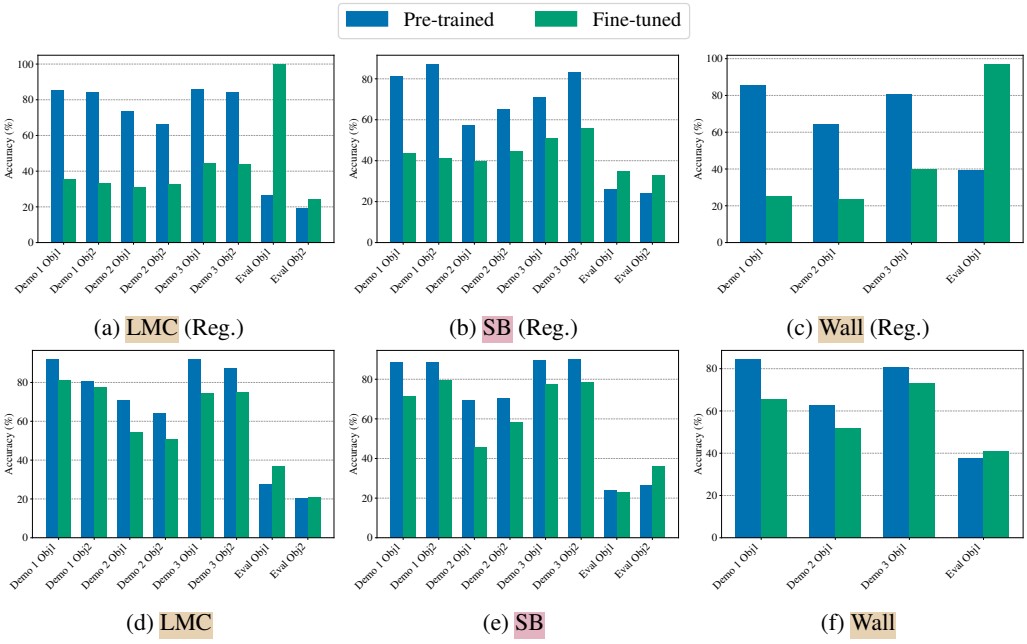

Figure 26: Accuracy of linear probes on hidden states from pre-trained and fine-tuned InternVL3-1B models in classifying the colors of the objects in demonstration and evaluation samples.

**(T2) Do PT and FT LMMs perceive object attributes differently?** Through task (T1), we saw that the hidden states from both PT and FT InternVL3-1B models carried sufficient information from the demonstration samples to classify the underlying scenario. It is not possible to semantically evaluate these scenario-specific differences in the hidden states exhaustively. Therefore, as a first step, in this task, we will quantify the object attribute information in the hidden states. Similar to task (T1),

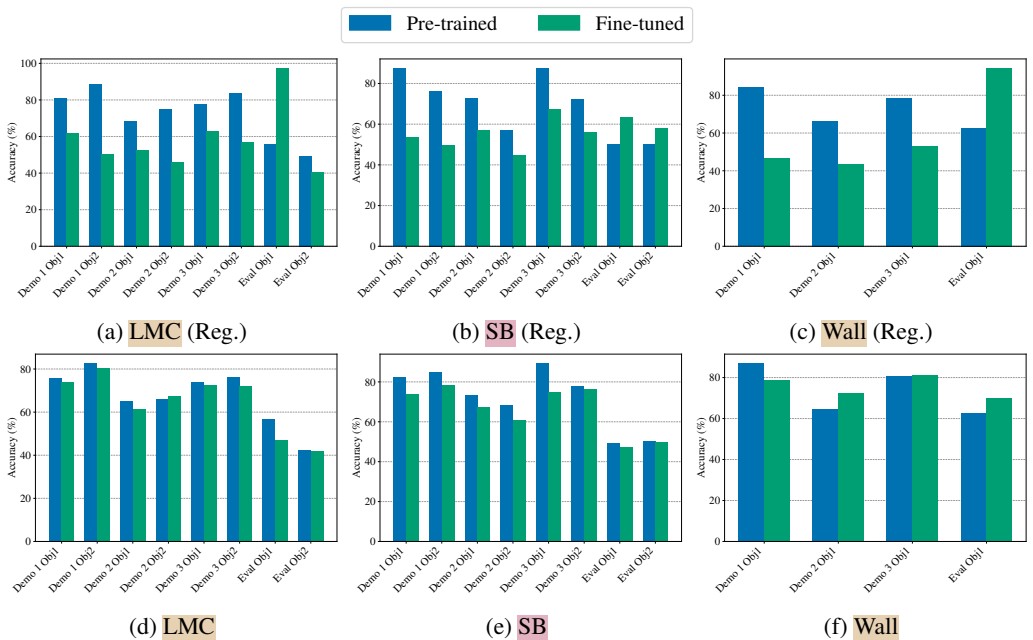

Figure 27: Accuracy of linear probes on hidden states from pre-trained and fine-tuned InternVL3-1B models in classifying the shapes of the objects in demonstration and evaluation samples.

we will use linear probes on the hidden states to classify the shapes and the colors of the objects in demonstration and evaluation samples. In total, the objects in INPHYRE were composed from three shapes and ten colors. The frames in LMC and SB showed two objects, while those in Wall showed only one. We will compare the attribute classification accuracy across scenarios and between PT and FT InternVL3-1B models.

Figs. 26 and 27 show the attribute classification accuracies on color and shape, respectively, of linear probes on the hidden states from PT and FT models in various scenarios. In each figure, the top row shows regular scenarios, while the bottom row shows irregular scenarios. By using accuracy as a proxy for the amount of information in the hidden states about the predicted attributes, we make the following observations:

(O1)  PT models contain more attribute-specific information about the objects in the demonstration samples than the objects in the evaluation sample. This pattern is evident across various scenarios and on both color and shape attributes.

(O2)  **FT models adapt their hidden states to the underlying scenario**. In irregular scenarios, their hidden states contain more information from the demonstration samples, and in regular scenarios, their hidden states contain more information from the evaluation sample. Indeed, in regular scenarios, the demonstration samples are not required since the parametric knowledge is sufficient. However, in irregular scenarios, demonstration samples are key in physical reasoning. Thus, this adaptation aligns with the expected behavior of an LMM in irregular scenarios.

(O1) is a surprising result, since we expect PT models to have less information about the demonstration samples due to their poor performance in irregular scenarios. Moreover, it is not clear from the linear probe whether this information came from the frames or the question-answer pairs in the prompt. (O2) suggests that fine-tuning introduces adaptive behavior when the fine-tuning dataset potentially contradicts parametric knowledge. While (O2) is an interesting finding, it does not explain whether information in the hidden states of PT and FT models are of different natures.

**Conclusion of our analysis**: Although our results indicate that LMMs indeed understand regular and irregular scenarios differently, they do not shed light on what causes them to be different. Understanding the cause of this difference is a prerequisite for building solutions to improve inductive

physical reasoning. Our findings through task (T2) – that LMMs adapted their visual attribute reasoning after fine-tuning – are encouraging as the first steps in understanding inductive physical reasoning. Our analysis was also limited by the lack of reliable tools to interpret the outputs and the hidden states of LMMs. Therefore, future analysis would require better interpretability tools for LMMs, specifically regarding the visual inputs.

# G  USE OF LLMs AND GENERATIVE AI

Grammarly[8] and Writefull[9] (embedded in Overleaf) were used in correcting grammatical errors and spellcheck. LLMs were used to obtain feedback on the writing style. They were not used in generating sentences or summarizing paragraphs. Some figures in the paper used clip-art style images generated using Gemini[10] – the cartoon image of a man holding a torch in the title, robot head, electric bulb, and book in Fig. 1, and palette in Fig. 3. Generative AI was not used in dataset creation.

---

[8]https://app.grammarly.com/
[9]https://www.writefull.com/
[10]https://gemini.google.com/app

