# OpenReview forum: "InPhyRe Discovers: Large Multimodal Models Struggle in Inductive Physical Reasoning"
_ICLR.cc/2026/Conference — Submitted to ICLR 2026_

### Official Review · Reviewer_WEJ2 · 2025-10-15

**Soundness:** 3
**Presentation:** 3
**Contribution:** 2
**Rating:** 4
**Confidence:** 3

**Summary:**

This paper introduces INPHYRE, a VQA benchmark designed to evaluate inductive physical reasoning in LMMs. INPHYRE assesses whether LMMs can infer and apply unseen or violated physical laws from demonstration videos. The benchmark uses algorithmically generated synthetic videos of collision events that violate real-world physical laws. Through extensive evaluation of 13 open-source and proprietary LMMs, the authors find that LMMs struggle with inductive physical reasoning, rely heavily on language bias, and often ignore visual inputs.

**Strengths:**

1. First benchmark to systematically evaluate inductive physical reasoning in LMMs using physically impossible scenarios.

2. This paper's writing is well-organized, clearly written, and richly illustrated.

**Weaknesses:**

1. Limited Insight: The paper's conclusion that "LMMs struggle with inductive physical reasoning (L68)" may stem from the limited scale of SFT data and potential conflicts between counterfactual scenarios and pre-trained knowledge. Moreover, the observation that "inductive physical reasoning in LMMs suffers from language bias and largely ignores visual inputs" has already been extensively discussed in prior literature.

2. Lack of Closed-Source Model Evaluation: The study does not include evaluations on closed-source models such as GPT-4, Gemini, or Doubao. It is recommended that the authors incorporate results from at least one of these models to enhance the comprehensiveness of their analysis.

3. Narrow Physical Scope: While the benchmark concentrates mainly on collision events and mechanical reasoning, the authors suggest that their findings are likely generalizable. However, key physical domains such as fluid dynamics and thermodynamics remain unexamined, limiting the scope of the study.

**Questions:**

See weakness.

---

> ### Author Response · Authors · 2025-11-21
> **Response to reviewer WEJ2**
>
> We thank the reviewer for their comments. We address the concerns regarding insight depth and scope by pointing to specific results in our **original submission** (Closed-Source Evaluation) and **new experiments** added in the revision (attention map analysis (App. F.1), all-frames evaluation (App. E.10)).
>
> **Summary of the rebuttal**:
>
> 1. **Potential conflicts with pre-trained knowledge**: LMMs perform poorly in the SB scenario (Fig. 4, page 8), although it does not contradict any physics prior. This indicates the poor performance is indeed due to poor inductive physical reasoning, not due to contradicting physics knowledge.
> 2. **We extend prior evidence of language bias** through (1) evidence of poor performance without textual demos (Sec. 4.5), (2) chain-of-thought experiments (App. E.3), (3) [**new**] analysis of attention values over image tokens (App. F.1), and (4) [**new**] demonstrating poor performance when all evaluation frames are given (App. E.10).
> 3. **Lack of closed-source model evaluation**: Our original submission contained results on closed-source models such as GPT 4.1 and Gemini 2 and 2.5 (App. E.2, page 25).
> 4. **Narrow physical scope**: Collision events are likely to be well-understood by LMMs. Since LMMs failed to show IPR even on this simple scenario, we expect our conclusions to generalize. Our new experiments on a visually more complex version of InPhyRe support this.
>
> > W1. (part 1) Limited Insight: The paper's conclusion that "LMMs struggle with inductive physical reasoning (L68)" may stem from the limited scale of SFT data and potential conflicts between counterfactual scenarios and pre-trained knowledge.
> >
>
> We provide **counter-evidence** against any “potential conflict”:
>
> SB scenario follows true physics and only challenges the visual prior (“large = heavy”) that LMMs may have. If LMMs’ failure in other scenarios was due to conflicts with pre-trained physics knowledge, they should have succeeded in SB. Instead, they ignore the context (”the small object is heavier”) and fail in SB. A tabular form of the demos’ relation with physics prior is given below:
>
> | Demos’ relation with prior | Performance | Experiment |
> | --- | --- | --- |
> | Align | Good | Sec. 4.3, page 7 |
> | Contradict | Poor | Sec. 4.4, page 7 (all scenarios except SB) |
> | Does not contradict | Poor | Sec. 4.4, page 7 (SB scenario) |
>
> > W1. (part 2) Moreover, the observation that "inductive physical reasoning in LMMs suffers from language bias and largely ignores visual inputs" has already been extensively discussed in prior literature.
> >
>
> While language bias is known, we provide **four specific mechanisms** unique to physical reasoning that have not been reported before:
>
> 1. The accuracy drops when the demonstration samples consist of only video frames (Sec 4.5, page 8),
> 2. articulating the underlying physical laws explicitly through chain-of-thought (CoT) prompting improves performance (App. E.3, page 26),
> 3. [**new experiment**] LMMs spend around an order of magnitude less attention over image tokens compared to text tokens (App. F.1, page 45).
> 4. [**new experiment**] LMMs struggled to infer the underlying physical laws even when all evaluation frames were provided (App. E.10, page 43).
>
> > W2. Lack of Closed-Source Model Evaluation: The study does not include evaluations on closed-source models such as GPT-4, Gemini, or Doubao. It is recommended that the authors incorporate results from at least one of these models to enhance the comprehensiveness of their analysis.
> >
>
> The results on closed-source models **were included in the original submission** (line 298, page 6 and App. E.2, page 25). We found that closed-source models, including GPT 4.1 models and Gemini 2 and 2.5 models, also struggle in inductive physical reasoning.
>
> > W3. Narrow Physical Scope: While the benchmark concentrates mainly on collision events and mechanical reasoning, the authors suggest that their findings are likely generalizable. However, key physical domains such as fluid dynamics and thermodynamics remain unexamined, limiting the scope of the study.
> >
>
> A desirable domain to evaluate IPR is one:
>
> 1. whose physical laws are sufficiently learnable from visual inputs,
> 2. that has ubiquitous presence in the training set, and
> 3. has minimum confounding visual features.
>
> **Collision events satisfy the above desiderata**, as physical laws such as momentum conservation are ubiquitous in natural videos, and hence likely to be *well-understood* by LMMs.
>
> Since our findings suggest that IPR is absent even in a simple, well-understood domain like collision, we expect our conclusions to extend to more complex, specialized domains, such as aerodynamics and thermodynamics.
>
> Our **new experiments** using a visually more complex InPhyRe benchmark result in similar or poorer IPR (App. E.8, page 41).
>
> ---
> ## We request the reviewer to raise their rating if we have addressed all their concerns satisfactorily.

---

### Official Review · Reviewer_q4tW · 2025-10-17

**Soundness:** 1
**Presentation:** 3
**Contribution:** 2
**Rating:** 2
**Confidence:** 2

**Summary:**

This paper presents InPhyRe, a benchmark designed to evaluate the inductive physical reasoning of Large Multimodal Models. By constructing synthetic visual scenarios that intentionally violate universal physical laws, InPhyRe provides a method to quantitatively measure a model's ability to infer and apply novel physical principles from in-context visual examples. The benchmark is used to evaluate over LMMs on their parametric knowledge, inductive reasoning, and susceptibility to language bias. The results show that current models struggle significantly on InPhyRe, demonstrating weak inductive capabilities and a strong reliance on language cues over genuine visual understanding.

**Strengths:**

1. Clear definition and writing. The paper is clearly written and very effectively identifies the current shortcoming of current LLM/VLMs in inductive physical reasoning.
2. The authors conducted comprehensive experiments on curent LMMs.
3. I think the paper overall challenges the in-context viusal reasoning capabilities of LMMs, pushing the community to develop more adversarial and out-of-distribution benchmarks to probe the true limits of these models.

**Weaknesses:**

1. My biggest concern is the positioning of this paper. The authors select examples that violate physical principles in order to avoid being covered by the LLM's pretrained parametric knowledge. However, I think the model's weak performance in such an environment only indicates that it is dominated by a strong physics prior, not necessarily that its inductive reasoning ability is weak. To the best of my knowledge, experiments related to the violation-of-expectation paradigm in psychology [1] have shown that human infants, when encountering phenomena that appear to violate their expectations, also tend to assume their internal world model is correct, rather than overturning their established mental simulation model based on a few sampled trajectories.

2. There is a huge gap between the benchmark's design and the real world applicability. Real-world scenarios, such as autonomous driving on diverse ground materials mentioned, are still physically consistent and will not feature impossible events like violation of object permanence.

3. I think the dataset design is a bit simple. The dataset only includes basic geometric shapes on a flat surface, meaning the model does not need to handle any real-world visual challenges such as lighting and shadows, occlusion, complex backgrounds, or material reflections. Have the authors considered including more diverse object materials (e.g., soft bodies/cloth) and richer shapes?

4. The paper lacks further experiments to investigate the cause of this phenomenon, such as visualizing specific attention maps or applying linear probing to inner representations.

[1] Piloto LS, et al. Intuitive physics learning in a deep-learning model inspired by developmental psychology. Nature Human Behaviour, 2022.

**Questions:**

The authors argue that "inductive physical reasoning, is a hallmark of intelligence that humans develop at a very young age." Previous experiments on inductive and physical reasoning in infants were mostly conducted in real-world environments and did not involve scenarios that violate real-world physics. Could the authors provide further human studies to support this claim?

I'll be glad to increase my score if the concerns above are properly resolved.

---

> ### Author Response · Authors · 2025-11-21
> **Response to reviewer q4tW -  part 1/2**
>
> We thank the reviewer for their comments. To our understanding, the reviewer was primarily concerned with whether our findings suggest a strong physics prior or poor inductive reasoning, and how our experiments are connected to child cognition experiments.
>
> **Summary of the rebuttal**
>
> 1. **Strong physics prior vs inductive physical reasoning (IPR)**: LMMs perform poorly in the SB scenario (Fig. 4, page 8), although it does not contradict any physics prior. This indicates the observed failure is not due to any strong physics prior.
> 2. **Connection to cognition experiments**: [R4, R5, R6] showed that infants and children can adapt their internal model when provided with contradicting physical environments, in addition to the violation of expectation.
> 3. **Simple dataset design** allowed us to establish the absence of IPR in LMMs without confounding factors. If an LMM struggles in IPR on a simple scene, it can only perform worse in a more complex scene. **New experiments on a more complex version** of InPhyRe (App. E.8, page 41) confirm this.
> 4. **Explaining poor IPR**: We have **added new preliminary findings** (App. F, page 45) using linear probes and attention maps:
>     1. The hidden states of a corresponding regular-irregular scenario pair carry enough information to distinguish between the scenarios,
>     2. A model fine-tuned on InPhyRe adapted its hidden states to scenario changes, and
>     3. Image tokens get less attention by an order of magnitude compared to text tokens.
>
> > W1. My biggest concern is the positioning of this paper. The authors select examples that violate physical principles in order to avoid being covered by the LLM's pretrained parametric knowledge. However, I think the model's weak performance in such an environment only indicates that it is dominated by a strong physics prior, not necessarily that its inductive reasoning ability is weak. To the best of my knowledge, experiments related to the violation-of-expectation paradigm in psychology [1] have shown that human infants, when encountering phenomena that appear to violate their expectations, also tend to assume their internal world model is correct, rather than overturning their established mental simulation model based on a few sampled trajectories.
>
> > Q1. The authors argue that "inductive physical reasoning is a hallmark of intelligence that humans develop at a very young age." Previous experiments on inductive and physical reasoning in infants were mostly conducted in real-world environments and did not involve scenarios that violate real-world physics. Could the authors provide further human studies to support this claim?
>
> This is an excellent perspective. We provide two counter-arguments against the possibility of a strong physics prior causing our observations — (1) LMMs’ failure in the SB scenario, and (2) connection to cognitive science.
>
> 1. **LMMs fail in SB despite no contradicting physics**. The SB scenario only challenges the visual prior “large = heavy” that LMMs may have. If LMMs had a strong physics prior, they should have succeeded in SB. Instead, they ignore the context (”the small object is heavier”) and fail in SB. A tabular form of the demos’ relation with physics prior is given below:
>
>     | Demos’ relation with physics prior | Performance | Experiment |
>     | --- | --- | --- |
>     | Align | Good | Sec. 4.3, page 7 |
>     | Contradict | Poor | Sec. 4.4, page 7 (all scenarios except SB) |
>     | Does not contradict | Poor | Sec. 4.4, page 7 (SB scenario) |
>
>     This confirms that LMMs lack inductive ability.
>
> 2. **Cognitive experiments show that infants/children adapt to changing environments** after the initial surprise due to violation-of-expectation (VoE):
>     1. [R2]: Infants/toddlers adapted when the friction in opening/closing the drawer was temporally altered. See force adaptation and prior works discussed in the first paragraph of [R2].
>     2. [R3]: Infants adjusted their probabilistic understanding from physical constraints conditioned on color (like Red-LMC and Red-Pass). See the discussion on page 19 of [R3].
>     3. [R4]: Infants “used the habituation event to calibrate their predictions about the test events” when the object sizes in the experiment were changed. See abstract of [R4].
>
>     These works and physical reasoning benchmarks that used VoE are also discussed in App. C (page 22).

---

> > ### Author Response · Authors · 2025-11-21
> > **Response to reviewer q4tW - part 2/2**
> >
> > > W2. There is a huge gap between the benchmark's design and the real world applicability. Real-world scenarios, such as autonomous driving on diverse ground materials mentioned, are still physically consistent and will not feature impossible events like violation of object permanence.
> > >
> >
> > We clarify that the "impossible" scenarios are a **methodological tool**, not the end goal.
> >
> > Through InPhyRe, **we do not aspire to future works that improve LMMs on impossible scenarios, but those that improve LMMs’ ability to infer physical laws from demonstration samples**. However, the ideal way to evaluate IPR using only real-world scenarios requires access to LMM training sets to avoid confounded evaluation. Therefore, we resorted to using impossible physical scenarios to avoid our core evaluation being confounded by possible parametric knowledge.
> >
> > > W3. I think the dataset design is a bit simple. The dataset only includes basic geometric shapes on a flat surface, meaning the model does not need to handle any real-world visual challenges such as lighting and shadows, occlusion, complex backgrounds, or material reflections. Have the authors considered including more diverse object materials (e.g., soft bodies/cloth) and richer shapes?
> > >
> >
> > IPR is agnostic to the nature of the task and the scene. As the first work that evaluates IPR in LMMs, we avoided **scene complexities** that **may confound our core evaluation**. LMMs’ struggle with IPR despite InPhyRe’s simple design suggests poor IPR in more complex scenarios.
> >
> > We have **included new experiments on a more complex version of InPhyRe** that includes changes in lighting directions, camera poses, and background objects in App. E.8 (page 41). We found that LMMs showed similar or worse inductive physical reasoning compared to the original InPhyRe.
> >
> > > W4. The paper lacks further experiments to investigate the cause of this phenomenon, such as visualizing specific attention maps or applying linear probing to inner representations.
> > >
> >
> > We thank the reviewer for this suggestion. Investigating the cause of our findings requires interpreting the LMM outputs — a difficult task that currently lacks sufficiently reliable tools. However, we have **included new preliminary results** in App. F (page 45) using linear probes and attention maps:
> >
> > **Analysis**:
> >
> > 1. Linear probes on hidden states from pre-trained models could accurately predict whether the scenario was regular or irregular. **Conclusion**: This difference in how LMMs perceive regular and irregular scenarios might contribute to the poor IPR performance.
> > 2. A fine-tuned LMM with improved IPR showed adaptive behavior about the information from demonstration and evaluation samples in its hidden states w.r.t. the nature of the scenario, unlike a pre-trained LMM.
> > 3. Across scenarios, LMMs spent **an order of magnitude less** attention on image tokens. This explains the observed language bias.
> >
> > **References**:
> >
> > [R2] L. Faßbender et al., “A comparison of force adaptation in toddlers and adults during a drawer opening task”, Scientific Reports 15 (article 3699), 2025.
> >
> > [R3] S. Denison and F. Xu, “Integrating Physical Constraints in Statistical Inference
> > by 11-Month-Old Infants”, Cognitive Science 34 (p 885–908), 2010.
> >
> > [R4] L. Kotovsky and R. Baillargeon, “The development of calibration-based reasoning about collision events in young infants”, Cognition 67(3) (p 311-351), 1998.
> >
> > ---
> > ## We request the reviewer to raise their rating if we have addressed all their concerns satisfactorily.

---

### Official Review · Reviewer_unjk · 2025-10-30

**Soundness:** 2
**Presentation:** 3
**Contribution:** 1
**Rating:** 2
**Confidence:** 5

**Summary:**

The paper introduces INPHYRE, a synthetic CLEVR‑style benchmark that gives exemplar videos and asks VQA about outcomes of collisions, including rule violations. The authors report three findings. Limited parametric physics knowledge, weak adaptation to unseen rules, and strong language bias.

**Strengths:**

- Clear motivation about adaptation to out‑of‑distribution physics.
- Simple, reproducible pipeline built on PyBullet and Blender with a scenario taxonomy.
- Broad model sweep and a few ablations on number of exemplars, CoT, and quantization.

**Weaknesses:**

- Firstness claim is incorrect or overstated. Prior work already probes counterfactual or violated physics and adaptation in VQA or video settings, for example IntPhys, CLEVRER, CoPhy, ComPhy, Physion and Physion++, ContPhy, PhysBench, Physics Context Builders, and context conditioned physics efforts, and related context driven benchmarks.

- Core novelty is thin. This is another templated synthetic collision suite with attribute rules tied to color. The few‑shot prompting setup is standard. It is unclear why the original CLEVRER dataset cannot be used for this task.

- Evaluation design confounds language bias by construction. Exemplars often include QA pairs, while the test uses only a single image frame. This invites template copying and makes the video‑only drop unsurprising. Claims that models ignore vision are therefore not well supported.

- Many questions require motion information that a single frame cannot provide. Penalizing Not enough data may punish reasonable uncertainty.

- No human baseline, no test of option prior balance, limited statistical analysis, and no side‑by‑side with earlier benchmarks under a shared protocol.

Overall, the paper’s empirical sweep is useful, but the novelty and claims are not strong, and key design choices undermine the main conclusions.

**Questions:**

- What exact boundary makes this benchmark first? If the novelty is exemplar videos plus VQA under rule changes, state that narrowly and revise the claim.

- Why is the query only a first frame? Do results hold if the model sees the full video at test.

---

> ### Author Response · Authors · 2025-11-21
> **Response to Reviewer unjk (part 1/2)**
>
> We thank the reviewer for their critical assessment. To address the concerns regarding task validity and experimental rigor, we have conducted significant additional experiments, including a **Human Baseline (App. E.9)**, **Full-Video Evaluation (App. E.10)**, **Attention Map Analysis (App. F.1)**, and **Prompt Perturbation Analysis (App. E.11)**.
>
> **Summary of Rebuttal:**
>
> - **Novelty (W1, W2, Q1):** InPhyRe is the first benchmark where physical laws **change between training and testing** **and** **requires the model to infer the rule from context** rather than relying on memory.
> - **"Template Copying" (W3):** Models fail even when QA pairs are provided, proving they cannot simply copy templates. Furthermore, attention analysis reveals they ignore visual tokens, confirming language bias is a *finding*, not a design flaw.
> - **Evaluation Setup (W4, Q2):** Our new experiment shows that providing **all video frames** actually *degrades* performance, validating our single-frame design.
> - **Baselines & Rigor (W5):** Humans achieve **~90% accuracy** with *less* context than the models, proving the task is visually solvable. We also confirm robustness to prompt perturbations.
>
> > W1. Firstness claim is incorrect or overstated. Prior work already probes counterfactual or violated physics and adaptation in VQA or video settings, for example IntPhys, CLEVRER, CoPhy, ComPhy, Physion and Physion++, ContPhy, PhysBench, Physics Context Builders, and context conditioned physics efforts, and related context driven benchmarks.
>
> Q1. What exact boundary makes this benchmark first? If the novelty is exemplar videos plus VQA under rule changes, state that narrowly and revise the claim.
> >
>
> The reviewer asks for the "exact boundary" of our novelty. We define this boundary precisely: **InPhyRe is the first benchmark where the physical laws themselves change between training and testing, requiring the model to infer the rule from context rather than memory**. InPhyRe is compared against existing benchmarks below (also included as Tab. 1 on page 3).
>
> | Benchmark | Task | Example query | Do physical conditions/laws change between training and testing? | Require on-the-fly physical condition/law inference from demos/test sample? |
> | --- | --- | --- | --- | --- |
> | CLEVRER | Factual and counterfactual physical reasoning | “What shape is the object that collides with the cyan cylinder?” | No | No |
> | ComPhy | General physical reasoning requiring latent property reasoning | “Which event would happen if the grey cylinder were not magnetic?” | No | Yes |
> | CoPhy | Counterfactual physical reasoning | “what happens if we push this ball to the right?” | No | No (models are trained to do counterfactual reasoning) |
> | PhysBench | General physical reasoning about property, dynamics, relations, etc. | “Which object will the cart hit first?” “What is the object closest to the teacup in the Figure?” | No | No |
> | IntPhys (v1 and v2) | Physical plausibility prediction | Predict the next frame based on the initial frames of some event | Yes | No |
> | Physion | Object contact prediction | The initial frames of a video showing falling dominoes | No | No |
> | Physion++ | Object contact prediction through physical property understanding | The initial frames of a video showing falling dominoes | No | No |
> | ContPhy | Physical property and dynamics prediction | “Is the density of the orange fluid greater than that of the green fluid?” ”What can we do to guide most of the orange fluid into cyan cylinder?” | No | No |
> | PCB | Train smaller VLMs to support models during inference | Not an evaluation benchmark. PCBs provide scene narration to support the inference model. | No | No |
> | **InPhyRe** | Infer physical laws from demo samples and apply them to the evaluation sample. | “Will the velocity of the green cube increase after collision with the red sphere?” | **Yes** | **Yes** |
>
> PCB is Physics Context Builder (V. Balazadeh et al., CVPR 2025), discussed in the revision.
>
> **IPR vs counterfactual physical reasoning (CPR)**: The physical laws do not change between training and testing in CPR. Thus, CPR checks the memory of the LMM. Compared to CPR, IPR measures a stronger form of generalization.
>
> **IPR vs intuitive physical reasoning**: The goal in intuitive physical reasoning is to evaluate physical reasoning in models through its ability to detect violated physics. **Violation detection does not require understanding the differing rules**. In contrast, IPR requires inferring the underlying rules from demos.
>
> We have made these points clear in the revised PDF (Sec. 2, page 3, and refer to lines 1167-1179, page 22).

---

> > ### Author Response · Authors · 2025-11-21
> > **Response to reviewer unjk (part 2/2)**
> >
> > > W2. Core novelty is thin. This is another templated synthetic collision suite with attribute rules tied to color. The few‑shot prompting setup is standard. It is unclear why the original CLEVRER dataset cannot be used for this task.
> > >
> >
> > We explicitly avoided the CLEVRER dataset because it follows standard physics. Using it would confound our evaluation: **we would not know if the model reasoned inductively or simply accessed its training priors**. By using irregular scenarios (Fig. 2), we ensure that success must come from in-context inference.
> >
> > Note that only 2 of 7 scenarios rely on color; the rest rely on pure dynamics.
> >
> > > W3. Evaluation design confounds language bias by construction. Exemplars often include QA pairs, while the test uses only a single image frame. This invites template copying and makes the video‑only drop unsurprising. Claims that models ignore vision are therefore not well supported.
> > >
> >
> > We provide two counter-evidences against the “template copying” hypothesis:
> >
> > 1. If the LMMs were simply copying the templates, they would have achieved good performance in irregular scenarios when QA pairs were available. However, the results in Fig. 4 (page 8) show massive performance drops, proving they fail to map the template to the visual dynamics. Without QA pairs in demos, performance worsened.
> > 2. **Two new lines of evidence for language bias**: (a) LMMs were unable to leverage all the provided evaluation frames (App. E.10, page 43), and (b) LMMs spent an order of magnitude less attention on image tokens (App. F.1, page 45).
> >
> > > W4. Many questions require motion information that a single frame cannot provide. Penalizing Not enough data may punish reasonable uncertainty.
> >
> > > Q2. Why is the query only a first frame? Do results hold if the model sees the full video at test.
> >
> > Using only the first frame of a key event is standard in evaluating physical reasoning, since physical reasoning is a normative task of predicting the outcome of a physical event before it has occurred. Please see our case study on page 1 for an example.
> >
> > We have included **a new experiment where all video frames are included** in the evaluation sample in App. E.10 (page 43). **Finding**: Including all frames resulted in similar or worse performance compared to using only the first frame, potentially due to attention dilution. This is also further evidence of language bias.
> >
> > The difference in accuracy for (all-frames - only-the-first-frame) is shown below:
> >
> > | LMM | LMC | Wall | AMC | Red-LMC | Red-Pass | SB | CC |
> > | --- | --- | --- | --- | --- | --- | --- | --- |
> > | InternVL3-1B | -6.43 | -6.93 | 2.21 | -25.16 | -27.77 | 2.86 | 10.65 |
> > | VideoLLaMA3-2B | 6.48 | -6.53 | 10.30 | 4.16 | 9.67 | -0.45 | 5.68 |
> > | InternVL3-2B | -8.19 | -1.81 | 1.36 | -19.20 | -20.85 | 1.91 | -2.21 |
> > | Gemma 3-4B | -20.30 | -1.11 | 0.65 | -18.65 | -13.98 | 1.86 | -25.93 |
> > | LLaVA-NeXT-Vid | 0.90 | -13.87 | -4.67 | -6.47 | 0.75 | -0.15 | -1.71 |
> > | InternVL3-8B | -28.84 | -5.93 | 0 | -35.79 | -17.59 | -14.27 | -23.27 |
> > | LLaVA-OneVision | -3.62 | -3.32 | 0.15 | -9.77 | -2.51 | 5.18 | -6.98 |
> > | VideoLLaMA3-7B | 8.99 | -17.24 | 2.36 | -6.17 | -11.18 | -0.35 | -3.82 |
> > | LLaVA-NeXT-IL | -40.45 | -10.40 | -4.57 | -43.61 | -45.16 | -10.60 | -17.34 |
> > | Qwen2-VL | -38.44 | 0.05 | 4.97 | -13.63 | -18.75 | -1.21 | 11.31 |
> > | Qwen2.5-Omni | -18.04 | 22.21 | 4.92 | -14.74 | -30.18 | 13.57 | 10.65 |
> > | Gemma 3-12B | -16.88 | -19.20 | -3.57 | -7.22 | -14.34 | -4.67 | -37.49 |
> >
> > > W5. No human baseline, no test of option prior balance, limited statistical analysis, and no side‑by‑side with earlier benchmarks under a shared protocol.
> > >
> >
> > **Added a human baseline** in App. E.9 (page 42). Even with only one demonstration sample, without question-answer pairs, human performance was significantly better than LMMs. They relatively struggled more in Red-LMC, Red-Pass, and CC. The mean performance of 10 human subjects is below:
> >
> > | Scenario | LMC (Reg.) | SB (Reg.) | Wall (Reg.) | LMC | Wall | AMC | Red-LMC | Red-Pass | SB | CC |
> > | --- | --- | --- | --- | --- | --- | --- | --- | --- | --- | --- |
> > | Mean acc. (%) | 90 | 100 | 90 | 70 | 100 | 100 | 60 | 20 | 90 | 60 |
> >
> > To account for any **option prior imbalance**, the options were shuffled for each question. Any non-uniform option prior will also not affect our findings since we are measuring IPR as the accuracy difference between regular and irregular scenarios.
> >
> > **Evaluation of effects of prompt perturbation added** to App. E.11 (page 44). No scenario had the coefficient of variation (CoV) of accuracy outside the acceptable range for the majority of LMMs. CoV was less than 0.1 for most larger models.
> >
> > As we mentioned before, there is no benchmark whose physical laws vary between training and testing, and require the model to infer these laws at test time. Please see the prior works comparison table above.
> >
> > ---
> > ## We request the reviewer to raise their rating if we have addressed all their concerns satisfactorily.

---

### Official Review · Reviewer_nZYb · 2025-11-03

**Soundness:** 3
**Presentation:** 4
**Contribution:** 3
**Rating:** 6
**Confidence:** 3

**Summary:**

This paper's motivation is that LMMs fail at inductive physical reasoning, which is the human-like ability to infer and apply unseen physical laws from a few examples. The authors introduce INPHYRE, a novel VQA benchmark of synthetic collision videos, to measure this skill as distinct from memorized parametric knowledge. The methodology compares LMM performance on regular scenarios (obeying true physics) against irregular ones that algorithmically violate universal laws (e.g., momentum conservation). Key findings show that LMMs not only possess weak parametric knowledge (struggling to apply laws they can recite) but also demonstrate poor inductive reasoning. The limited reasoning they do show is heavily language-biased, relying on textual cues in examples while largely ignoring the visual data.

**Strengths:**

- The paper introduces a new benchmark, INPHYRE, which uses physically impossible videos to test if LMMs can learn new physical laws from examples, rather than just repeating what they memorized. A key finding is that LMMs show a strong language bias; they primarily rely on text descriptions in the examples and struggle to learn from the visual-only videos.

- The study shows that LMMs are not only weak at learning new physical laws from examples but also struggle to correctly apply their existing, memorized knowledge of universal physics.

- The paper is well organized with clear logic. Finding 2 and 3 are interesting.

**Weaknesses:**

- The benchmark uses visually simple scenes with primitive shapes. Whether this failure on simple tasks means LMMs would perform even worse on complex, real-world videos, which are much noisier?

- The finding that LMMs ignore visual input in favor of text points to a serious trustworthiness problem. If models can recite a physical law but fail to apply it visually, does this mean they lack any real world model and are just manipulating text-based facts without genuine understanding?

- Some concurrent work is exploring the use of LMMs to guess world parameters, such as depth[1]. Perhaps the weakness highlighted in Finding 1 is simply because the models were not trained on that specific kind of data. Correct me if I am wrong.

[1] Cai, Z., Yeh, C. F., Xu, H., Liu, Z., Meyer, G., Lei, X., ... & Shi, Y. (2025). DepthLM: Metric Depth From Vision Language Models. arXiv preprint arXiv:2509.25413.

**Questions:**

See in the weakness.

---

> ### Author Response · Authors · 2025-11-21
> **Response to Reviewer nZYb**
>
> We thank the reviewer for their comments and for sharing relevant concurrent works.
>
> **Summary of the rebuttal**:
>
> 1. **Simplicity of the scene (W1)**: To establish the absence of IPR in LMMs without confounding factors, we intentionally used a simple dataset. If an LMM struggles in IPR on a simple scene, it can only perform worse in a more complex scene. New experiments on a more complex version of InPhyRe have been added (App. E.8, page 41).
> 2. **Do LMMs struggle to apply known physical laws to visual inputs? (W2)** Yes. As the reviewer pointed out and as we wrote in line 478, our findings suggest a serious trustworthiness problem in LMMs.
> 3. **Will more training help? (W3)** Although fine-tuning helps (App. E.4, page 26), it is impractical if the user and the LMM provider are not willing to share data and/or weights for training. See the illustrative example provided in the common response.
>
> **Detailed rebuttal**:
>
> > W1. The benchmark uses visually simple scenes with primitive shapes. Whether this failure on simple tasks means LMMs would perform even worse on complex, real-world videos, which are much noisier?
> >
>
> IPR is agnostic to the nature of the task and the scene. As the first work that evaluates IPR in LMMs, we avoided scene complexities that may confound our core evaluation. LMMs struggling despite InPhyRe’s simple design suggests poor IPR in more complex scenarios. Thus, InPhyRe **establishes an upper bound** for IPR.
>
> We have **included new experiments on a more complex version of InPhyRe** that includes changes in lighting directions, camera poses, and background objects in App. E.8 (page 41). LMMs performed similarly or worse in this more complex version.
>
> > W2. The finding that LMMs ignore visual input in favor of text points to a serious trustworthiness problem. If models can recite a physical law but fail to apply it visually, does this mean they lack any real-world model and are just manipulating text-based facts without genuine understanding?
> >
>
> InPhyRe revealed the serious gap that even though LMMs **could recollect the exact physical laws**, they **could not apply these laws for reasoning** (page 29, lines 1537-1538), sometimes making misleading assumptions (page 31, lines 1657-1658).
>
> Inductive physical reasoning significantly improved using chain-of-thought (CoT) prompting (App. E.3, page 26), **further asserting the language bias**. Note that CoT prompting is not possible for arbitrary evaluation scenarios since the underlying physical laws may not be known.
>
> > W3. Some concurrent work is exploring the use of LMMs to guess world parameters, such as depth [1]. Perhaps the weakness highlighted in Finding 1 is simply because the models were not trained on that specific kind of data. Correct me if I am wrong.
> [1] Cai, Z., Yeh, C. F., Xu, H., Liu, Z., Meyer, G., Lei, X., ... & Shi, Y. (2025). DepthLM: Metric Depth From Vision Language Models. arXiv preprint arXiv:2509.25413.
> >
>
> We share three lines of evidence that Finding 1 was not due to any training issues, but rather due to a lack of inductive adaptability.
>
> **Evidence 1**: LMMs performed well in regular scenarios and leveraged demonstration samples in regular scenarios. This shows that they understood the task and the domain.
>
> **Evidence 2**: They failed only when the underlying scenarios challenged their prior information under a similar evaluation setup.
>
> **Evidence 3**: LMMs also performed poorly in the SB scenario, although that scenario did not contradict true physical laws, but only challenged their visual prior “large = heavy.”
>
> We highlight that **IPR is especially critical when the evaluation scenario is outside the training set**. That is, the scenario suggested by the reviewer is our target. Fine-tuning experiments in App. E.4 (page 26) showed that LMMs had the capability to reason in irregular scenarios.
>
> We also thank the reviewer for sharing [1] with us. We highlight that this appeared on ArXiv after the ICLR submission deadline. Nonetheless, as a relevant work, we have discussed [1] in our revision, where we describe our fine-tuning results (App. E.4, page 26).
>
> ---
> ## We request the reviewer to raise their rating if we have addressed all their concerns satisfactorily.

---

### Author Response · Authors · 2025-11-21
**Global Response - Clarification of our objective (part 1/2)**

We thank the reviewers for their comments. Below, we list the strengths of our work as identified by the reviewers:

1. Broad, comprehensive, and extensive evaluation [unjk, q4tW, WEJ2]
2. Interesting findings [nZYb]
3. “Well-organized”, “clearly written” presentation [nZYb, q4tW, WEJ2]

To address concerns regarding task validity, novelty, and rigor, we have added **four major experiments** in the revised PDF:

1. **Human Baseline (App. E.9):** We evaluated 10 human subjects. Even with only one* demonstration and no QA pairs, humans achieved **~90% accuracy** on most scenarios. This definitely proves the task is well-defined and visually solvable.
2. **All-Frames Evaluation (App. E.10):** We tested providing all video frames during evaluation. Performance did not improve (and often degraded), confirming that the bottleneck is **reasoning**, not perception/motion estimation.
3. **Attention Analysis (App. F.1):** We visualized attention maps (e.g., Gemma3-12B), revealing that LMMs attend to image tokens **an order of magnitude less** than text tokens. This provides a structural explanation for the language bias we observed.
4. **Prompt Robustness (App. E.11):** We analyzed sensitivity to prompt perturbations. Results show a low coefficient of variation for larger models, confirming experimental stability.

The concerns raised by reviewers stem from a **misunderstanding of our evaluation goal**. We briefly state our goal and its importance, along with an example below.

**Physical reasoning** is the task of predicting the outcome of a physical event based on initial observations. For example, given the dashcam footage from a car, we wish to predict whether a collision will occur or not, whether the car will enter the opposite lane, and so on. Note that physical reasoning is a **normative task**, i.e., predictions are about events yet to happen from the initial conditions based on some physical rules.

**InPhyRe’s Goal**: Evaluate whether LMMs can infer and apply physical laws from demonstration samples when the underlying rules were not observed during training.

**Comparison of InPhyRe with other datasets**

PCB is Physics Context Builder (V. Balazadeh et al., CVPR 2025).

| Benchmark | Task | Example query | Do physical conditions/laws change between training and testing? | Require on-the-fly physical condition/law inference from demos/test sample? |
| --- | --- | --- | --- | --- |
| CLEVRER | Factual and counterfactual physical reasoning | “What shape is the object that collides with the cyan cylinder?” | No | No |
| ComPhy | General physical reasoning requiring latent property reasoning | “Which event would happen if the grey cylinder were not magnetic?” | No | Yes |
| CoPhy | Counterfactual physical reasoning | “what happens if we push this ball to the right?” | No | No (models are trained to do counterfactual reasoning) |
| PhysBench | General physical reasoning about property, dynamics, relations, etc. | “Which object will the cart hit first?” “What is the object closest to the teacup in the Figure?” | No | No |
| IntPhys (v1 and v2) | Physical plausibility prediction | Predict the next frame based on the initial frames of some event | Yes | No |
| Physion | Object contact prediction | The initial frames of a video showing falling dominoes | No | No |
| Physion++ | Object contact prediction through physical property understanding | The initial frames of a video showing falling dominoes | No | No |
| ContPhy | Physical property and dynamics prediction | “Is the density of the orange fluid greater than that of the green fluid?” ”What can we do to guide most of the orange fluid into cyan cylinder?” | No | No |
| PCB | Train smaller VLMs to support models during inference | Not an evaluation benchmark. PCBs provide scene narration to support the inference model. | No | No |
| **InPhyRe** | Infer physical laws from demo samples and apply them to the evaluation sample. | “Will the velocity of the green cube increase after collision with the red sphere?” | **Yes** | **Yes** |

**Inductive physical reasoning (IPR) vs counterfactual physical reasoning (CPR)**: The physical laws do not change between training and testing in CPR. Compared to CPR, IPR measures a stronger form of generalization. E.g., ComPhy does not check whether the LMM actually understood the underlying physics or made a guess based on a memorized training set.

**IPR vs intuitive physical reasoning**: The goal in intuitive physical reasoning is to evaluate physical reasoning in models through its ability to detect violated physics. **Violation detection does not require understanding the differing rules**. In contrast, IPR requires inferring the underlying rules from demos.

---

> ### Author Response · Authors · 2025-11-21
> **Global Response - Clarification of our objective (part 2/2)**
>
> **Importance of IPR**: In scenarios where data and/or model weights cannot be exchanged for fine-tuning, demonstration samples are the most reasonable method to leverage local data for inference. Consider the following example.
>
> **Example**: NASA wishes to use a flagship model from OpenAI for some astronomical reasoning task. NASA may not share its data with OpenAI for training, and OpenAI may not provide model weights to NASA for any local fine-tuning. Here, the best available solution for NASA is to use their data as demonstration samples through some secure prompting channel [R1].
>
> **Learning from demos vs physics prior?** SB scenario in InPhyRe only challenges the visual prior (”large” = “heavy”) and does not contradict any physical knowledge. LMMs’ failure in SB is evidence that (1) the observed failure is not due to any physics prior, and (2) LMMs fail to infer context from demos (”the smaller object has more mass”).
>
> | Demos’ relation with prior | Performance | Experiment |
> | --- | --- | --- |
> | Align | Good | Sec. 4.3, page 7 |
> | Contradict | Poor | Sec. 4.4, page 7 (all scenarios except SB) |
> | Does not contradict | Poor | Sec. 4.4, page 7 (SB scenario) |
>
> **References**
>
> [R1] I. Gim et al., “Confidential Prompting: Protecting User Prompts from Cloud LLM Providers”, ArXiv 2024.

---

### Author Response · Authors · 2025-11-28
**Reminder about the rebuttal**

Dear reviewers,

Thank you for your constructive feedback on our work.

We posted our rebuttal to your reviews around 6 days ago. We believe our rebuttal sufficiently answers all your concerns. If not, we are happy to discuss further and provide the necessary evidence to convince the reviewers about the importance of our work.

Since there is only about a week left for the discussion period to close, we kindly request the reviewers to let us know if we have adequately addressed their concerns.

---

### Author Response · Authors · 2025-12-01
**Rebuttal Summary for AC (part 1/2)**

We provide a summary of the reviewers' concerns and our responses here. Our work introduces a **new research question:** Can LMMs adapt to novel unseen scenarios by inferring underlying physical laws from a few demonstration examples? We call this “inductive physical reasoning.” Such a capability will significantly expand the deployment scope and effectiveness of LMMs into novel scenarios by simply relying on demonstration samples using this “inductive physical reasoning” rather than the prevailing solution of expensive fine-tuning. The reviewers’ negative response appears to stem from a misunderstanding of our research question and how it differs from the existing works (see the table in the global response).

Our research insights **identify generalization limits and language bias of LMMs** in physical reasoning. We believe our work will inspire the research community to develop solutions to address this limitation.

---

**Short summary**: Our work identified a new drawback in physical reasoning using LMMs, namely, **inductive physical reasoning**, and provided a benchmark to evaluate inductive physical reasoning in LMMs. Many of the reviewers’ concerns:

1. were due to **incorrectly understanding inductive physical reasoning as similar to existing research directions** such as counterfactual physical reasoning (see table in the global response for the differences), and
2. required **clarifications about how we arrived at our conclusions**.

We directly address the reviewers’ concerns **through existing or additional experiments** (listed below).

---

**List of experiments added during rebuttal**:

E1. **Human Baseline (App. E.9):** Even with only one demonstration sample and no QA pairs, ten human subjects achieved **~90% mean accuracy** on most scenarios. This definitively proves the task is well-defined and visually solvable.

E2. **All-Frames Evaluation (App. E.10):** Performance did not improve (and often degraded) when all evaluation frames were provided. This confirms that the bottleneck is **reasoning**, not perception/motion estimation.

E3. **Attention Analysis (App. F.1):** Visualizing attention maps in Gemma3-12B revealed that LMMs attended to image tokens **an order of magnitude less** than text tokens. This provides a structural explanation for the language bias we observed.

E4. **Prompt Robustness (App. E.11):** We analyzed the sensitivity of evaluation accuracy to prompt perturbations. Results show a low coefficient of variation for larger models, confirming experimental stability.

E5. **Visually more complex scene (App. E.8)**: LMMs performed similarly or worse on InPhyRe with visually more complex and diverse scenarios (e.g., background objects, changing camera pose, illumination direction). This suggests that performance can only worsen with further scene complexity.

---

---

**Questions asked by the reviewers and our responses**:

1. [unjk] **Is inductive physical reasoning the same as counterfactual/intuitive physical reasoning?** No. In counterfactual/intuitive physical reasoning, the evaluated model does not have to infer any unseen physical laws. The objective of inductive physical reasoning has never been considered in prior works (see table in global response).
2. [unjk] **Are existing benchmarks sufficient?** No. To measure inductive physical reasoning, (1) the test samples must follow physical laws different from the train samples, and (2) demo samples with consistently different physical laws must be available for the LMM to infer the underlying laws on-the-fly. No such benchmark exists.
3. [nZYb, q4tW, WEJ2] **Aren’t the observations due to a lack of training data with impossible physics?** If the observed poor performance was only due to impossible physics, then LMMs would have performed well in our SB scenario that didn’t violate any true physics. This is evidence that LMMs indeed have a problem with reasoning from demo samples.
4. [nZYb, q4tW, WEJ2] **Will observations from proposed InPhyRe generalize to more complex scenes?** Our observations were made using (a) physically well-understood scenarios, (b) without confounding visual factors. In a more complex scene, performance can only get worse (Exp. E5).
5. [q4tW] **Is the broader goal of InPhyRe to improve LMM reasoning on impossible physics?** No, impossible physics is our methodology, not the end goal. Our broader goal is adaptive physical reasoning using LMMs.
6. [unjk] **Is the observed language bias due to template copying?** If the template copying hypothesis were true, then LMMs would have performed well in irregular scenarios with question-answer pairs in demos by directly matching the answer templates from demo and evaluation samples.

---

> ### Author Response · Authors · 2025-12-01
> **Rebuttal Summary for AC (part 2/2)**
>
> 7. Questions answered using new experiments:
>     1. [unjk] **Why use a single evaluation frame instead of all frames**: Using a single frame is standard in physical reasoning. Our experiments (Exp. E2) also showed that LMMs failed to reason even with all evaluation frames, confirming the observed phenomenon is a reasoning problem.
>     2. [unjk] **Human baseline**: Humans (ten subjects) achieved ~90% mean accuracy in InPhyRe with only one demonstration sample without any question-answer pairs (Exp. E1), showing that the task is well-defined and visually solvable.
>     3. [unjk] **Effect of prompt design**: Our new experiment shows that all scenarios in InPhyRe have an acceptable coefficient of variation (CoV ≤ 0.3) for most LMMs due to prompt perturbations. Additionally, CoV ≤ 0.1 for most large models across scenarios (Exp. E4).
>     4. [q4tW] **Analysis on causes of inductive physical reasoning**: Through our new experiments (Exp. E3), we find: (1) linear probes on hidden states show LMMs perceive regular and irregular scenarios differently, (2) fine-tuned models show adaptive behavior w.r.t. object attributes in evaluation and demonstration samples, and (3) attention maps show that image tokens get an order of magnitude less attention than text tokens. (1) and (2) explain why fine-tuning improved performance on InPhyRe, and (3) explains the observed language bias.
>     5. [q4tW, WEJ2] **Effect of scene simplicity**: With visually more complex scenes (e.g., background objects, changing camera poses, and illumination directions) in InPhyRe, LMMs behaved similarly or worse (e.g, Gemma3-4B acc. dropped by ~23 points), suggesting further scene complexity can lead to worse performance (Exp. E5).
>     6. [WEJ2] **Evaluation of closed-source models**: Included in the original submission (App. F). Closed-source models showed poor inductive physical reasoning, similar to open-source models.

---

### Meta-Review · Area_Chair_vj1L · 2026-01-13

**Summary:**

This submission introduces InPhyRe, a synthetic VQA benchmark to test inductive physical reasoning in LMMs, i.e., inferring and applying previously unseen physical rules from a few in-context demonstrations. Reviewers acknowledge the paper is clearly written and presents an extensive model sweep, and some found the empirical observations on weak adaptation and language bias interesting. However, the overall recommendation towards negtive, driven primarily by concerns about novelty/positioning and benchmark validity. The rebuttal is strong, but given the remaining concerns about conceptual novelty and the results support the claimed takeaways, ACs recommend Reject.

**Reviewer Concerns:**

A key reviewer argues the firstness claim is overstated relative to a large body of prior physics reasoning/violation benchmarks, and that the core contribution appears to be another templated synthetic collision suite with standard few-shot prompting rather than a distinctly new capability test. Multiple reviewers also question whether the observed failures truly isolate inductive reasoning, noting the design may confound language bias and may penalize reasonable uncertainty when motion information is missing.

**Reviewer Scores:**

The scores are 2, 2, 4, 6. The rebuttal adds several helpful experiments (human baseline, all-frames evaluation, attention analysis, prompt robustness, and a more visually complex variant) and clarifies the intended distinction between inductive vs. counterfactual/intuitive physical reasoning. These additions strengthen the empirical package and address some rigor gaps.

---

### Decision · Program_Chairs · 2026-01-26

Reject